

SciPost Phys. Lect. Notes 7 (2019)

# Tangent-space methods for uniform matrix product states

**Laurens Vanderstraeten[1], Jutho Haegeman[1] and Frank Verstraete[1,2]**

**1** Department of Physics and Astronomy, University of Ghent,
Krijgslaan 281, 9000 Ghent, Belgium
**2** Vienna Center for Quantum Science and Technology, Faculty of Physics,
University of Vienna, Boltzmanngasse 5, 1090 Vienna, Austria

## Abstract

In these lecture notes we give a technical overview of tangent-space methods for matrix product states in the thermodynamic limit. We introduce the manifold of uniform matrix product states, show how to compute different types of observables, and discuss the concept of a tangent space. We explain how to variationally optimize ground-state approximations, implement real-time evolution and describe elementary excitations for a given model Hamiltonian. Also, we explain how matrix product states approximate fixed points of one-dimensional transfer matrices. We show how all these methods can be translated to the language of continuous matrix product states for one-dimensional field theories. We conclude with some extensions of the tangent-space formalism and with an outlook to new applications.

doi:[10.21468/SciPostPhysLectNotes.7](10.21468/SciPostPhysLectNotes.7)

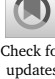
# 1  Introduction

The quantum many-body problem is of central importance in diverse fields of physics such as quantum chemistry, condensed-matter physics and quantum field theory. This is the reason that for the last 90 years most of research in theoretical quantum physics has focused on this problem. In the last decade, the field of quantum information has opened a new viewpoint into this problem by rephrasing it in terms of entanglement theory. It has become clear that equilibrium states of strongly-correlated quantum systems are very special, in the sense that they exhibit area laws for the entanglement entropy. This has led to the introduction of tensor-network states, which can be understood as the most general variational wavefunctions exhibiting such area laws.[1]

The essential property which makes tensor networks appealing is that they allow for an exponential compression of the many-body wavefunction by modeling the entanglement degrees of freedom in the system, rather than the correlation functions as is done in traditional many-body theory. This makes them interesting both from the conceptual and the computational point of view. First, it has allowed to identify the corner of Hilbert space parameterizing ground states of gapped local Hamiltonians, and this has led to the classification of topological phases

---

[1]For a more general introduction to tensor networks, we refer the reader to Refs. [1–5].

of matter for interacting many-body systems. Second, the exponential compression allows to view tensor networks as variational ansätze for which expectation values can be calculated efficiently, and makes them well suited for ground-state calculations of strongly-interacting systems such as the Hubbard model. The ubiquitous density matrix renormalization group (DMRG) is the prime example of this computational approach, and has been used extensively for modeling recent experiments in condensed-matter and atomic physics. DMRG however represents just one aspect of tensor networks. First of all, the matrix product states (MPS) used in that approach can readily be generalized to other geometries and continuum quantum field theories. Second, post-MPS and tensor network methods can be formulated to access spectral and dynamical information about the systems of interest. The natural way of describing those novel tensor network methods is through the low-dimensional manifold that those states span in the full Hilbert space. This manifold picture provides a unifying framework by which both DMRG and time-dependent and spectral MPS methods can be understood.

The main objective of these lecture notes is to highlight the novel aspects of quantum tensor networks that are made apparent by looking at them through the lens of this manifold picture and, more specifically, by studying the tangent spaces of this manifold. Those tangent spaces play a central role as they parameterize the directions in Hilbert space which are accessible from within the manifold. It will be shown that the time-dependent variational principle, a unifying way of looking at both stationary and time-dependent methods for dealing with tensor networks, amounts to projecting the full Hamiltonian on this tangent space. It turns out that the elementary excitations or quasiparticles in the full many-body system can also be understood by such a projection, and even topological nontrivial excitations such as spinons and anyons can be understood within this framework.

The structure of the lecture notes is as follows. In Sec. 2 we introduce uniform matrix product states (MPS) in the thermodynamic limit, compute expectation values and derive canonical forms. In Sec. 3 we discuss the notion of a tangent space on the MPS manifold, and derive efficient parametrizations and associated expressions for the tangent-space projectors. In Sec. 4 these notions are used to develop variational ground-state optimization algorithms, and the vumps algorithm is explained in detail. In Sec. 5 the time-dependent variational principle is derived, and we discuss how the flow equations are integrated. In Sec. 6 we introduce the quasiparticle excitation ansatz, and show how to compute elementary excitation spectra of generic spin chains. Finally, we extend all the previous notions to the case of one-dimensional transfer matrices in Sec. 7, and we look at the continuous version of MPS for one-dimensional field theories in Sec. 8. We close these lecture notes with an outlook to further applications and extensions in Sec. 9.

## 2 Matrix product states in the thermodynamic limit

In this first section, we will introduce the class of translation-invariant MPS in the thermodynamic limit [6, 7]. As we will see, these *uniform matrix product states* have some useful properties that enable us to work with them in a computationally efficient way. In the next sections, we will show that, although most state-of-the-art MPS-based methods rather work on finite systems, working directly in the thermodynamic limit has a number of conceptual and numerical advantages.

### 2.1   Uniform matrix product states, gauge transformations and canonical forms

A uniform MPS in the thermodynamic limit is introduced as

$$|\Psi(A)\rangle = \sum_{\{s\}} \boldsymbol{v}_L^\dagger \left[ \prod_{m\in\mathbb{Z}} A^{s_m} \right] \boldsymbol{v}_R |\{s\}\rangle \,, \tag{1}$$

where $A^s$ is a $D \times D$ matrix for every entry of the index $s$. Alternatively we can interpret the object $A$ as a three-index tensor of dimensions $D \times d \times D$, where $d$ is the dimension of the physical Hilbert space at every site in the chain and $D$ is the so-called *bond dimension*. The latter determines the amount of correlations in the MPS and can be tuned in numerical simulations – it is expected that MPS results for gapped systems are exact in the limit $D \to \infty$, and the complexity of all MPS algorithms scales as $\mathcal{O}(D^3)$.

In these lecture notes, we will make use of the diagrammatic language of tensor networks. In this language we represent tensors by geometrical shapes where the indices are indicated by lines sticking out; whenever two indices of two different tensors are contracted (i.e., summed over), the corresponding legs are connected in the diagram. Using this language, we can represent a uniform MPS as

$$|\Psi(A)\rangle = \ldots - \boxed{A} - \boxed{A} - \boxed{A} - \boxed{A} - \boxed{A} - \ldots. \tag{2}$$

In this representation, the right-hand side is a big tensor, written as a contraction of a number of smaller tensors, describing the coefficient for a given configuration of spins that appears in the superposition in Eq. (1). In the following, we will just use the notation that the right-hand side is the state itself. In this diagrammatic representation, the definition of a uniform MPS is obvious: we just repeat the same tensor $A$ on every site in the lattice, giving rise to a state that is translation invariant by construction.[2]

In Eq. (1) we have also introduced two boundary vectors $\boldsymbol{v}_L^\dagger$ and $\boldsymbol{v}_R$, but, as we work on an infinite system, the boundary conditions will never have any physical meaning. Indeed, translation-invariant MPS for which the boundary conditions do matter are called non-injective, and correspond to macroscopic superpositions (cat states), where the specific superposition is encoded in the boundary vectors. Non-injective MPS tensors appear with measure zero in the space of all possible MPS tensors and are not considered throughout these notes[3]. For injective MPS (the generic case), we will show that the physical properties (expectation values) of the state $|\Psi(A)\rangle$ only depend on the tensor $A$, and therefore the MPS tensor truly describes the bulk properties of the state.

The central object in all our calculations is the transfer operator or *transfer matrix*, defined as

$$E = \sum_{s=1}^{d} A^s \otimes \bar{A}^s = \begin{array}{c} -\boxed{A}- \\ | \\ -\boxed{\bar{A}}- \end{array} \,, \tag{3}$$

which is an operator acting on the space of $D \times D$ matrices. From its definition it follows that the transfer matrix is a completely positive map [9], where the MPS matrices $A^s$ play the role of Kraus operators. The transfer matrix has the property that the leading eigenvalue is a positive number $\eta$, which should be scaled to one by rescaling the MPS tensor as $A \to A/\sqrt{\eta}$ for a proper normalization of the state in the thermodynamic limit. In the generic (i.e. injective) case, this

---

[2]We could introduce states that are translation invariant over multiple sites by working with a repeated unit cell of different matrices $A_1, A_2, \ldots$, and all methods that we will discuss can be extended to the case of larger unit cells (see Sec. 9).

[3]We refer the reader to Ref. [8] for additional details.

leading eigenvalue is non-degenerate[4] and the corresponding left and right fixed points $l$ and $r$, i.e. the leading eigenvectors of the eigenvalue equation

$$\qquad\qquad = \qquad\qquad \text{and} \qquad\qquad = \qquad\qquad \tag{4}$$

are positive matrices. They can be normalized such that $\text{Tr}(lr) = 1$, or, diagrammatically,

$$\qquad\qquad = 1. \tag{5}$$

With these properties in place, the norm of an MPS can be computed as

$$\langle \Psi(\bar{A})|\Psi(A)\rangle = \ldots \qquad\qquad \ldots$$

$$= \left(v_L v_L^\dagger\right)\left(\prod_{m\in\mathbb{Z}} E\right)\left(v_R v_R^\dagger\right). \tag{6}$$

The infinite product reduces to a projector on the fixed points,

$$\lim_{N\to\infty} E^N = \qquad\qquad \tag{7}$$

so that the norm reduces to the overlap between the boundary vectors and the fixed points. We will now choose the boundary vectors such that these overlaps equal unity – there is no effect of the boundary vectors on the bulk properties of the MPS anyway – so that the MPS is properly normalized as $\langle \Psi(\bar{A})|\Psi(A)\rangle = 1$.

Although the state is uniquely defined by the tensor $A$, the converse is not true, as different tensors can give rise to the same physical state. This can be easily seen by noting that the *gauge transform*

$$\qquad A \qquad \to \qquad X^{-1} \qquad A \qquad X \qquad \tag{8}$$

leaves the state in Eq. (1) invariant. In fact, it can be shown [8,10] that this is the only freedom in the parametrization[5], and it can be fixed (partially) by imposing canonical forms on the MPS tensor $A$.

As is well known from DMRG and other MPS algorithms on finite chains, the use of canonical forms helps to ensure the numerical stability of the resulting algorithms, and this extends to algorithms for infinite systems discussed below. First, we can always find a representation of $|\Psi(A)\rangle$ in terms of a new MPS tensor $A_L$

$$\qquad A_L \qquad \to \qquad L \qquad A \qquad L^{-1} \qquad . \tag{9}$$

---

[4]In the non-injective case, there would be additional eigenvalues of magnitude $\eta$, which can have a commensurate phase $\exp(ik2\pi/N)$ for some integer $N$.

[5]As before, again by restricting to the set of injective MPS. The mapping from parameter space to physical space as such acquires the structure of a principal fibre bundle [11].

such that the MPS tensor obeys the following condition

$$(10)$$

The matrix $L$ is found by decomposing the fixed point $l$ of $A$ as $l = L^\dagger L$, because with that choice we indeed find

$$(11)$$

The representation of an MPS in terms of a tensor $A_L$ is called the *left-orthonormal form*. This gauge condition still leaves room for unitary gauge transformations,

$$(12)$$

which can be used to bring the right fixed point $r$ in diagonal form. Similarly, a *right-orthonormal form* $A_R$ can be found such that

$$(13)$$

and where the left fixed point $l$ is diagonal.

These left- and right-orthonormal forms now allow us to define a *mixed gauge* for the uniform MPS. The idea is that we choose one site, the 'center site', bring all tensors to the left in the left-orthonormal form, all the tensors to the right in the right-orthonormal form, and define a new tensor $A_C$ on the center site. Diagrammatically, we obtain the following form

$$|\Psi(A)\rangle = \ldots$$

$$= \ldots \qquad (14)$$

This mixed gauge form has an intuitive interpretation. First of all, we introduce a new tensor $C = LR$ which implements the gauge transform that maps the left-orthonormal tensor into the right-orthonromal one, and which defines the center-site tensor $A_C$:

$$(15)$$

This allows us to rewrite the MPS with only the $C$ tensor on a virtual leg, linking the left- and right orthonormal tensors,

$$|\Psi(A)\rangle = \ldots \qquad (16)$$

In a next step, the tensor $C$ is brought into diagonal form by performing a singular-value decomposition $C = USV^\dagger$, and taking up $U$ and $V^\dagger$ in a new definition of $A_L$ and $A_R$ – remember

that we still had the freedom of unitary gauge transformations on the left- and right-canonical form:

$$
\text{—}A_L\text{—} \rightarrow \text{—}U^\dagger\text{—}A_L\text{—}U\text{—} \qquad \text{and} \qquad \text{—}A_R\text{—} \rightarrow \text{—}V^\dagger\text{—}A_R\text{—}V\text{—}. \tag{17}
$$

The above form of the MPS, with a diagonal $C$, now allows to straightforwardly write down a Schmidt decomposition of the state[6] across an arbitrary bond in the chain:

$$
|\Psi(A)\rangle = \sum_{i=1}^{D} C_i |\Psi_L^i(A_L)\rangle \otimes |\Psi_R^i(A_R)\rangle, \tag{18}
$$

where the states

$$
|\Psi_L^i(A_L)\rangle = \ldots\text{—}A_L\text{—}A_L\text{—}i, \qquad |\Psi_R^i(A_R)\rangle = i\text{—}A_R\text{—}A_R\text{—}\ldots \tag{19}
$$

are orthonormal states on half of the lattice,

$$
\langle\Psi_L^i(\bar{A}_L)|\Psi_L^j(A_L)\rangle = \delta_{ij}, \qquad \langle\Psi_R^i(\bar{A}_R)|\Psi_R^j(A_R)\rangle = \delta_{ij}. \tag{20}
$$

This implies that the diagonal elements $C_i$ in this (diagonal) mixed canonical form are exactly the Schmidt numbers of any bipartition of the MPS. The bipartite entanglement entropy is given by

$$
S = -\sum_i C_i^2 \log\left(C_i^2\right). \tag{21}
$$

Next we discuss how to characterize the overlap or fidelity between two uniform MPS. Given two properly normalized MPS $|\Psi(A_1)\rangle$ and $|\Psi(A_2)\rangle$, the overlap is given by

$$
\langle\Psi(\bar{A}_2)|\Psi(A_1)\rangle = \ldots \quad\cdots = \lim_{N\to\infty} \left(\begin{matrix} \text{—}A_1\text{—} \\ \text{—}\bar{A}_2\text{—} \end{matrix}\right)^N. \tag{22}
$$

This expression is either one (up to a phase factor) or zero, depending on whether $\lambda_{\max}(E_2^1)$, the largest eigenvalue (in magnitude) of this mixed transfer matrix $E_2^1$, is on the unit circle or not. Supposing that we have fixed the relative phase between the two tensors $A_1$ and $A_2$ such that $\lambda_{\max}(E_2^1)$ is positive, the overlap is given by

$$
\langle\Psi(\bar{A}_2)|\Psi(A_1)\rangle = \begin{cases} 0 & \text{if} \quad \lambda(E_2^1) < 1 \\ 1 & \text{if} \quad \lambda(E_2^1) = 1 \end{cases}. \tag{23}
$$

This result is known as the orthogonality catastrophe, according to which states in the thermodynamic limit are either equal or orthogonal. The condition $\lambda_{\max}(E_2^1) = 1$ is indeed sufficient to conclude that there exists a gauge transformation between $A_1$ and $A_2$. A more physical quantity to express whether two MPS in the thermodynamic limit are 'close', is the fidelity per site, which exactly corresponds to

$$
f(A_1, A_2) = \left|\lambda(E_2^1)\right|. \tag{24}
$$

---

[6]This representation corresponds to $\lambda = C$ and $\Gamma^s = C^{-1}A_L^s = A_R^s C^{-1}$ in the notation of Ref. [6].

## 2.2 Truncating a uniform MPS

The mixed canonical form enables us to truncate an MPS efficiently [12], which is one of the primitive tasks in any MPS toolbox. Such a problem typically occurs when one is multiplying a MPS with a matrix product operator (see Sec. 7), for which one is interested in reducing the bond dimension again.

The sum in the above Schmidt decomposition can be truncated, giving rise to a new MPS that has a reduced bond dimension for that bond. This truncation is optimal in the sense that the norm between the original and the truncated MPS is maximized, but the resulting MPS is no longer translation invariant – it has a lower bond dimension on one leg. We can, however, introduce a translation invariant MPS with a lower bond dimension by transforming *every* tensor $A_L$ or $A_R$ as in Eq. 17, but where we have truncated the number of columns in $U$ and $V$, giving rise to the isometries $\tilde{U}$ and $\tilde{V}$. The truncated MPS in the mixed gauge is then given by

$$
|\Psi(A)\rangle_{\text{trunc}} = \ldots - \tilde{U}^\dagger - A_L - \tilde{U} - \tilde{S} - \tilde{V}^\dagger - A_R - \tilde{V} - \ldots
$$

$$
= \ldots - \tilde{A}_L - \tilde{A}_L - \tilde{S} - \tilde{A}_R - \tilde{A}_R - \ldots , \tag{25}
$$

with $\tilde{S}$ the truncated singular values of $C$, and

$$
- A_L - \;\rightarrow\; - \tilde{U}^\dagger - A_L - \tilde{U} - \qquad \text{and} \qquad - A_R - \;\rightarrow\; - \tilde{V}^\dagger - A_R - \tilde{V} - . \tag{26}
$$

This procedure is not guaranteed to find the MPS with a lower bond dimension that is globally optimal, in the sense that it minimizes the error on the global (thermodynamic limit) state. A variational optimization of the cost function

$$
\left\| |\Psi(A)\rangle - |\Psi(\tilde{A})\rangle \right\|^2 \tag{27}
$$

would find the optimal truncated MPS tensor $A$, but the above approximate algorithm has, of course, the advantage of being numerically efficient. In Sec. 3 we will discuss a variational method for optimizing this cost function.

## 2.3 Algorithm for finding canonical forms

Above we have seen that the set of uniform MPS can be parametrized in two different ways:

  (i) the uniform gauge, where we have one tensor $A$ that is repeated on every site in the chain as in Eq. (1), and

 (ii) the mixed gauge, where we have a set of three matrices $\{A_L, A_R, C\}$ obeying the relation (15), specifying the MPS as in Eq. (14).

In the algorithms in these notes we will often need to switch between these two gauges, so that a reliable algorithm is needed for extracting a set $\{A_L, A_R, C\}$ from a given uniform MPS tensor $A$. In principle the above relation (9) yields an algorithm for finding a left-orthonormal tensor $A_L$, and a similar relation yields $A_R$ and $C$. In practice, however, this algorithm is suboptimal in terms of numerical accuracy. While $l$ and $r$ are theoretically known to be positive hermitian matrices (up to a phase), at least one of them will nevertheless have small eigenvalues, say of order $\eta$, if the MPS is supposed to provide a good approximation to an actual state. In practice, $l$ and $r$ are determined using an iterative eigensolver (Arnoldi method) and will only be accurate up to a specified tolerance $\epsilon$, so that hermiticity and positivity of the smallest eigenvalues might be violated and need to be 'fixed'. Upon taking the 'square roots' $L$ and $R$, the

numerical precision will go down to $\min(\sqrt{\epsilon}, \epsilon/\sqrt{\eta})$. Indeed, computing $L$ and $R$ from $l$ and $r$ is analoguous to computing the singular values of a matrix $M$ from the eigenvalues of $M^\dagger M$. Furthermore, gauge transforming $A$ with $L$ or $R$ requires the potentially ill-conditioned inversion of $L$ and $R$, and will typically yield $A_L$ and $A_R$ which violate the orthonormalization condition in the same order $\epsilon/\sqrt{\eta}$. Both problems are resolved by taking recourse to single-layer algorithms, i.e. algorithms that only work on the level of the MPS tensors in the ket layer, and never consider operations for which contractions with the bra layer are needed.

Suppose we are given an MPS tensor $A$, and we want to find the left-orthonormal tensor $A_L$ and the matrix $L$, such that $A_L = L^{-1}AL$.[7] The idea is to solve the equation $LA_L = AL$ iteratively, where in every iteration ($i$) we start from a matrix $L^i$, (ii) we construct the tensor $L^i A$, (iii) we take a QR decomposition to obtain $A_L^{i+1} L^{i+1} = L^i A$, and (iv) we take $L^{i+1}$ to the next iteration. The QR decomposition is represented diagrammatically as

$$-\!\!\underbrace{L^i}\!\!-\!\!\underbrace{A}\!\!- \xrightarrow{QR} -\!\!\underbrace{A_L^{i+1}}\!\!-\!\!\underbrace{L^{i+1}}\!\!- . \tag{28}$$

Because the QR decomposition is unique – in fact, it is made unique by the additional condition that the diagonal elements of the triangular matrix be positive – this iterative procedure is bound to converge to a fixed point for which $L^{(i+1)} = L^{(i)} = L$ and $A_L$ is left orthonormal by construction:

$$-\!\!\underbrace{L}\!\!-\!\!\underbrace{A}\!\!- \xrightarrow{QR} -\!\!\underbrace{A_L}\!\!-\!\!\underbrace{L}\!\!- . \tag{29}$$

The convergence rate of this approach is the same as that of a power method for finding the left fixed point $l = L^\dagger L$ of $A$, which is typically insufficient if the transfer matrix has a small gap. We can however speed up this QR algorithm by, after having found an updated guess $A_L^{i+1}$ according to Eq. (28), further improving the guess $L^{i+1}$ by replacing it with the fixed point $\tilde{L}^{i+1}$ of the map

$$\underbrace{X} \quad \rightarrow \quad \underbrace{X}\,\overset{\displaystyle\boxed{A}}{\underset{\displaystyle\boxed{\bar{A}_L^{i+1}}}{\big|}}\ , \tag{30}$$

which can be found by an Arnoldi eigensolver. Note that we don't need to solve this eigenvalue problem for $\tilde{L}^{i+1}$ to high precision early in the algorithm. In particular, we don't want to restart the eigensolver (and thus only build the Krylov subspace once), as the outer iteration $i$ of the algorithm acts as the restart loop. The resulting algorithm for left orthonormalization is presented in Algorithm 1 and a similar algorithm for right orthonormalization follows readily. Algorithm 2 combines both to impose the mixed gauge with diagonal $C$.

## 2.4 Computing expectation values

Suppose we want to compute the expectation value of an extensive operator

$$O = \frac{1}{|\mathbb{Z}|} \sum_{n \in \mathbb{Z}} O_n, \tag{31}$$

where the extra factor $|\mathbb{Z}|^{-1}$ represents the number of sites, and is introduced to obtain a finite value in the thermodynamic limit – in fact, we are evaluating the density corresponding to operator $O$. Because of translation invariance, we only have to evaluate one term where $O$

---

[7]We apply a slight abuse of notation here: The expressions $AX$ and $XA$, with $A$ an MPS tensor and $X$ a matrix are meant as $A^s X$, $\forall s$.

---

**Algorithm 1** Gauge transform a uniform MPS $A$ into left-orthonormal form

---

1: **procedure** LEFTORTHONORMALIZE$(A, L_0, \eta)$      ▷ Initial guess $L_0$ and a tolerance $\eta$
2:      $L \leftarrow L/\|L\|$      ▷ Normalize $L$
3:      $L_{\text{old}} \leftarrow L$
4:      $(A_L, L) \leftarrow$ QRPOS$(LA)$      ▷ QR decomposition according to Eq. (28)
5:      $\lambda \leftarrow \|L\|, L \leftarrow \lambda^{-1}L$      ▷ Normalize new $L$ and save norm change
6:      $\delta \leftarrow \|L - L_{\text{old}}\|$      ▷ Compute measure of convergence
7:      **while** $\delta > \eta$ **do**      ▷ Repeat until converged to specified tolerance
8:          $(\sim, L) \leftarrow$ ARNOLDI$(X \rightarrow E(X), L, \delta/10)$      ▷ Compute fixed point of transfer map in
    Eq. (30) using initial guess $L$, up to a tolerance depending on $\delta$
9:          $(\sim, L) \leftarrow$ QRPOS$(L)$
10:          $L \leftarrow L/\|L\|$
11:          $L_{\text{old}} \leftarrow L$
12:          $(A_L, L) \leftarrow$ QRPOS$(LA)$      ▷ QR decomposition according to Eq. (28)
13:          $\lambda \leftarrow \|L\|, L \leftarrow \lambda^{-1}L$
14:          $\delta \leftarrow \|L - L_{\text{old}}\|$
15:      **end while**
16:      **return** $A_L, L, \lambda$
17: **end procedure**

---

**Algorithm 2** Find mixed gauge $\{A_L, A_R, C\}$ from a uniform MPS tensor $A$

---

1: **procedure** MIXEDCANONICAL$(A, \eta)$      ▷ Initial guesses $L_0$ and $C_0$ and a tolerance $\eta$
2:      $(A_L, \sim, \lambda) \leftarrow$ LEFTORTHONORMALIZE$(A, L_0, \eta)$      ▷ Algorithm 1
3:      $(A_R, C, \sim) \leftarrow$ RIGHTORTHONORMALIZE$(A_L, C_0, \eta)$      ▷ Analoguous to Algorithm 1
4:      $(U, C, V) \leftarrow$ SVD$(C)$      ▷ Diagonalize $C$ matrix
5:      $A_L \leftarrow U^\dagger A_L U$      ▷ Transform $A_L$ according to Eq. (17)
6:      $A_R \leftarrow V^\dagger A_R V$      ▷ Transform $A_R$ according to Eq. (17)
7:      **return** $A_L, A_R, C, \lambda$
8: **end procedure**

---

acts on an arbitrary site. The expectation value is then – assuming the MPS is already properly normalized

$$\langle \Psi(\bar{A}) | O | \Psi(A) \rangle = \ldots \quad \ldots . \quad (32)$$

We can now use the left and right fixed points of the transfer matrix to contract everything to the left and to the right of the operator, to arrive at the contraction

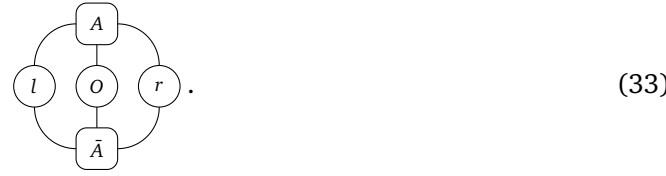

$$(33)$$

An even easier contraction is obtained by going to the mixed gauge, and locating the center site where the operator is acting. Indeed, then everything to the left and right is contracted to the identity and we obtain

$$\begin{matrix} A_C \\ O \\ \bar{A}_C \end{matrix} \quad . \tag{34}$$

A two-site operator such as a hamiltonian term $h$ is evaluated as

$$\langle \Psi(\bar{A}) | \, h \, | \Psi(A) \rangle = \; l \; \begin{matrix} A & A \\ h \\ \bar{A} & \bar{A} \end{matrix} \; r \; = \; \begin{matrix} A_L & A_C \\ h \\ \bar{A}_L & \bar{A}_C \end{matrix} \; = \; \begin{matrix} A_C & A_R \\ h \\ \bar{A}_C & \bar{A}_R \end{matrix} \quad . \tag{35}$$

Correlation functions are computed similarly. Let us look at

$$c^{\alpha\beta}(m,n) = \langle \Psi(\bar{A}) | \, (O_m^\beta)^\dagger O_n^\alpha \, | \Psi(A) \rangle \,, \tag{36}$$

where $m$ and $n$ are arbitrary locations in the chain, and, because of translation invariance, the correlation function only depends on the difference $m-n$. Again, we contract everything to the left and right of the operators by inserting the fixed points $l$ and $r$, so that

$$c^{\alpha\beta}(m,n) = \; l \; \begin{matrix} A & A \\ O^\alpha \\ \bar{A} & \bar{A} \end{matrix} \; \cdots \; \begin{matrix} A & A \\ O^\beta \\ \bar{A} & \bar{A} \end{matrix} \; r \; . \tag{37}$$

From this expression, we learn that it is the transfer matrix that determines the correlations in the ground state. Indeed, if we apply the eigendecomposition,

$$\left( \begin{matrix} A \\ \bar{A} \end{matrix} \right)^n = \; r \; l \; + \sum_i \lambda_i^n \; \lambda_i \; \lambda_i \; , \tag{38}$$

we can see that the correlation function reduces to

$$c^{\alpha\beta}(m,n) = \; l \; \begin{matrix} A \\ O^\alpha \\ \bar{A} \end{matrix} \; r \; \times \; l \; \begin{matrix} A \\ O^\beta \\ \bar{A} \end{matrix} \; r \; + \sum_i (\lambda_i)^{m-n-1} \; l \; \begin{matrix} A \\ O^\alpha \\ \bar{A} \end{matrix} \; \lambda_i \; \times \; \lambda_i \; \begin{matrix} A \\ O^\beta \\ \bar{A} \end{matrix} \; r \; . \tag{39}$$

The first part is just the product of the expectation values of $O^\alpha$ and $O^\beta$, called the disconnected part of the correlation function, and the rest is an exponentially decaying part. This expression implies that connected correlation functions of an MPS *always* decay exponentially, which is one of the reasons why MPS are not well suited for capturing critical states. The largest $\lambda$, i.e. the second largest eigenvalue of the transfer matrix, determines the correlation length $\xi$ and the pitch vector of the correlations $Q$ as[8]

$$\xi = -\frac{1}{\log|\lambda_{\max}|} \quad \text{and} \quad Q = \arg(\lambda_{\max}). \tag{40}$$

---

[8]The $\xi_i$ and $Q_i$ corresponding to the subleading eigenvalues typically have a physical meaning as well, because they point to subleading correlations in the system. Especially in the case where incommensurate correlations are formed, it is instructive to inspect the full spectrum of the transfer matrix [13]. The correlation length is a particularly hard quantity to converge in MPS simulations, but efficient extrapolations have been devised that work directly in the thermodynamic limit [14].

## 2.5 The static structure factor

In experimental set-ups, one typically has access to the Fourier transform of the correlation function, called the (static) *structure factor*. Since we are working in the thermodynamic limit, we can easily compute this quantity with a perfect resolution in the momentum range $[0, 2\pi)$.

The structure factor corresponding to two operators $O^\alpha$ and $O^\beta$ is defined as

$$s^{\alpha\beta}(q) = \frac{1}{|\mathbb{Z}|} \sum_{m,n \in \mathbb{Z}} e^{iq(m-n)} \langle \Psi(\bar{A})| (O^\beta_n)^\dagger O^\alpha_m |\Psi(A)\rangle_c \,, \tag{41}$$

where $\langle \dots \rangle_c$ denotes that we only take the connected part of the correlation function. This can be implemented by redefining the operators such that their expectation value is zero,

$$O^{\alpha,\beta}_n \to O^{\alpha,\beta}_n - \langle \Psi(\bar{A})| O^{\alpha,\beta}_n |\Psi(A)\rangle \,. \tag{42}$$

This quantity can be computed directly in momentum space by a number of steps. First, due to translation invariance, every term in Eq. (41) only depends on the difference $(m-n)$, so that we can eliminate one of the two sums,

$$s^{\alpha\beta}(q) = \sum_{n \in \mathbb{Z}} e^{-iqn} \langle \Psi(\bar{A})| (O^\beta_n)^\dagger O^\alpha_0 |\Psi(A)\rangle_c \,. \tag{43}$$

Every term in the sum has the form of a connected correlation function of the form

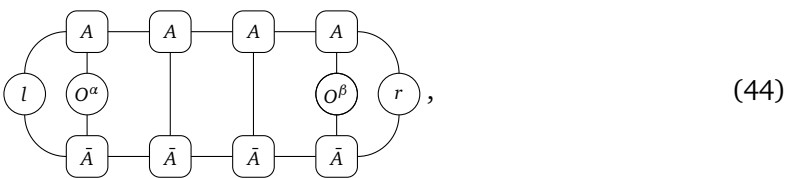

but we can resum all these diagrams in an efficient way. Indeed, all terms where the operator $O^\beta$ is to the right of $O^\alpha$ can be rewritten as

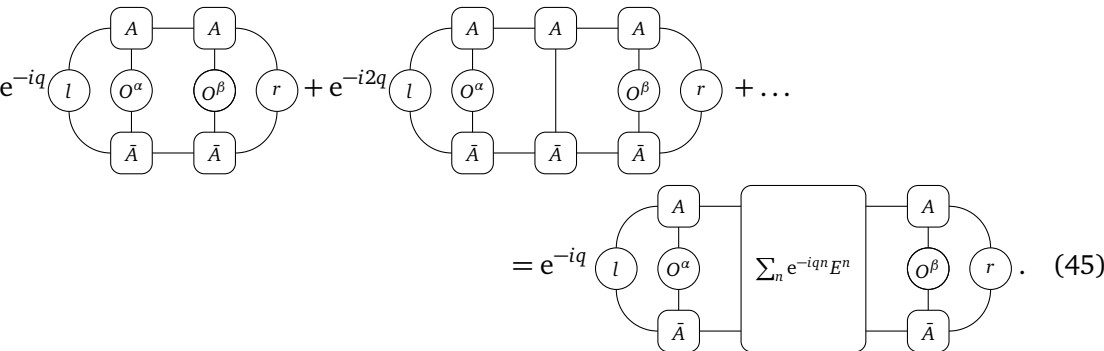

The geometric series are worked out as

$$e^{-iq} \sum_{n=0}^{\infty} e^{-iqn} E^n = e^{-iq} \sum_{n=0}^{\infty} e^{-iqn} \tilde{E}^n + e^{-iq} \sum_{n=0}^{\infty} e^{-iqn} P^n \tag{46}$$

$$= e^{-iq} \left(1 - e^{-iq} \tilde{E}\right)^{-1} + P \sum_{n=1}^{\infty} e^{-iqn} \,,$$

where we have defined a regularized transfer matrix $\tilde{E}$ as

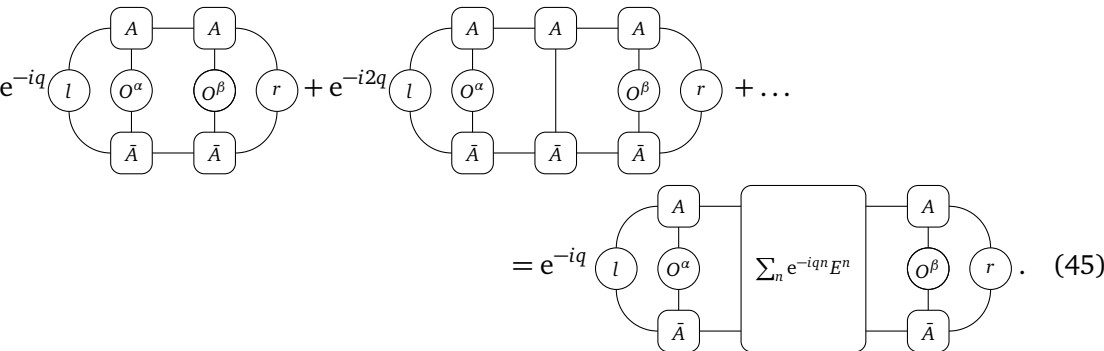

$$\tag{47}$$

and $P$ is the projector on the fixed points. Since the spectral radius of $\tilde{E}$ is strictly smaller than one, the geometric series converges and we can replace it with the inverse $(1 - e^{-iq}\tilde{E})^{-1}$. The second term above containing the projector $P$ could lead to a divergent part, but does not contribute because we have

$$
\langle\Psi(\bar{A})| O^\alpha |\Psi(A)\rangle \langle\Psi(\bar{A})| O^\beta |\Psi(A)\rangle , \tag{48}
$$

which we have set to zero in the definition of the operators $O^\alpha$ and $O^\beta$. We define a 'pseudo-inverse' of an operator as

$$
\left(1 - e^{-iq}\tilde{E}\right)^{-1} = \left(1 - e^{-iq}E\right)^P , \tag{49}
$$

implying that we project out the fixed point of $E$, and take the inverse of the operator on the complement.

The part where $O^\beta$ is to the left of $O^\alpha$ is treated similarly, and we also have the term where both are acting on the same site; we obtain the following final expression:

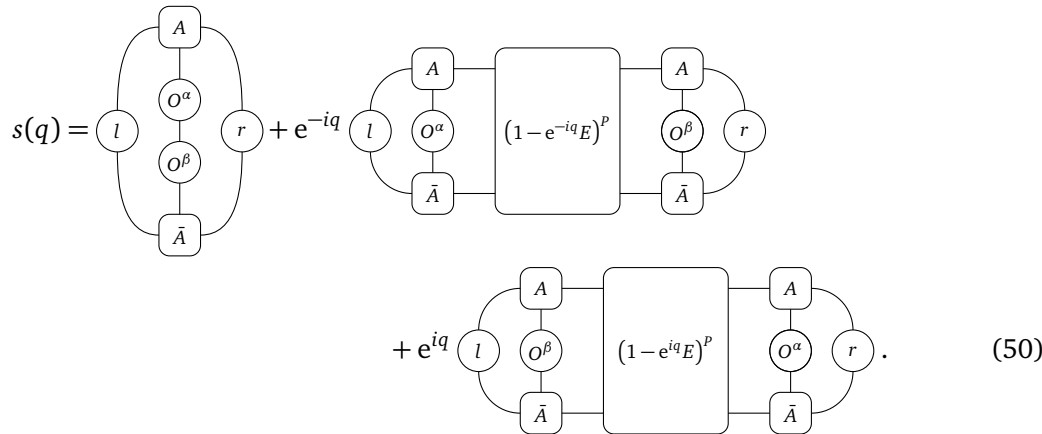

$$
\tag{50}
$$

Note that we don't have to compute the pseudo-inverse $(1 - e^{-iq}E)^P$ explicitly – that would entail a computational complexity of $\mathcal{O}(D^6)$. Instead, we will compute e.g. the partial contraction

$$
\tag{51}
$$

i.e. the action of the pseudo-inverse $(1 - e^{-iq}E)^P$ on a given left-hand side, as the solution of a linear problem of the form $(1 - e^{-iq}\tilde{E}) \times x = y$. This linear equation can then again be solved using Krylov-based interative methods (generalized minimal residual or biconjugate gradient methods), where only the action of $(1 - e^{-iq}\tilde{E})$ on a given vector needs to implemented. This reduces the computational complexity to only $\mathcal{O}(D^3)$.

We can compute the right-side partial contraction in the same way,

$$
\tag{52}
$$

so that we can compute the structure factor as a simple contraction

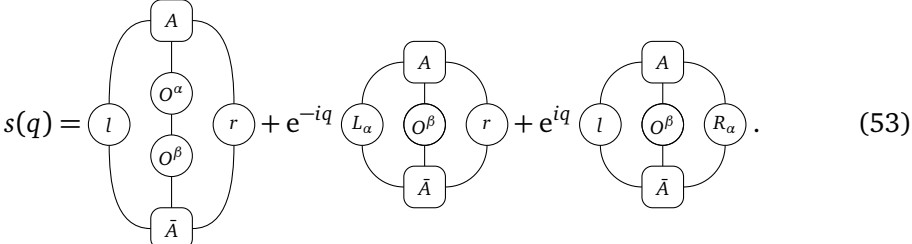

$$s(q) = \underbrace{\left( \begin{array}{c} A \\ O^\alpha \\ O^\beta \\ \bar{A} \end{array} \right)}_{l \quad r} + e^{-iq} \underbrace{\left( \begin{array}{c} A \\ O^\beta \\ \bar{A} \end{array} \right)}_{L_\alpha \quad r} + e^{iq} \underbrace{\left( \begin{array}{c} A \\ O^\beta \\ \bar{A} \end{array} \right)}_{l \quad R_\alpha}. \tag{53}$$

In the mixed canonical form, this expression reduces to

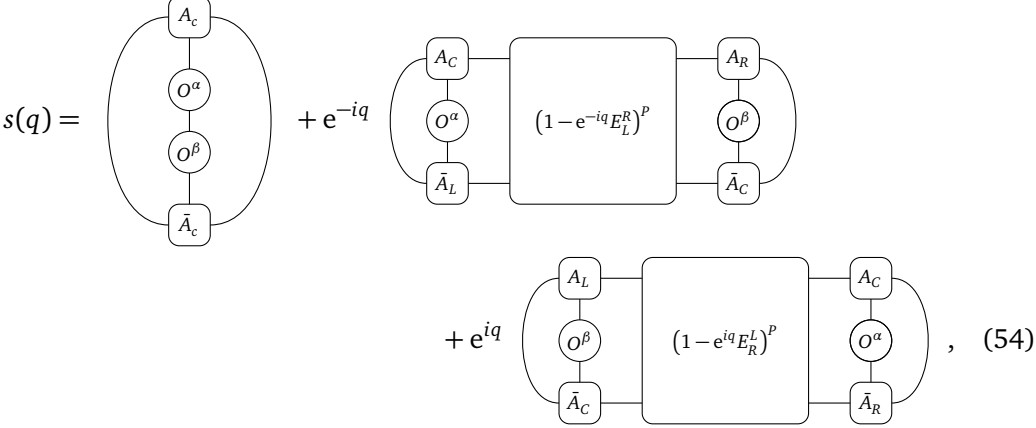

$$s(q) = \left( \begin{array}{c} A_c \\ O^\alpha \\ O^\beta \\ \bar{A}_c \end{array} \right) + e^{-iq} \left( \begin{array}{ccc} A_C & & A_R \\ O^\alpha & \left(1 - e^{-iq}E_L^R\right)^P & O^\beta \\ \bar{A}_L & & \bar{A}_C \end{array} \right)$$

$$+ e^{iq} \left( \begin{array}{ccc} A_L & & A_C \\ O^\beta & \left(1 - e^{iq}E_R^L\right)^P & O^\alpha \\ \bar{A}_C & & \bar{A}_R \end{array} \right), \tag{54}$$

where we have introduced the notations for the transfer matrices

$$E_L^L = \begin{array}{c} A_L \\ \bar{A}_L \end{array}, \qquad E_L^R = \begin{array}{c} A_R \\ \bar{A}_L \end{array}, \qquad E_R^L = \begin{array}{c} A_L \\ \bar{A}_R \end{array}, \qquad E_R^R = \begin{array}{c} A_R \\ \bar{A}_R \end{array}. \tag{55}$$

Here, we have chosen to associate the location of the center site in ket (bra) with the position where $O^\alpha$ ($O^\beta$) acts, as this will generalize when discussing quasiparticle excitations in Sec.6.

## 3 The tangent space and tangent vectors

Let us now introduce the unifying concept of these lecture notes: the *MPS tangent space*. First, we interpret the set of uniform MPS with a given bond dimension as a manifold [11] within the full Hilbert space of the system we are investigating. The manifold is defined by the map between the set of $D \times d \times D$ tensors $A$ and physical states in Hilbert space $|\Psi(A)\rangle$. The resulting manifold is not a linear subspace as any sum of two MPS with a given bond dimension $D$ clearly does not remain within the manifold. Therefore, it makes sense to associate a tangent space [15] to every point $|\Psi(A)\rangle$. By differentiating with respect to the parameters in $A$, an (overcomplete) basis for this tangent space is obtained. The MPS manifold, with a tangent space associated to every point, is represented graphically in Fig. 1.

A *tangent vector* is defined as

$$|\Phi(B;A)\rangle = B^i \frac{\partial}{\partial A^i} |\Psi(A)\rangle = B^i |\partial_i \Psi(A)\rangle, \tag{56}$$

where $i$ is a collective index (combined physical and virtual indices) for all entries of the tensor $A$ and is summed over as in the summation convention. The new tensor $B$ describes a general

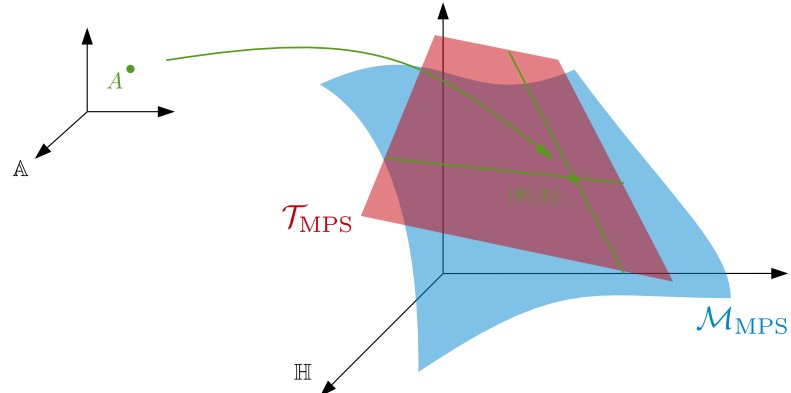

Figure 1: The tangent space on the MPS manifold.

linear combination of the partial derivatives and parametrizes the full tangent space; obviously, the tangent space is a linear subspace of the Hilbert space. The overlap between two tangent vectors can be written as

$$\langle\Phi(\bar{B}';\bar{A})|\Phi(B;A)\rangle = \bar{B}^{i\prime}G_{ij}(\bar{A},A)B^{j}, \tag{57}$$

where $G_{ij}(\bar{A},A) = \langle\partial_i\Psi(\bar{A})|\partial_j\Psi(A)\rangle$ is the Gram matrix or the metric on the tangent space as parametrized by the tensor $B$. As we will see later on, this Gram matrix is singular because of the over-completeness of the basis of partial derivatives, which can be traced back to the gauge redundancy in the MPS description.

In the following sections, we often need an expression for the projector that implements an orthogonal projection of a given vector $|\chi\rangle$ in Hilbert space onto the tangent space. This expression is found by realizing that, due to the Euclidean inner product in Hilbert space, we are in fact looking for the tangent vector $|\Phi(B,A)\rangle$ which maximizes the overlap with the vector $|\chi\rangle$, or

$$\min_{B}\left\||\chi\rangle - |\Phi(B;A)\rangle\right\|^{2}. \tag{58}$$

In the following we will see that this condition implies that the tangent-space projector formally looks like

$$P_A \sim |\partial_i\Psi(A)\rangle (G^{-1})^{ij}\langle\partial_j\Psi(\bar{A})|. \tag{59}$$

As the Gram matrix is singular in general, this expression is not well-defined, and we first have to find a good parametrization of the tangent space that eliminates all singular parts. In the following two subsections we work out two different parametrizations, describe the properties and derive the expressions for the corresponding tangent-space projectors.

### 3.1 Tangent vectors in the uniform gauge

If we work in the uniform gauge, the MPS is parametrized by a single tensor $A$, and a general tangent vector has the form

$$|\Phi(B;A)\rangle = B^{i}\frac{\partial}{\partial A_{i}}|\Psi(A)\rangle = \sum_{n}\ldots \boxed{A}\!-\!\boxed{A}\!-\!\boxed{B}\!-\!\boxed{A}\!-\!\boxed{A}\ldots. \tag{60}$$

The over-completeness in the parametrization of tangent vectors follows from studying the infinitesimal gauge transform $G = \mathrm{e}^{\epsilon X}$ of $|\Psi(A)\rangle$. To first order, we obtain

$$A^{s} \rightarrow \mathrm{e}^{-\epsilon X}A^{s}\mathrm{e}^{\epsilon X} = A^{s} + \epsilon\left(A^{s}X - XA^{s}\right) + \mathcal{O}(\epsilon^{2}), \tag{61}$$

which can be brought to the level of states,

$$|\Psi(A)\rangle \rightarrow |\Psi(A)\rangle + \epsilon |\Phi(B;A)\rangle + \mathcal{O}(\epsilon^2), \tag{62}$$

with $B^s = A^s X - X A^s$. But, since this is a gauge transform in the MPS manifold, the tangent vector $|\Phi(B;A)\rangle$ should be zero. This implies that every transformation on $B$ of the form

$$-\boxed{B}- \rightarrow -\boxed{B}- + -(X)-\boxed{A}- - -\boxed{A}-(X)-, \tag{63}$$

with $X$ an arbitrary $D \times D$ matrix, leaves the tangent vector $|\Phi(B;A)\rangle$ invariant. This gauge freedom can be easily checked by substituting this form in the state (60), and observing that all terms cancel, leaving the state invariant.

The gauge degrees of freedom can be eliminated by imposing a gauge-fixing condition, which can again be chosen so as to be useful from an algorithm perspective. The easiest choice is the so-called *left gauge-fixing condition* (there is of course a right one, too), given by

$$\tag{64}$$

Note that this corresponds to $D^2$ scalar complex-valued equations, whereas we have $D^2$ complex parameters in the gauge transform $X$ in Eq. (63). However, the component $X \sim \mathbb{1}$ does not actually modify $B$ and should therefore not be counted. If we try to explicitly transform a given $B$ according to Eq. (63) so that it satisfies the left gauge-fixing condition [Eq. 64], this amounts to the equation

$$\tag{65}$$

or

$$\tag{66}$$

As the left hand side of this equation is annihilated when contracting with $r$, it can only have a solution for $X$ if also the right hand side satisfies

$$\tag{67}$$

This is, however, a natural condition, because this is precisely saying that the tangent vector is orthogonal to the original MPS. Indeed, one can easily see that the overlap between an MPS and a tangent vector is given by

$$\langle \Psi(\bar{A})|\Phi(B;A)\rangle = 2\pi\delta(0) \tag{68}$$

The factor corresponds to the system size, which diverges in the thermodynamic limit. We will denote this diverging factor as $2\pi\delta(0)$, inspired by the representation of the $\delta$ function as

$$\sum_{n\in\mathbb{Z}} e^{ipn} = 2\pi\delta(p). \tag{69}$$

For a tangent vector $|\Phi(B,A)\rangle$ that is orthogonal to $|\Psi(A)\rangle$, we can then always find a parametrization in terms of a tensor $B$ satisfying Eq. (63) by solving the linear system for the gauge transorm $X$ using $(1-\tilde{E})^{-1} = (1-E)^P$ as regularized inverse, exactly as we have seen in Sec. 2.5.

The restriction to tangent vectors that are orthogonal to the original MPS is crucial in several of the algorithms that follow. In fact, we can implement the left gauge-fixing condition [Eq. (64)] explicitly by constructing an effective parametrization for the $B$ tensor that automatically fixes all gauge degrees of freedom, and which has some nice advantages for all later calculations. First, we construct the tensor $V_L$ such that

$$\vcenter{\hbox{[diagram with $V_L$, $l^{1/2}$, $\bar{A}$]}} = 0, \tag{70}$$

where the right index of $V_L$ has dimension $D(d-1)$. Put differently, $V_L$ corresponds to the $D(d-1)$-dimensional null space of the matrix

$$\vcenter{\hbox{[diagram with $l^{1/2}$, $\bar{A}$]}}. \tag{71}$$

We orthonormalize $V_L$ as

$$\vcenter{\hbox{[diagram with $V_L$, $\bar{V}_L$]}} = \vcenter{\hbox{[diagram]}}. \tag{72}$$

Next, the $B$ tensor is expressed in terms of a new matrix $X$ as

$$-\boxed{B}- = -\left(l^{-\frac{1}{2}}\right)-\boxed{V_L}-\boxed{X}-\left(r^{-\frac{1}{2}}\right)-, \tag{73}$$

where $X$ is a $(D(d-1)\times D)$-dimensional tensor. This parametrization of the $B$ tensor constitutes an effective parametrization for the tangent space that automatically enforces the left-gauge fixing condition and eliminates all degrees of freedom.

Yet, the great advantage of this particular choice of effective parametrization concerns the overlap between two tangent vectors. The overlap between $|\Phi(B;A)\rangle$ and $|\Phi(B';A)\rangle$ is computed similarly to the structure factor in Sec. 2.5: we have two infinite terms, but we can eliminate one sum because of the translation invariance of the MPS; this sum will again result in a factor $2\pi\delta(0)$. There still remains a sum over all relative positions between $B$ and $B'$. Now the power of the left gauge fixing condition is revealed: all terms vanish, except the term where $B$ and $B'$ are on the same site. Indeed, all terms of the form

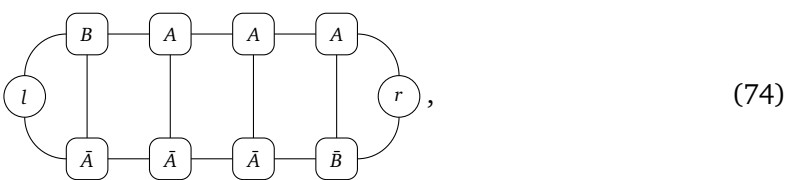

$$, \tag{74}$$

are automatically zero because of Eq. 64. Consequently, the norm reduces to

$$\langle \Phi(\bar{B}';\bar{A})|\Phi(B;A)\rangle = 2\pi\delta(0) \quad (75)$$

or in terms of the effective parameters in $X$ and $X'$,

$$\langle \Phi(\bar{B}(X');\bar{A})|\Phi(B(X);A)\rangle = 2\pi\delta(0) \quad = 2\pi\delta(0) \quad = 2\pi\delta(0)\mathrm{Tr}\big((X')^{\dagger}X\big). \quad (76)$$

The fact that the overlap of tangent vectors reduces to the Euclidean inner product on the effective parameters $X$ and $X'$ will prove to be a very useful property in all tangent-space algorithms. More formally, this implies that the Gram matrix (see Eq. (57)) for the tangent space as parametrized by the matrix $X$ reduces to the unit matrix.

With this effective parametrization of the tangent space in place, we can now derive the tangent-space projector $\mathcal{P}_A$, i.e. the operator that orthogonally projects any state $|\chi\rangle$ onto the tangent space associated to a given MPS $|\Psi(A)\rangle$. The orthogonal projection on a linear subspace of Hilbert space is equivalent to minimizing

$$\min_{X} \big\| |\chi\rangle - |\Phi(B(X);A)\rangle \big\|^2 = \min_{X} \Big( \langle \Phi(\bar{B}(X);\bar{A})|\Phi(B(X);A)\rangle$$
$$- \langle \chi|\Phi(B(X);A)\rangle - \langle \Phi(\bar{B}(X);\bar{A})|\chi\rangle \Big). \quad (77)$$

As this minimization problem is quadratic in $X$ and $\bar{X}$ with a quadratic term $\mathrm{Tr}(X^{\dagger}X)$, the solution is given by $X = \partial_{\bar{X}}(\dots)$. Since the overlap between two tangent vectors is given by Eq. (76), the solution of the minimization problem is found as

$$2\pi\delta(0)X = \frac{\partial}{\partial\bar{X}}\langle \Phi(\bar{B}(X);\bar{A})|\chi\rangle, \quad (78)$$

or, if $|\chi\rangle$ is translation invariant we can cancel the $2\pi\delta(0)$ factors,

$$\boxed{X} = \dots \quad (79)$$

The vector that results from the tangent-space projector should again be of the form of Eq. 60, so we transform the above $X$ tensor back into the form of a $B$ tensor according Eq. 73, such that we have

$$|\Phi(B(X);A)\rangle = \sum_{n} \dots \quad (80)$$

yielding the final form of the tangent space projector as

$$
\mathcal{P}_A = \sum_n \ldots \qquad (81)
$$

### 3.2 Tangent vectors in the mixed gauge

The above tangent-space projector contains inverses of $l$ and $r$, which are potentially ill-conditioned. Therefore, we also derive the expression for the projector in the mixed gauge. We first write down a tangent vector in the mixed gauge:

$$
|\Phi(B;A_L,A_R)\rangle = \sum_n \ldots \qquad (82)
$$

The crucial difference with the standard form of the tangent vector is that the elements of $B$ are now not directly related to derivatives with respect to the parameters in the MPS tensors $A_L$ and $A_R$, and that we need to derive the projector onto the tangent space in a slightly more involved way.

As we still have the gauge freedom $B \rightarrow B + A_L X - X A_R$, we again start by imposing the left-gauge fixing condition, which now has the simpler form

$$
= = 0. \qquad (83)
$$

We define the effective parametrization of the tangent vector in terms of the matrix $X$ as

$$
\text{—}\boxed{B}\text{—} = \text{—}\boxed{V_L}\text{—}\boxed{X}\text{—} , \qquad (84)
$$

where the tensor $V_L$ obeys the usual conditions

$$
= 0 \qquad \text{and} \qquad = . \qquad (85)
$$

Interpreting $A_L$ as the first $D$ columns of a $(Dd) \times (Dd)$ unitary matrix, $V_L$ corresponds to the remaining $D(d-1)$ columns thereof. This effective parametrization has the same useful properties as before, but now does not require taking any inverses of $l$ or $r$.

Suppose now again we have an abritrary state $|\chi\rangle$, which we want to project on a given tangent vector $|\Phi(B;A_L,A_R)\rangle$. This is equivalent to minimizing

$$
\min_X \big\| |\chi\rangle - |\Phi(B(X);A_L,A_R)\rangle \big\|^2 = \min_X \Big( \langle\Phi(\bar{B}(X);\bar{A}_L,\bar{A}_R)|\Phi(B(X);A_L,A_R)\rangle
$$

$$
- \langle\chi|\Phi(B(X);A_L,A_R)\rangle - \langle\Phi(\bar{B}(X);\bar{A}_L,\bar{A}_R)|\chi\rangle \Big). \quad (86)
$$

We have a quadratic minimization problem as before and, since the Gram matrix with respect to the effective tangent-space parametrization $X$ is again the unit matrix, the solution of the minimization problem is found as

$$2\pi\delta(0)X = \frac{\partial}{\partial\bar{X}}\langle\Phi(\bar{B}(X);\bar{A}_L,\bar{A}_R)|\Psi\rangle\,, \tag{87}$$

which yields



$$\tag{88}$$

The corresponding tangent vector is

$$|\Phi(B;A_L,A_R)\rangle = \sum_n \ldots \tag{89}$$

This form of the projector can be cast into an even more useful form for the tangent-space algorithms below, by rewriting the projector on $V_L$ as

$$\tag{90}$$

so that the final form of the tangent space projector is given by

$$P_{\{A_L,A_R\}} = \sum_n \ldots \tag{91}$$

In contrast to the simpler form of the previous section, in the mixed canonical representation the tangent-space projector has two different terms, but, precisely because we work with a mixed canonical form, the projector does not require the inversion of potentially ill-conditioned matrices $l^{-1/2}$ and $r^{-1/2}$.

## 3.3 Variational optimization of the overlap

The concept of a tangent space and its associated projector now allow us to formulate variational MPS methods that implement energy optimization for ground-state approximations, real-time evolution within the MPS manifold, or low-energy excitations on top of an MPS ground state. In the following sections we will develop these algorithms in full detail, but here we can already explain a simple variational algorithm for maximizing the overlap of an MPS $|\Psi(A)\rangle$ with a given reference MPS $|\Psi(\tilde{A})\rangle$. Typically, the latter has a larger bond dimension and the following algorithm is a variational method for truncating an MPS, which is one of the primitive tasks in any MPS toolbox (remember Sec. 2.2 for a non-variational method for truncating a uniform MPS).

The optimization algorithm can be written down as

$$\max_A \frac{|\langle\Psi(\bar{A})|\Psi(\tilde{A})\rangle|^2}{\langle\Psi(\bar{A})|\Psi(A)\rangle}. \tag{92}$$

Because of the orthogonality catastrophe, this objective function might seem ill-defined in the thermodynamic limit, as it evaluates to either zero or one. Nevertheless, the resulting extremal condition (where $|\Psi(A)\rangle$ is assumed to be normalized)

$$\langle\partial_i\Psi(\bar{A})|\left(1-|\Psi(\bar{A})\rangle\langle\Psi(A)|\right)|\Psi(\tilde{A})\rangle = 0 \tag{93}$$

is valid and meaningful in the thermodynamic limit, and states that $|\Psi(\tilde{A})\rangle$, after subtracting the contribution parallel to $|\Psi(A)\rangle$, should be orthogonal to the tangent space of the MPS manifold at the point $|\Psi(A)\rangle$. This condition serves as a variational optimality condition, in the sense that there are no infinitesimal directions on the manifold that improve the overlap in first order. Geometrically, we can write this condition as $P_A|\Psi(\tilde{A})\rangle = 0$, with $P_A$ the projector onto the tangent space (or at least that part of tangent space which is itself orthogonal to $|\Psi(A)\rangle$).

Using the above derivation of the tangent-space projector, we can work out this expression as

$$|\Phi(G;A_L,A_R)\rangle = P_{\{A_L,A_R\}}|\Psi(\tilde{A})\rangle\,, \tag{94}$$

with $G = A'_C - A_L C'$,

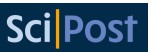 (95)

and

 (96)

Together with the consistency equations for the mixed gauge, an optimal MPS is therefore characterized by the equations

$$A_L C = C A_R = A_C, \tag{97}$$
$$A'_C = A_L C'. \tag{98}$$

It is clear that these equations are satisfied if an MPS is found in the mixed gauge $\{A_L, A_R, C, A_C\}$ such that $A'_C = A_C$ and $C' = C$. A straightforward algorithm for finding this fixed-point solution

---

**Algorithm 3** Variational algorithm for maximizing overlap with MPS $|\Psi(\tilde{A})\rangle$

---

1: **procedure** MAXIMIZEOVERLAP($\{\tilde{A}, \eta\}$)       ▷ tolerance $\eta$
2:     $\{\tilde{A}_L, \tilde{A}_R, \tilde{C}, \tilde{A}_C\} \leftarrow$ MIXEDCANONICAL($\tilde{A}, \eta$)       ▷ Algorithm 2
3:     **repeat**
4:         $(\lambda, L) \leftarrow$ ARNOLDI($X \rightarrow$ map($X, \tilde{A}_L, \bar{A}_L$), $L_0, \delta/10$)       ▷ map in Eq. (99)
5:         $(\sim, R) \leftarrow$ ARNOLDI($X \rightarrow$ map($X, \tilde{A}_R, \bar{A}_R$), $R_0, \delta/10$)       ▷ map in Eq. (100)
6:         $A_C \leftarrow$ COMPUTEAC($L, R, \tilde{A}_C$)       ▷ Eq. (101)
7:         $C \leftarrow$ COMPUTEC($L, R, \tilde{C}$)       ▷ Eq. (102)
8:         $(A_L, A_R) \leftarrow$ MINACC($A_C, C$)       ▷ see Algorithm 5
9:         $\delta \leftarrow \|A_L C - A_C/\lambda\|$       ▷ error function
10:     **until** $\delta < \eta$
11:     **return** $A_L, A_R, C, \lambda$
12: **end procedure**

---

is a simple power method: start from a random MPS $\{A_L^0, A_R^0, C^0, A_C^0\}$, in every iteration (i) compute a new $A_C'$ and $C'$ from the above equations, (ii) distract a new $A_L^i$ and $A_R^i$, and repeat until convergence. Each iteration requires that we can represent the infinite strips in Eqs. (95) and (96) or that we find the fixed points of the maps

$$\tag{99}$$

and

$$\tag{100}$$

Indeed, this allows us to rewrite the above equations for $A_C'$ and $C'$ as

$$\tag{101}$$

and

$$\tag{102}$$

A more involved step requires us to extract a new $A_L^i$ and $A_R^i$ from a $A_C'$ and $C'$. In Sec. 4.4 we show how to efficiently do this. These steps are summarized in Algorithm 3.

# 4 Finding ground states

Now that we have introduced the manifold of matrix product states and the concept of the tangent space, we should explain how to find the point in the manifold that provides the best approximation for the ground state of a given hamiltonian $H$. In these notes, we only consider nearest-neighbour interactions so that the hamiltonian is of the form

$$H = \sum_n h_{n,n+1}, \tag{103}$$

where $h_{n,n+1}$ is a hermitian operator acting non-trivially on the sites $n$ and $n+1$. We refer the reader to Ref. [16] for the generalization to arbitrary long-range hamiltonians.

As in any variational approach, the variational principle serves as a guide for finding ground-state approximations, viz. we want to minimize the expectation value of the energy,

$$\min_A \frac{\langle \Psi(\bar{A}) | H | \Psi(A) \rangle}{\langle \Psi(\bar{A}) | \Psi(A) \rangle}. \tag{104}$$

In the thermodynamic limit the energy diverges with system size, but, since we are working with translation-invariant states only, we should rather minimize the energy density. Also, we will restrict to properly normalized states. Diagrammatically, the minimization problem is recast as

$$\min_A \quad \text{(diagram)} . \tag{105}$$

Traditionally, this minimization problem is not treated directly, but recourse is taken to imaginary-time evolution using the time-evolving block decimation algorithm [6, 17], or to infinite DMRG methods [18]. In this section, we will rather treat this problem in a more straightforward way, in the sense that we will use numerical optimization strategies for minimizing the energy density directly. This approach has the advantage that it is, by construction, optimal in a *global* way, because we never take recourse to local updates of the tensors – we always use routines that are optimal for the MPS wavefunction directly in the thermodynamic limit. As a result, we have a convergence criterion on the energy density for the infinite system.

## 4.1 The gradient

Any optimization problem relies on an efficient evaluation of the gradient, so let us first compute this quantity. The objective function $f$ that we want to minimize is a real function of the complex-valued $A$, or, equivalently, the independent variables $A$ and $\bar{A}$. The gradient $g$ is then obtained by differentiating $f(\bar{A}, A)$ with respect to $\bar{A}$,[9]

$$g = 2 \times \frac{\partial f(\bar{A}, A)}{\partial \bar{A}} \tag{106}$$

$$= 2 \times \frac{\partial_{\bar{A}} \langle \Psi(\bar{A}) | h | \Psi(A) \rangle}{\langle \Psi(\bar{A}) | \Psi(A) \rangle} - 2 \times \frac{\langle \Psi(\bar{A}) | h | \Psi(A) \rangle}{\langle \Psi(\bar{A}) | \Psi(A) \rangle^2} \partial_{\bar{A}} \langle \Psi(\bar{A}) | \Psi(A) \rangle \tag{107}$$

$$= 2 \times \frac{\partial_{\bar{A}} \langle \Psi(\bar{A}) | h | \Psi(A) \rangle - e \partial_{\bar{A}} \langle \Psi(\bar{A}) | \Psi(A) \rangle}{\langle \Psi(\bar{A}) | \Psi(A) \rangle}, \tag{108}$$

$$\tag{109}$$

---

[9]Numerical optimization schemes are typically developed for functions over real parameters. In order to translate these algorithms to complex parameters, we take $x = x_r + i x_i$, and take the gradient $g = g_r + i g_i$ with $g_r = \partial_{x_r} f$ and $g_i = \partial_{x_i} f$, which is equal to $g = 2 \partial_{\bar{x}} f(x, \bar{x})$.

where we have clearly indicated $A$ and $\bar{A}$ as independent variables and $e$ is the current energy density given by

$$e = \frac{\langle \Psi(\bar{A})| h |\Psi(A)\rangle}{\langle \Psi(\bar{A})|\Psi(A)\rangle}. \tag{110}$$

In the implementation we will always make sure the MPS is properly normalized, such that the numerators drop out. Furthermore, we subtract from every term in the hamiltonian its current expectation value

$$h \to h - \langle \Psi(\bar{A})| h |\Psi(A)\rangle, \tag{111}$$

so that the gradient takes on the simple form

$$g = 2 \times \partial_{\bar{A}} \langle \Psi(\bar{A})| h |\Psi(A)\rangle. \tag{112}$$

The gradient is obtained by differentating the expression

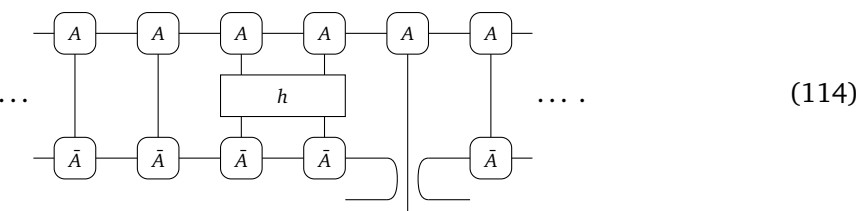

$$\tag{113}$$

with respect to $\bar{A}$. It is given by a sum over all sites, where in every term we differentiate with one tensor $\bar{A}$ in the bra layer. Differentiating with respect to one $\bar{A}$ tensor amounts to leaving out that tensor, and interpreting the open legs as outgoing ones, i.e. each term looks like

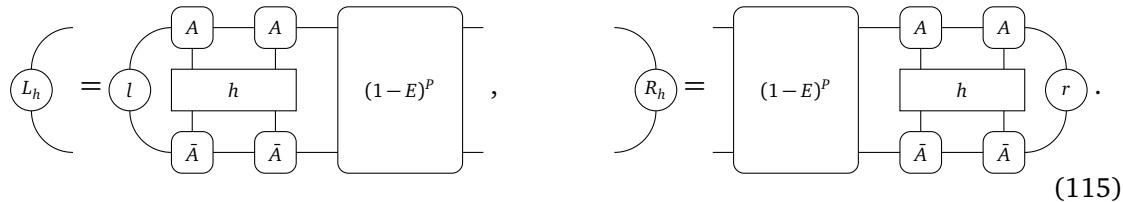

$$\tag{114}$$

For summing the infinite number of terms, we will use the same techniques as we did for evaluating the structure factor [Sec. 2.5]. Instead of varying the open spot in the diagram, we will vary the location of the hamiltonian operator $h$. Then, we first treat all terms where $h$ is either completely to the left or to the right of the open spot, by defining the partial contractions

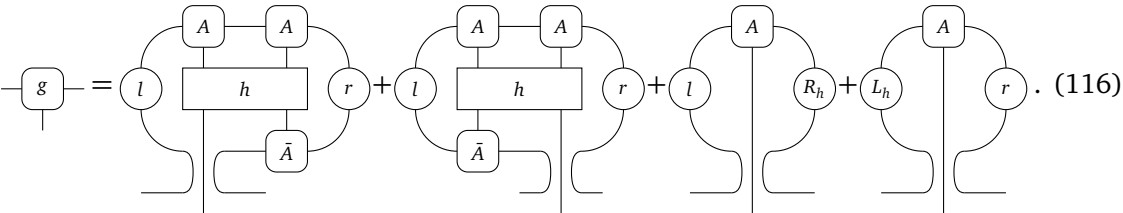

$$\tag{115}$$

As we have seen, taking these pseudo-inverses is equivalent to summing the infinite number of terms. Note that, because the expectation value of $h$ is by definition subtracted in the gradient of the normalized energy expectation value, we indeed only need to take the connected part into account and no diverging $\delta$-contribution is present. The partial contractions above are combined with the two contributions where $h$ acts on the open spot, so that we have the final expression for the gradient

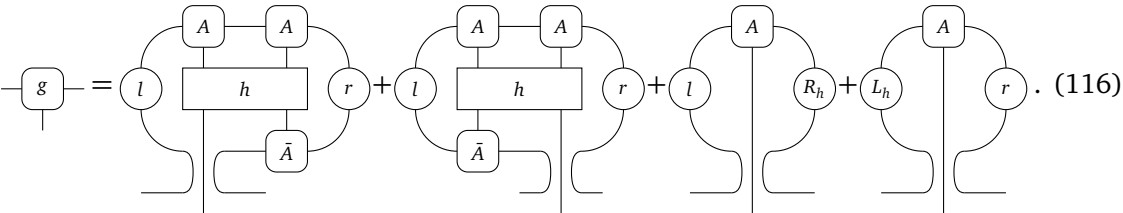

$$\tag{116}$$

As such, the gradient is an object that lives in the space of MPS tensors. However, we can further exploit the manifold structure of MPS by interpreting the gradient as a tangent vector – the gradient is, indeed, supposed to indicate a direction on the manifold along which we can lower the energy. In order to consistently interpret the gradient as a tangent vector, we need some additional steps. First, let us note the meaning of the gradient as defined above in terms of the first-order approximation of a change in the tensor $A + \epsilon B$

$$\langle \Psi(\bar{A})| h |\Psi(A)\rangle \rightarrow \langle \Psi(\bar{A})| h |\Psi(A)\rangle + \epsilon \boldsymbol{g}^\dagger \boldsymbol{B} + \mathcal{O}(\epsilon^2), \tag{117}$$

where vectorized versions of tensors are denoted in bold.

Now we realize that $g$ is a vector in the space of tensors, not a state in the full Hilbert space. So how do we lift the notion of the gradient to the level of a state? Note that an infinitesimal change in the tensor $A + \epsilon B$ corresponds to a tangent vector

$$|\Psi(A + \epsilon B)\rangle \rightarrow |\Psi(A)\rangle + \epsilon |\Phi(B;A)\rangle + \mathcal{O}(\epsilon^2), \tag{118}$$

so that we would like to write the first-order change in the energy through an overlap between this tangent vector and a 'gradient vector' $|\Phi(G;A)\rangle$

$$\langle \Psi(\bar{A})| h |\Psi(A)\rangle \rightarrow \langle \Psi(\bar{A})| h |\Psi(A)\rangle + \epsilon \langle \Phi(\bar{G};\bar{A})|\Phi(B;A)\rangle + \mathcal{O}(\epsilon^2). \tag{119}$$

We know, however, that building $|\Phi(G;A)\rangle$ using the tensor $g$ is not correct, because the overlap $\langle \Phi(G;A)|\Phi(B;A)\rangle \neq \boldsymbol{G}^\dagger \boldsymbol{B}$. Instead, we will have to determine the *tangent-space gradient* by its reduced parametrization, where, as usual

$$X_G = \begin{array}{c} \\ \end{array} \tag{120}$$

so that the tangent-space gradient is given by the usual expression for a tangent vector,

$$|\Phi(G;A)\rangle = \sum_n \dots \boxed{A} \boxed{A} \boxed{G} \boxed{A} \boxed{A} \dots, \tag{121}$$

with the tensor $G$ given by

$$\boxed{G} = \boxed{l^{-\frac{1}{2}}} \boxed{V_L} \boxed{X_G} \boxed{r^{-\frac{1}{2}}}. \tag{122}$$

The difference between these two notions of a gradient can be elucidated by looking at the variational manifold from the perspective of differential geometry. Whereas the gradient is a covariant vector living in the cotangent bundle, we can use the (non-trivial) metric of the MPS manifold to define a corresponding (contravariant) vector living in the tangent bundle [11, 15]. The latter is what we have defined as the tangent-space gradient.

Note that we can also derive the expression for the tangent-space gradient from the tangent-space projector in Eq. (81). Indeed, we can readily check that

$$|\Psi(G;A)\rangle = \mathcal{P}_A\Big(H - \langle \Psi(\bar{A})| H |\Psi(A)\rangle\Big) |\Psi(A)\rangle. \tag{123}$$

This expression for the gradient will be the starting point for the vumps algorithm in Sec. 4.4.

## 4.2 Optimizing the tensors

Using these expressions for the different types of gradient, we can easily implement a gradient-search method for minimizing the energy expectation value.

The first obvious option is a steepest-descent method, where in every iteration the tensor $A$ is updated in the direction of the parameter-space gradient:

$$A_{i+1} = A_i - \alpha g. \tag{124}$$

The size of $\alpha$ is determined by doing a line search: we find a value for which the energy density has decreased. In principle, we could try to find the optimal value of $\alpha$, for which we can no longer decrease the energy by taking the direction $-g$ in parameter space. In practice, we will be satisfied with an approximate value of $\alpha$, for which certain conditions [19] are fulfilled. Other optimization schemes based on an evaluation of the gradient, such as conjugate-gradient or quasi-Newton methods, are more efficient. Even more efficient would be an algorithm that requires an evaluation of the Hessian, which in principle we can also do with the techniques above.

As another road to more efficient optimization schemes we could take the tangent-space gradient a bit more seriously. A first option amounts to computing the tangent-space gradient, and then update the $A$ tensor by simply adding them in parameter space, i.e. do a line search of the form

$$A_{i+1} = A_i - \alpha G. \tag{125}$$

This scheme can, again, be improved by implementing conjugate-gradient or quasi-Newton optimization methods. It is expected that the use of the tangent-space gradient yields a more efficient energy optimization, because it takes the structure of the manifold (embedded in Hilbert space) into account. Taking a step further in this approach, instead of just adding $G$ in parameter space, we would like to do a line search along geodetic paths through the manifold, which would involve integrating the geodesic equation. It remains an open question, however, whether this could lead to more efficient optimization schemes.

Crucially, this way of variationally optimizing an MPS has a clear convergence criterion: we say that we have reached a – possibly local – optimum if the norm of the gradient is sufficiently small.

## 4.3 The energy variance

In any variational approach, finding an optimal set of parameters does not guarantee that the state provides a good approximation to the true ground state of the hamiltonian. We do have access, however, to an unbiased measure of how well the MPS approximates *any* eigenstate of the system, called the variance. It is defined by

$$v = \langle \Psi(\bar{A}) | H^2 | \Psi(A) \rangle, \tag{126}$$

where we have subtracted the ground-state energy density from the local nearest-neighbour term in the hamiltonian, i.e. $h_{n,n+1} \rightarrow h_{n,n+1} - \langle \Psi(\bar{A}) | h_{n,n+1} | \Psi(A) \rangle$. This quantity can be readily interpreted as a zero-momentum structure factor, so we can apply the formulas from Sec. 2.5. The formulas are a bit more complicated, since we have a two-site operator. In the end, the

variance is given by

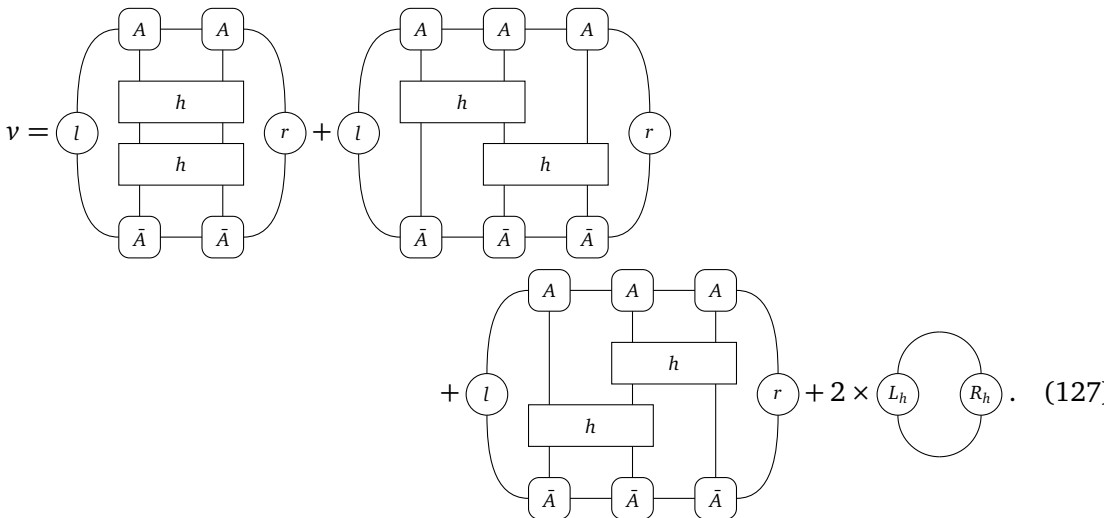

$$v = \left( \ \ \right) + \left( \ \ \right) + \left( \ \ \right) + 2 \times \left( \ \ \right). \tag{127}$$

This quantity is supposed to be a small number obtained by a cancellation of large terms, and therefore this expression can give rise to errors. In Ref. [20] a slightly different error was proposed that avoids this numerical inaccuracy.

## 4.4 The vumps algorithm

We can now combine the notion of a tangent-space gradient with the use of the mixed gauge in order to develop an efficient variational ground-state optimization algorithm known as vumps[10] [16]. We have seen above that a variational optimum is characterized by the condition that the gradient be zero. The error measure is therefore given by

$$\epsilon = \left( \langle \Phi(\bar{G}; \bar{A}_L; \bar{A}_R) | \Phi(G; A_L, A_R) \rangle \right)^{1/2} = \left( G^\dagger G \right)^{1/2}. \tag{128}$$

In the mixed canonical form this vector is given by using the tangent-space projector

$$|\Phi(G; A_L, A_R)\rangle = \mathcal{P}_{\{A_L, A_R\}}(H - E) |\Psi(A_L, A_R)\rangle, \tag{129}$$

yielding

$$G = A'_C - A_L C' \qquad \text{or} \qquad G = A'_C - C' A_R, \tag{130}$$

where $A'_C$ and $C'$ are given by

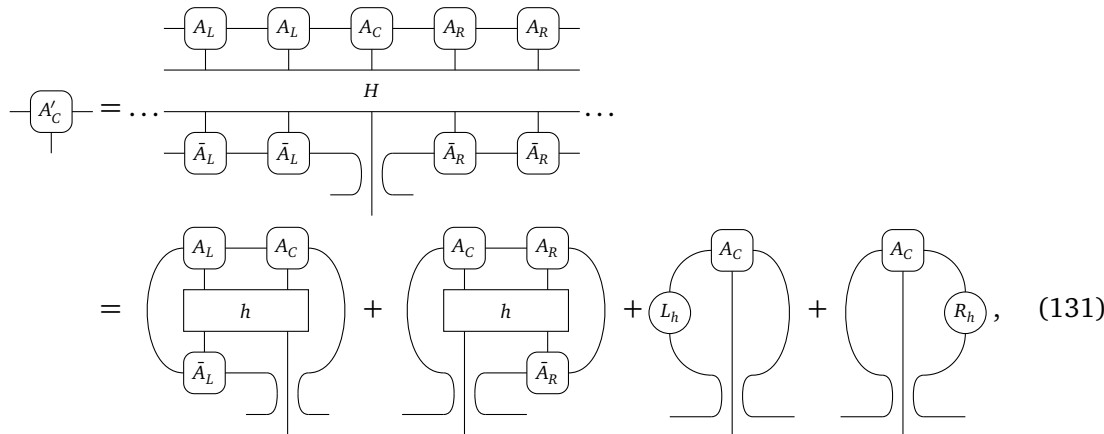

$$\tag{131}$$

---

[10]The name vumps is the acronym for variational uniform matrix product states.

and

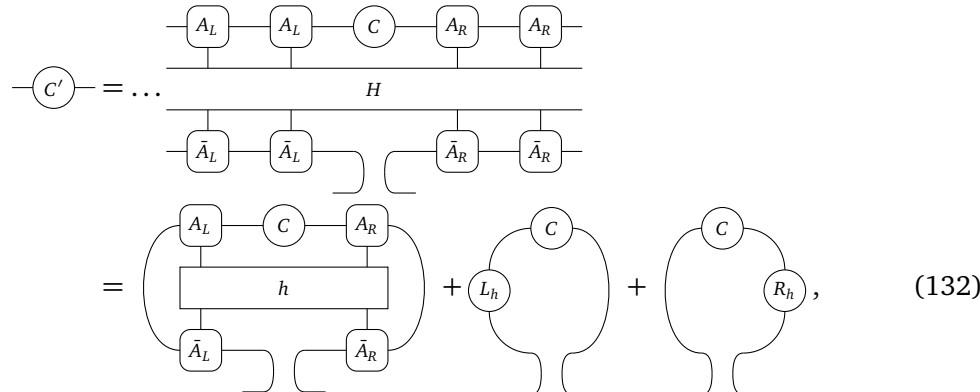

$$\tag{132}$$

with

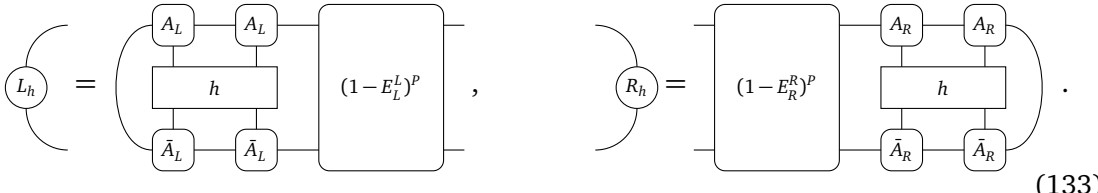

$$\tag{133}$$

These equations define effective hamiltonians $H_{A_C}(\cdot)$ and $H_C(\cdot)$ such that

$$A'_C = H_{A_C}\left(A_C\right) \tag{134}$$

$$C' = H_C\left(C\right). \tag{135}$$

Since the gradient should be zero in the variational optimum, we can characterize this point as $A'_C = A_L C' = C' A_R$. In turn this implies that in the optimum the MPS should obey the following set of equations

$$H_{A_C}\left(A_C\right) \propto A_C \tag{136}$$

$$H_C\left(C\right) \propto C \tag{137}$$

$$A_C = A_L C = C A_R. \tag{138}$$

The vumps algorithm consists now of an iterative method for finding an $\{A_L, A_R, A_C, C\}$ that satisfies these equations simultaneously. In every step of the algorithm we first solve the two eigenvalue equations, yielding two new tensors $\tilde{A}_C$ and $\tilde{C}$. An obvious choice for the global updates now seems to be given by $\tilde{A}_L = \tilde{A}_C \tilde{C}^{-1}$ and $\tilde{A}_R = \tilde{C}^{-1}\tilde{A}_C$, but then we face additional problems. Away from the converged solution, the resulting $\tilde{A}_L$ and $\tilde{A}_R$ will in general not be isometric. Furthermore, we would like to avoid taking (possibly ill-conditioned) inverses of $\tilde{C}$ altogether, as this ruins the advantage of working with the center site. As an alternative strategy, we determine global updates $\tilde{A}_L$ and $\tilde{A}_R$ as the left and right isometric tensors that minimize

$$\epsilon_L = \min\|\tilde{A}_C - \tilde{A}_L \tilde{C}\|_2 \tag{139}$$

$$\epsilon_R = \min\|\tilde{A}_C - \tilde{C}\tilde{A}_R\|_2. \tag{140}$$

In exact arithmetic, the solution of these minimization problems is known, namely $\tilde{A}_L$ will be the isometry in the polar decomposition of $\tilde{A}_C \tilde{C}^{\dagger}$. Computing the singular-value decompositions yields

$$\tilde{A}_C \tilde{C}^{\dagger} = U_l \Sigma_l V_l^{\dagger} \rightarrow \tilde{A}_L = U_l V_l^{\dagger} \tag{141}$$

$$\tilde{C}^{\dagger}\tilde{A}_C = U_r \Sigma_r V_r^{\dagger} \rightarrow \tilde{A}_R = U_r V_r^{\dagger}. \tag{142}$$

---

**Algorithm 4** Vumps algorithm for nearest-neighbour hamiltonian $h$

---

1: **procedure** VUMPS($h, A, \eta$)            ▷ Initial guess $A$ and a tolerance $\eta$
2:      $\{A_L, A_R, C\} \leftarrow$ MIXEDCANONICAL($A, \eta$)          ▷ Algorithm 2
3:      **repeat**
4:          $e \leftarrow$ EVALUATEENERGY($A_L, A_R, A_C, H$)
5:          $\tilde{h} \leftarrow h - e\mathbb{1}$            ▷ subtract energy expectation value
6:          $L_h \leftarrow$ SUMLEFT($A_L, \tilde{h}, \delta/10$)          ▷ Eq. (133)
7:          $R_h \leftarrow$ SUMRIGHT($A_R, \tilde{h}, \delta/10$)          ▷ Eq. (133)
8:          $(\sim, A'_C) \leftarrow$ ARNOLDI($X \rightarrow H_{A_C}(X), A_C$,'sr',$\delta/10$)    ▷ the map $H_{A_c}$ in Eq. (131)
9:          $(\sim, C') \leftarrow$ ARNOLDI($X \rightarrow H_C(X), C$,'sr',$\delta/10$)      ▷ the map $H_C$ in Eq. (132)
10:        $(A_L, A_R, C, A_C) \leftarrow$ MINACC($A'_C, C'$)          ▷ Algorithm 5
11:        $\delta \leftarrow \|H_{A_C}(A_C) - A_L H_C(C)\|$         ▷ norm of the gradient [Eq. (130)]
12:      **until** $\delta < \eta$
13:      **return** $A_L, A_R, C, e$
14: **end procedure**

---

---

**Algorithm 5** Find $\{\tilde{A}_L, \tilde{A}_R\}$ from a given $\tilde{A}_C$ and $\tilde{C}$

---

1: **procedure** MINACC($A'_C, C'$)
2:      $(U^l_{A_C}, P^l_{A_C}) \leftarrow$ POLARLEFT($\tilde{A}_C$)
3:      $(U^l_C, P^l_C) \leftarrow$ POLARLEFT($\tilde{C}$)
4:      $\tilde{A}_L \leftarrow U^l_{A_C}(U^l_C)^\dagger$
5:      $(U^r_{A_C}, P^r_{A_C}) \leftarrow$ POLARRIGHT($\tilde{A}_C$)
6:      $(U^r_C, P^r_C) \leftarrow$ POLARRIGHT($\tilde{C}$)
7:      $\tilde{A}_r \leftarrow (U^r_C)^\dagger U^r_{A_C}$
8:      **return** $\tilde{A}_L, \tilde{A}_R$
9: **end procedure**

---

Notice that close to (or at) an exact solution $A^s_C = A^s_L C = C A^s_R$, the singular values contained in $\Sigma_{l/r}$ are the square of the singular values of $C$, and might well fall below machine precision. Consequently, in finite precision arithmetic, corresponding singular vectors will not be accurately computed. An alternative that has proven to be robust and still close to optimal is given by directly using the following left and right polar decompositions

$$\tilde{A}_C = U^l_{A_c} P^l_{A_C}, \quad \tilde{C} = U^l_C P^l_C, \tag{143}$$

$$\tilde{A}^r_C = P^r_{A_C} U^r_{A_C}, \quad \tilde{C} = P^r_C U^r_C, \tag{144}$$

to obtain

$$\tilde{A}_L = U^l_{A_C}(U^l_C)^\dagger, \qquad \tilde{A}_R = (U^r_C)^\dagger U^r_{A_C}. \tag{145}$$

This concludes one step of the iteration procedure, yielding a new set of tensors $\{\tilde{A}_L, \tilde{A}_R, \tilde{A}_C, \tilde{C}\}$. This process is repeated until the norm of the gradient $\epsilon$ is sufficiently small. Another error measure is the value of either $\epsilon_L$ or $\epsilon_R$, which can be proven to be proportional to $\epsilon$ close to convergence.

# 5 The time-dependent variational principle

Although DMRG was originally developed for finding the ground state, and, possibly, the first low-lying states of a given hamiltonian, the scope of DMRG simulations has since been extended to dynamical properties as well. One of the many new applications has been the simulation of time evolution, where the MPS formalism has been of crucial value for coming up with algorithms such as the time-evolving block decimation [21–23]. In this section, we discuss another algorithm for simulating time evolution in the thermodynamic limit, based on the time-dependent variational principle (TDVP) [24–26]. This approach has the advantage of being variational – in a sense that we will explain below – and, therefore, more controlled, and leading to a set of a symplectic differential equations. The MPS version of the TDVP [7,15,27] has been applied to spin chains [28], gauge theories [29,30], and spin systems with long-range interactions [27,31–34].

In these notes we are exclusively interested in the case where the initial state is a uniform MPS and the time evolution is governed by a translation-invariant hamiltonian (global quench). However, the framework can be extended to so-called local quenches as well.

## 5.1 Time evolution on a manifold

The algorithm relies on the manifold interpretation of uniform matrix product states, and, in particular, the concept of a tangent space. We start from the Schrödinger equation,

$$i\frac{\partial}{\partial t}|\Psi(A)\rangle = H|\Psi(A)\rangle, \tag{146}$$

which dictates how a quantum state evolves in time. The problem with this equation is the fact that, for a generic hamiltonian, an initial MPS $|\Psi(A)\rangle$ is immediately taken out of the MPS manifold. Nonetheless, we would like to find a path inside the manifold $|\Psi(A(t))\rangle$, which approximates the time evolution in an optimal way. The time-derivative of this time-evolved MPS is a tangent vector,

$$i\frac{\partial}{\partial t}|\Psi(A(t))\rangle = |\Phi(\dot{A};A)\rangle, \tag{147}$$

but, again, the right-hand side is not. Indeed, the vector $H|\Psi(A(t))\rangle$ points out of the manifold, so that an exact integration of the Schrödinger equation is out of the question. Finding $\dot{A}$ for which the corresponding tangent vector provides the best approximation to $H|\Psi(A(t))\rangle$ amounts to the minimization problem

$$\dot{A} = \arg\min_{B}\left\| H|\Psi(A)\rangle - |\Phi(B;A)\rangle\right\|_2^2. \tag{148}$$

Note that the solution of this minimization problem is equivalent to projecting the time evolution orthogonally onto the tangent space,

$$i\frac{\partial}{\partial t}|\Psi(A(t))\rangle = P_{A(t)}H|\Psi(A(t))\rangle. \tag{149}$$

This projection transforms the linear Schrödinger equation into a highly non-linear differential equation on a manifold, and is illustrated graphically in Fig. 2.

We can work out the formal properties of the TDVP in a bit more detail. The TDVP equation can be rewritten as

$$i\frac{\partial}{\partial t}|\Psi(A)\rangle = \left(\partial_{A_i}|\Psi(A)\rangle\right)(G^{-1})^{ij}\left(\partial_{\bar{A}_i}\langle\Psi(\bar{A})|\right)H|\Psi(A)\rangle \tag{150}$$

$$= \partial_{A_i}|\Psi(A)\rangle(G^{-1})^{ij}\partial_{\bar{A}_j}h(A,\bar{A}), \tag{151}$$

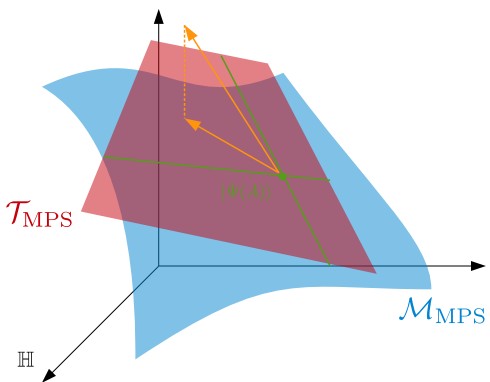

Figure 2: The time-dependent variational principle.

where we have defined

$$h(A, \bar{A}) = \langle \Psi(\bar{A})| H |\Psi(A)\rangle. \tag{152}$$

If we note that

$$\frac{\partial}{\partial t} |\Psi(A)\rangle = \partial_t A_i \partial_{A_i} |\Psi(A)\rangle, \tag{153}$$

we obtain the TDVP in the explicit form

$$i\partial_t A_i = [G^{-1}(A, \bar{A})]^{ij} \partial_{\bar{A}_j} h(A, \bar{A}), \tag{154}$$

where the non-linearity follows from the non-linear dependence of both $h$ and $G$ on $A$ (and $\bar{A}$). For arbitrary functions $f$ and $g$ of $A$ and $\bar{A}$, we can now introduce a Poisson bracket (henceforth suppressing the explicit $A$ dependence) as

$$\{f, g\} = -(\partial_{A_i} f)(G^{-1})^{ij}(\partial_{\bar{A}_j} g) + (\partial_{A_i} g)(G^{-1})^{ij}(\partial_{\bar{A}_j} f), \tag{155}$$

which is clearly anti-symmetric and bilinear, and can be shown to obey the Jacobi conditions. Here, $f$ (and $g$) would typically represent expectation values $f(A, \bar{A}) = \langle \Psi(\bar{A})|F|\Psi(A)\rangle$. It follows that the equations of motion for such expectation values are given by

$$\begin{aligned} \partial_t f &= (\partial_{A_i} f)\partial_t A_i + (\partial_{\bar{A}_i} f)\partial_t \bar{A}_i \\ &= i\{f, h\}. \end{aligned} \tag{156}$$

This equation, in the end, shows that the TDVP for the MPS manifold gives rise to an effective classical hamiltonian system with corresponding Poisson bracket. These equations are, however, highly non-linear because of the non-linearity of the tangent-space projector. Colloquially, we can state that the TDVP approach maps the linear quantum dynamics in an exponentially large Hilbert space to a set of non-linear semi-classical equations of motion for the expectation values, in terms of a smaller number of effective degrees of freedom (the variational parameters).

This observation has important consequences with respect to conservation laws. Indeed, it is trival to see that

$$\partial_t h = i\{h, h\} = 0, \tag{157}$$

so that the energy expectation value is exactly conserved under time evolution with the TDVP. Also, other conserved quantities are exactly conserved, under the condition that they commute with the tangent-space projector. Indeed, if we have that for the generator of the symmetry $K$ (i.e. $[K, H] = 0$) the condition

$$P_A K |\Psi(A)\rangle = K |\Psi(A)\rangle \tag{158}$$

is obeyed, one can show that

$$\partial_t k = i\{h, k\} = 0, \tag{159}$$

with $k = \langle \Psi(\bar{A})| K |\Psi(A)\rangle$. For the MPS manifold, this is the case for all symmetries which act as a tensor product of one-site gates, i.e. when the generator is a sum of one-site operators.

## 5.2   TDVP in the uniform gauge

Let us now work out the TDVP equation

$$|\Phi(\dot{A}; A(t))\rangle = -i\mathcal{P}_{A(t)} H |\Psi(A(t))\rangle \tag{160}$$

in the uniform gauge. In Sec. 3 we have written down the tangent-space projector in the uniform gauge. Both the original Schrödinger equation and the TDVP evolution are norm preserving (for real time evolution), but introduce a global phase proportional to the total energy, which is divergent with the system size. By imposing that $\langle \Psi(A)|\Phi(B; A)\rangle = 0$, this phase is eliminated and norm preservation is explicitly enforced (now even in the case of imaginary time evolution as introduced below). By now, we know very well that an effective parametrization of the tangent vector in terms of the matrix $X$ can be introduced which automatically enforces orthogonality.

In order to implement this projector, we compute the matrix element $\langle \Phi(B; A)| H |\Psi(A)\rangle$ for general $B$. Again, we have two infinite sums, but one is eliminated because of translation invariance and gives rise to a $2\pi\delta(0)$ factor. Then we need all terms where the hamiltonian term acts fully to the left and to the right of the $B$ tensor, but this can again be resummed efficiently by introducing pseudo-inverses of the transfer matrix in the following partial contractions:

$$\tag{161}$$

and

$$\tag{162}$$

In addition, we also have the terms where the hamiltonian operator acts directly on the site where the $B$ tensor is located. Putting all contributions together, we obtain

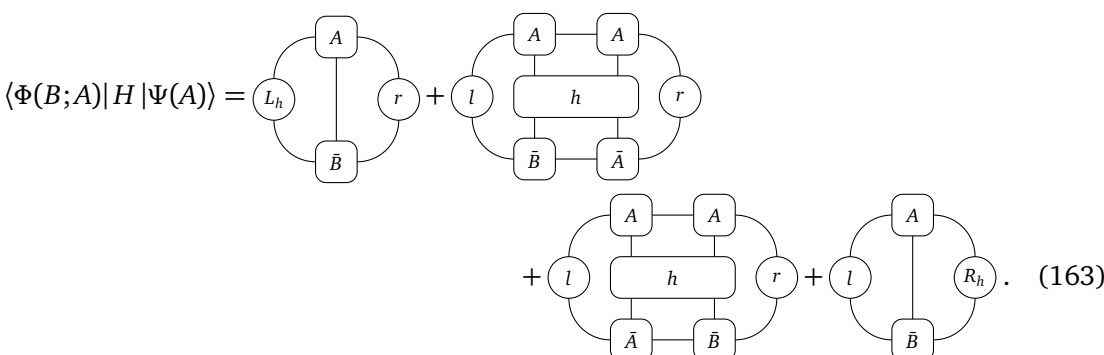

$$\tag{163}$$

**Algorithm 6** Compute $\dot{A}$ according to TDVP from a given $A$ and $h$

---

1: **procedure** TDVP($A, h, \eta$)                                                          ▷ Tolerance $\eta$
2:     Find $l$ and $r$ matrices
3:     $L_h \leftarrow$ SUMLEFT($A,l,h,\eta$)                                          ▷ Eq. (161)
4:     $R_h \leftarrow$ SUMRIGHT($A,r,h,\eta$)                                      ▷ Eq. (162)
5:     $F \leftarrow$ COMPUTEF($A,l,r,h,L_h,R_h$)                          ▷ Eq. (164)
6:     $V_L \leftarrow$ NULLSPACE($A,l^{1/2}$)                                  ▷ Eq. (70)
7:     $\dot{A} \leftarrow$ TANGENTPROJECTOR($V_L,l,r$)                  ▷ Eq. (165)
8:     **return** $\dot{A}$
9: **end procedure**

---

The tangent-space projected time evolution is then obtained by taking the tensor $F$,

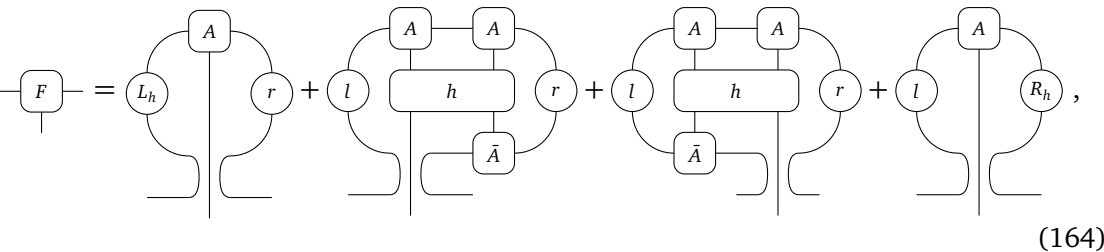

$$\tag{164}$$

and compute the time evolution of the MPS tensor according to the TDVP as

$$\tag{165}$$

This procedure for computing the time derivative of the MPS tensor $A$ according to the TDVP is summarized in Algorithm 6. The simplest option for integrating this differential equation for $A(t)$ consists of a simple Euler scheme, where $A(t + \delta t) = A(t) + \delta t \dot{A}(t)$, but a numerical integrator that does not destroy the symplectic properties of the TDVP equation is often preferred.

At this point, the attentive reader might already have noticed that these formulas are very similar to the ones that we obtained for the gradient of the energy that appears in a ground-state optimization algorithm – the tangent-space gradient is the same as the right hand side of the TDVP equation up to an imaginary factor. The connection is laid bare by noting that another road to a ground-state optimization algorithm is found by implementing an imaginary-time evolution ($t \rightarrow -i\tau$) on a random starting point $|\Psi(A_0)\rangle$, confined within the MPS manifold. Indeed, in the full Hilbert space, imaginary time evolution results in a projection onto the ground state in the infinite-time limit

$$\lim_{\tau \rightarrow \infty} \frac{e^{-H\tau} |\Psi\rangle}{\left\| e^{-H\tau} |\Psi\rangle \right\|} = |\Psi_0\rangle \tag{166}$$

for almost any initial state $|\Psi\rangle$. When using the TDVP to restrict imaginary time evolution to the manifold of MPS and integrating the resulting equations using a simple Euler scheme,

$$A(\tau + d\tau) = A(\tau) - d\tau \dot{A}(\tau), \tag{167}$$

we are effectively performing a steepest-descent optimization with the tangent-space gradient, where in every iteration the line search is replaced by taking a fixed step size $\alpha = d\tau$.

## 5.3 TDVP in the mixed gauge

We can now use this tangent-space projector to write down the TDVP equation for an MPS in the mixed canonical form. We have explained that the optimal way for implementing real-time evolution within the MPS manifold is by projecting the exact time derivative onto the tangent space at every point, i.e.

$$\frac{\partial}{\partial t}|\Psi(A_L, A_C, A_R)\rangle = -iP_A H|\Psi(A_L, A_C, A_R)\rangle. \tag{168}$$

In the mixed canonical gauge, the tangent space projector decomposes into two different parts corresponding to the two different types of terms in Eq. (91). For each of the terms individually, the corresponding differential equation can be integrated straightforwardly. Let us take the first part, for which we first define the partial contractions

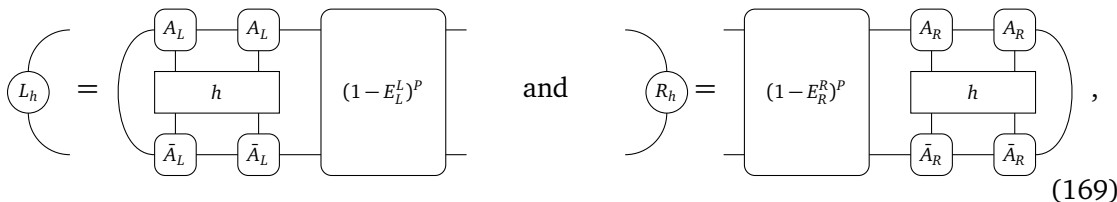

$$\tag{169}$$

which capture the contributions where the hamiltonian is completely to the left and to right of the open spot in the projector. Again, we have used pseudo-inverses for resumming the infinite number of terms. These are combined into

$$\tag{170}$$

such that

$$P^1_{|\Psi(A)\rangle}H|\Psi(A_L, A_C, A_R)\rangle = \sum_n \ldots \boxed{A_L} \boxed{A_L} \boxed{G_1} \boxed{A_R} \boxed{A_R} . \tag{171}$$

In order to obtain the second part, we need to contract the above $G_1$ tensor another time with $\bar{A}_L$,

$$\tag{172}$$

in order to arrive at

$$P^2_{|\Psi(A)\rangle}H|\Psi(A_L, A_C, A_R)\rangle = \sum_n \ldots \boxed{A_L} \boxed{A_L} \boxed{A_L} \boxed{G_2} \boxed{A_R} \boxed{A_R} . \tag{173}$$

The two parts of the differential equation can be solved separately, but in different representations of the MPS. Indeed, if we write the MPS in the mixed canonical form *with* center site, the first equation is simply

$$\dot{A}_C = -iG_1(A_C), \tag{174}$$

where $G_1(A)$ is interpreted as a linear operator working on $A_C$ according to Eq. (170); the solution is simply $A_C(t) = \mathrm{e}^{-iG_1 t} A_C(0)$. Alternatively, if the MPS is written in the mixed canonical form *without* center site, the second equation is

$$\dot{C} = +iG_2(C), \tag{175}$$

where the sign difference in the right hand side comes from having a minus sign in the second part of the tangent-space projector. Again, $G_2(A)$ is seen as a linear operator acting on $C$ according to Eqs. (170) and (172) with corresponding solution given by $C(t) = \mathrm{e}^{+iG_2 t} C(0)$. These exponentials can be evaluated efficiently by using Lanczos-based iterative methods.

**Integrating the TDVP equations**

The meaning of the TDVP equations is slightly different in this mixed canonical form, and a correct interpretation starts from considering the case of a finite lattice. There the meaning is clear: every site in the lattice has a different MPS tensor attached to it, and performing one time step amounts to doing one sweep through the chain. For every step in the sweep at site $n$, we

- start from a mixed canonical form with center site tensor $\hat{A}_C(n)$, all tensors $\tilde{A}_L(n-2)$, $\tilde{A}_L(n-1)$, etc, have already been updated, while tensors $A_R(n+1)$ and $A_R(n+2)$, etc., are still waiting for their update,

- we update the center-site tensor as $\tilde{A}_C(n) = \mathrm{e}^{-iG_1(n)\delta t}\hat{A}_C(n)$,

- we do a QR decomposition, $\tilde{A}_C(n) = \tilde{A}_L(n)\tilde{C}(n)$,

- we update the center matrix as $\hat{C}(n) = \mathrm{e}^{+iG_2(n)\delta t}\tilde{C}(n)$,

- we absorb this center matrix into the tensor on the right to define a new center-site tensor $\hat{A}_C(n+1) = \hat{C}(n)A_R(n+1)$.

The version for the infinite system can be derived by starting this procedure at an arbitrary site $n$ in the chain – say $n \to -\infty$, so that we will never notice the effect of this abrupt operation in the bulk of the system – and start applying exactly the same procedure until it converges. In this context, convergence at site $n$ would mean that the center-site that we obtain for the next site, $\hat{A}_C(n+1)$, would give us the same as the one we started from, $\hat{A}_C(n) = \hat{A}_C(n+1)$. Our real interest, however, goes out to the converged value of $\tilde{A}_L$, because this allows us to obtain $\tilde{A}_R, \tilde{A}_C$ and $\tilde{C}$. Only after we have obtained convergence in this sense, we have concluded the integration of one time step $\delta t$.

This procedure, where each time step requires solving a consistency relation, is very costly in practice, and therefore we propose a simpler integration procedure. The idea is that consistency of the above scheme requires that, after one iteration of the scheme, we find the same $C$ matrix as the one we started from. This allows to turns things around, where we assume that we retrieve the same $C$ matrix after an iteration of the above scheme, evolve it with the time-reversed operator to find $\tilde{C} = \mathrm{e}^{-iG_2\delta t}C$. Then, we can find an updated $\tilde{A}_L$ and $\tilde{A}_R$ from $\tilde{A}_C$ and $\tilde{C}$. This scheme for time evolving an MPS $\{A_L, A_R, A_C, C\}$ to $\{\tilde{A}_L, \tilde{A}_R, \tilde{C}, \tilde{A}_C\}$ after a time step $\delta t$ then boils down to

- time evolve the center-site tensor *forward* in time, $\tilde{A}_C = \mathrm{e}^{-iG_1\delta t}A_C$,

- time evolve the center matrix *backward* in time, $\tilde{C} = \mathrm{e}^{-iG_2\delta t}C$,

- find an updated $\tilde{A}_L$ and $\tilde{A}_R$ from $\tilde{A}_C$ and $\tilde{C}$.

The last step can be done using Algorithm 5.

Note that the imaginary-time version of this last scheme is very close to the vumps algorithm [Sec. 4.4]. Indeed, if we implement the above scheme with imaginary-time steps where the size of the step is taken to infinity, the evolution operators reduce to projectors on the leading eigenvectors, and therefore we recover the eigenvalue equations that are used in vumps. It remains a matter of further investigation to assess whether we could gain efficiency in ground-state optimization by working with finite imaginary-time steps.

# 6 Elementary excitations

We have seen that working directly in the thermodynamic limit has a number of conceptual and numerical advantages over finite-size algorithms, but the real power of the formalism is shown when we want to describe elementary excitations. These show up in dynamical correlation functions [see Sec. 6.4] that can be directly measured in e.g. neutron-scattering experiments. Typically, these experiments allow to probe the spectrum within a certain momentum sector, giving rise to excitation spectra that look like the one in Fig. 3. The isolated branches in such a spectrum – these will correspond to $\delta$ peaks in the spectral functions, and are seen as very strong resonances in experimental measurements – can be interpreted as quasiparticles, which can be thought of as local perturbations on the ground state, in a plane-wave superposition with well-defined momentum [35]. The rest of the low-energy spectrum can be reconstructed by summing up the energies and momenta of the isolated quasiparticles – in the thermodynamic limit these quasiparticles will never see each other, so these energies and momenta can be simply superposed. This picture implies that all the low-energy properties should in the end be brought back to the properties of these quasiparticles!

Crucially, this approach differs from standard approaches for describing quasiparticles in interacting quantum systems. Indeed, typically a quasiparticle is thought of as being defined by starting from a non-interacting limit, and acquires a finite lifetime as interactions are turned on – think of Fermi liquid theory as the best example of this perturbative approach. In contrast, our approach will be variational, as we will approximate exact eigenstates with a variational ansatz. This means that our quasiparticles have an infinite lifetime, and correspond to stationary eigenstates of the fully interacting hamiltonian.

This quasiparticle approach is only guaranteed to be applicable to gapped systems, where isolated excitation branches are expected to arise. Still, the approach continues to work for critical systems as well, where it leads to excellent estimates for gapless dispersion relations. The variational excited states that are obtained are no longer expected to the elementary excitations in these critical systems – it is not even clear if these can be unambiguously defined for a generic interacting (non-integrable) spin chain – but rather correspond to a superposition of many gapless excitations around the Fermi point, leading to a particle with a localized nature in a momentum superposition.

## 6.1 The quasiparticle ansatz

It is in fact very natural to construct quasiparticle excitations on top of an MPS ground state in the thermodynamic limit. The variational ansatz that we will introduce is a generalization of the single-mode approximation [36], which appeared earlier in the context of spin chains, and the Feynman-Bijl ansatz [37], which was used to describe the spectrum of liquid helium or quantum Hall systems [38]. In the context of MPS, a reduced version of the ansatz appeared earlier in Refs. [39, 40], but it was explored in full generality in Refs. [15, 41]. In recent years, the ansatz has been succesfully applied to spin chains [13, 42, 43], spin ladders [44], spin chains with long-range interactions [45], field theories [46], and local gauge theories [29].

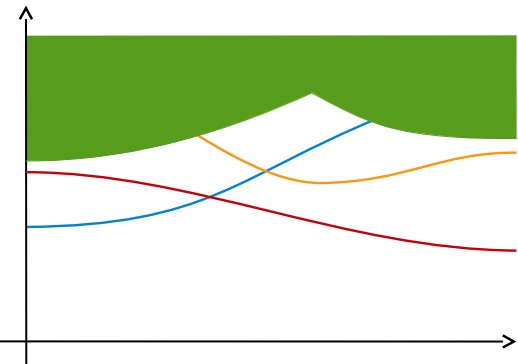

Figure 3: A typical excitation spectrum of a gapped quantum spin chain. The isolated lines are elementary quasiparticle branches, whereas the continuum consists of two-particle states.

The *quasiparticle ansatz* is given by

$$
\begin{aligned}
|\Phi_p(B)\rangle &= \sum_n e^{ipn} \sum_{\{s\}} \boldsymbol{v}_L^\dagger \left[ \prod_{m<n} A_L^{s_m} \right] B^{s_n} \left[ \prod_{m>n} A_R^{s_m} \right] \boldsymbol{v}_R \, |\{s\}\rangle \\
&= \sum_n e^{ipn} \dots \boxed{A_L} - \boxed{A_L} - \boxed{B} - \boxed{A_R} - \boxed{A_R} - , \qquad (176) \\
&\qquad\qquad\quad \dots \quad s_{n-1} \quad s_n \quad s_{n+1} \quad \dots
\end{aligned}
$$

i.e. we change one $A$ tensor of the ground state at site $n$ and make a momentum superposition. In this whole section we work with tangent vectors in the mixed gauge. The newly introduced tensor $B$ contains all the variational parameters of the ansatz, and perturbs the ground state over a finite region around site $n$ in every term of the superposition – it uses the correlations in the ground state, carried over the virtual degrees of freedom in the MPS to create a lump on the background state. Clearly, these excitations have a well-defined momentum, and, as we will see, a finite energy above the extensive energy (and thus, finite energy density) of the ground state.

Before we start optimizing the tensor $B$, we will investigate the variational space in a bit more detail. First note that the excitation ansatz is, in fact, just a boosted version of a tangent vector, so we will be able to apply all tricks and manipulations of the previous sections. For example, the $B$ tensor has gauge degrees of freedom: the state is invariant under an additive gauge transformation of the form

$$
- \boxed{B} - \;\rightarrow\; - \boxed{B} - \;+\; - (Y) - \boxed{A_R} - \;-\; e^{ip} - \boxed{A_L} - (Y) - , \qquad (177)
$$

with $Y$ an arbitrary $D \times D$ matrix. This gauge freedom can be easily checked by substituting this form in the state (176), and observing that all terms cancel, leaving the state invariant.

The gauge degrees of freedom can be eliminated – they correspond to zero modes in the variational subspace, which would make the variational optimization ill-conditioned – by

imposing a gauge fixing condition. Again, we can impose the left gauge-fixing condition[11]

$$= = 0. \qquad (178)$$

We can reuse the method for parametrizing the $B$ tensor such that it automatically obeys this gauge condition:

$$B = V_L \quad X \, . \qquad (179)$$

As before, this fixing of the gauge freedom entails that the excitation is orthogonal to the ground state, because

$$\langle \Psi(A) | \Phi_p(B) \rangle = 2\pi \delta(p) \left( \begin{array}{c} B \\ \bar{A}_c \end{array} \right) = 0. \qquad (180)$$

This effective parametrization has reduced the number of variational parameters in the quasiparticle ansatz to $D^2(d-1)$. Moreover, the overlap between two excitations $|\Phi_p(B)\rangle$ and $|\Phi_{p'}(B')\rangle$ is computed similarly as before: we have two infinite terms, but we can eliminate one sum because of the translation invariance of the ground state. Now this will result in a $2\pi\delta(p-p')$ function,

$$\sum_{n\in\mathbb{Z}} e^{i(p-p')n} = 2\pi\delta(p-p'), \qquad (181)$$

so excitations at different momenta are always orthogonal. Again, the physical norm on the excited states reduces to the Euclidean norm on the effective parameters,

$$\langle \Phi_{p'}(B(X')) | \Phi_p(B(X)) \rangle = 2\pi\delta(p-p')\mathrm{Tr}\left((X')^\dagger X\right). \qquad (182)$$

This will prove to be a useful property for optimizing the variational parameters. The presence of the $\delta$ indicates that these plane wave states cannot be normalized to one, as is well known from single-particle quantum mechanics.

## 6.2 Computing expectation values

Let us first write down the expressions for evaluating expectation values, or more generally, matrix elements of the form

$$\sum_i \langle \Phi_{p'}(B') | O_i | \Phi_p(B) \rangle, \qquad (183)$$

where the ground-state expectation value has already been subtracted, i.e. $\langle \Psi(A) | O_i | \Psi(A) \rangle = 0$. This implies that we will look at expectation values of $O$ relative to the ground state density. As we will see, this will give rise to finite quantities in the thermodynamic limit.

First we notice that the above matrix element is, in fact, a triple infinite sum. Again, one of the sums can be eliminated and yields the factor $2\pi\delta(p-p')$ that also appears in the norm $\langle \Phi_{p'}(B') | \Phi_p(B) \rangle$, so that henceforth we are only interested in all different relative positions of the operator $O$, the $B$ tensor in the ket layer, and the $B'$ tensor in the bra layer. Let us first

---

[11]Only for $p=0$ is there no contribution from the component $Y \sim C$, and does one need to additionally impose orthogonality to the ground state in order to satisfy the $D^2$ gauge fixing equations. For $p \neq 0$, orthogonality to the ground state is immediate.

define two partial contractions, corresponding to the orientations where $O$ is to the left and to the right of both $B$ and $B'$,

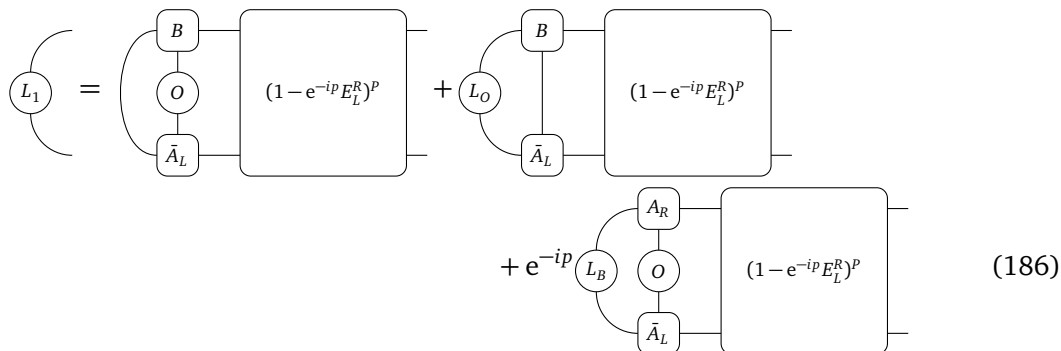

$$\text{(184)}$$

Here, we can again use the pseudo-inverse because $O$ was defined with zero ground state expectation value, so that there is no diverging 'disconnected' contribution.

Similarly, we define the partial contractions where $B$ travels to the outer left or right of the chain[12]:

$$\text{(185)}$$

In these expressions, $(1-e^{\pm ip}E)$ (where $E$ is $E_R^L$ or $E_L^R$) is not singular for $p \neq 0$ and $(1-e^{\pm ip}E)^P$ is not really a pseudo-inverse. It is still defined as subtracting the contribution in the subspace corresponding to eigenvalue 1 of $E$. This arises because the geometric sum $\sum_{n=0}^{\infty} e^{ipn}$ is strictly speaking not defined. Thanks to the gauge fixing condition, there is no contribution in this subspace anyway, so that we could also have used the regular inverse. In the following, this should always be kept in mind whenever a $(\dots)^P$ appears.

We use the above expressions to define all partial contractions where $B$ and $O$ are both either to the left or to the right of $B'$,

$$\text{(186)}$$

and

$$\text{(187)}$$

---

[12]The first of these expressions is actually zero because of the gauge-fixing condition in Eq. (178). In the following we will keep all terms in order to see the symmetry in the different terms, but in an implementation these are of course not computed.

The $e^{\pm ip}$ factors originate from the extra shift in the relative position of $B$ in these terms.

The final expression is

$$\langle \Phi_{p'}(B')| O |\Phi_p(B)\rangle = 2\pi\delta(p-p')$$

$$\Bigg( \text{(diagrams)} \Bigg). \tag{188}$$

### 6.3 Solving the eigenvalue problem

At this point, we still need to find the algorithm for the variational optimization of the $B$ tensor in the excitation ansatz. We have seen that the effective parametrization in terms of an $X$ matrix (i) fixes all gauge degrees of freedom, (ii) removes all zero modes in the variational subspace, (iii) makes the computation of the norm of an excited state particularly easy, and (iv) makes sure the excitation is orthogonal to the ground state, even at momentum zero. The variational optimization boils down to minimizing the energy function,

$$\min_X \frac{\langle \Phi_p(X)| H |\Phi_p(X)\rangle}{\langle \Phi_p(X)|\Phi_p(X)\rangle}, \tag{189}$$

where we have made sure to shift the hamiltonian such that the ground state has energy density zero. Because both numerator and denominator are quadratic functions of the variational parameters $X$, this optimization problem reduces to solving the generalized eigenvalue problem

$$H_{\text{eff}}(q)X = \omega N_{\text{eff}}(q)X, \tag{190}$$

where the effective energy and normalization matrix are defined as

$$2\pi\delta(p-p')(X')^\dagger H_{\text{eff}}(q)X = \langle \Phi_{p'}(X')| H |\Phi_p(X)\rangle \tag{191}$$

$$2\pi\delta(p-p')(X')^\dagger N_{\text{eff}}(q)X = \langle \Phi_{p'}(X')|\Phi_p(X)\rangle, \tag{192}$$

and $X$ is a vectorized version of the matrix $X$. Now since the overlap between two excited states is of the simple Euclidean form, the effective normalization matrix reduces to the unit matrix, and we are left with an ordinary eigenvalue problem.

Solving the eigenvalue problem requires us to find an expression of $H_{\text{eff}}$, or, rather, the action of $H_{\text{eff}}$ on a trial vector. Indeed, since we are typically only interested in finding its lowest eigenvalues, we can plug the action of $H_{\text{eff}}$ (which is hermitian) into a Lanczos-based iterative eigensolver. This has great implications on the computational complexity: The full computation and diagonalization of the effective energy matrix would entail a computational complexity of $\mathcal{O}(D^6)$, while the action on an input vector $Y$ can be done in $\mathcal{O}(D^3)$ operations.

So we need the action of $H_{\text{eff}}$ on an arbitray vector $\boldsymbol{Y}$. We first transform the matrix $Y$ to a tensor $B$ in the usual way. Then we need all different contributions that pop up in a matrix element of the form $\langle \Phi_{p'}(B')| H |\Phi_p(B)\rangle$, i.e. similarly to the expression (188), we need all different orientations of the nearest-neighbour operator of the hamiltonian, the input $B$ tensor and an output. Because we are confronted with a two-site operator here, the expressions are a bit more cumbersome. Let us again define the following partial contractions

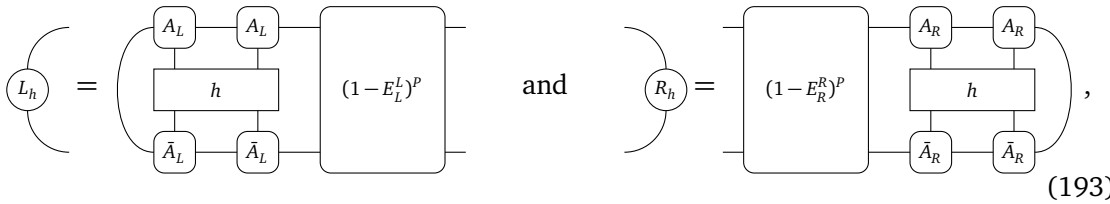

(193)

and

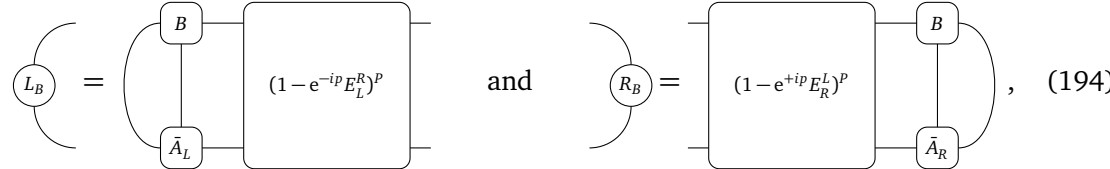

(194)

which we use for determining

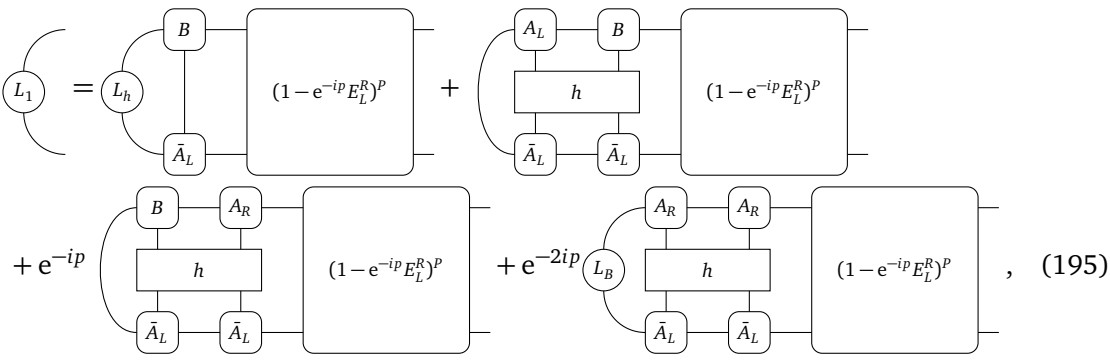

(195)

and

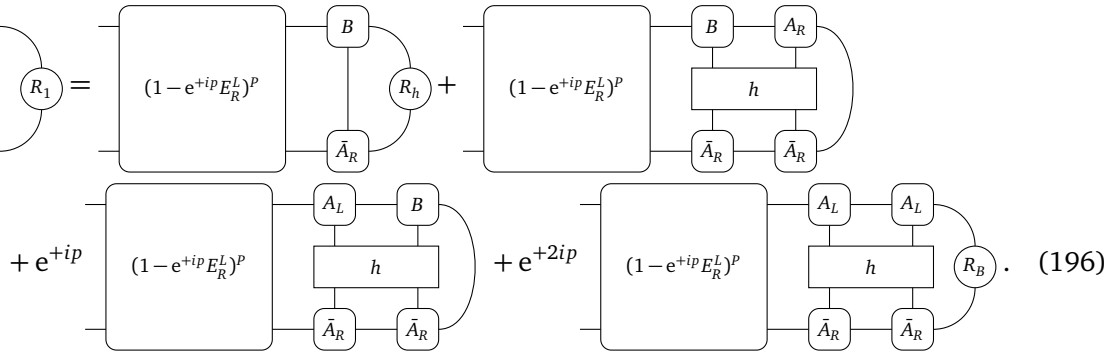

(196)

These partial contractions allow us now to implement the action of the effective energy matrix

on a given input vector **B** as

$$
\tilde{H}_{\text{eff}}(p)B(Y) = \cdots \quad (197)
$$

In the last step, we need the action of $H_{\text{eff}}(p)$ (without the tilde), so we need to perform the last contraction

$$
H_{\text{eff}}(p)X = \cdots . \quad (198)
$$

All contractions above have a computational complexity of $\mathcal{O}(D^3)$.

By solving this eigenvalue equation for all momenta, we obtain direct access to the full excitation spectrum of the system. Note that the eigenvalue equation has $D^2(d-1)$ solutions, but only the few lowest-lying ones have a physical meaning. Indeed, for a given value of the momentum, one typically finds a limited number of excitations living on an isolated branch in the spectrum, whereas all the other solutions fall within the continuous bands. It is not expected that these states are approximated well with the quasiparticle ansatz. The accuracy of the approximation can be assessed by computing the energy variance – just as we did with the ground state in Sec. 4.3 – but, for an excitation this is an involved calculation [44].

---

**Algorithm 7** Quasiparticle excitation ansatz for nearest-neighbour hamiltonian $h$

---

1: **procedure** QUASIPARTICLE($h, p, A, N, \eta$)
     ▷ Find $N$ lowest-lying momentum-$p$ quasiparticles on MPS $|\Psi(A)\rangle$ with tolerance $\eta$
2:     $\{A_L, A_R, C\} \leftarrow$ MIXEDCANONICAL($A, \eta$)                   ▷ Algorithm 2
3:     $e \leftarrow$ EVALUATEENERGY($A_L, A_R, A_C, h$)
4:     $h \leftarrow h - e\mathbb{1}$                            ▷ subtract energy expectation value
5:     $L_h \leftarrow$ SUMLEFT($A_L, \tilde{h}, \delta/10$)                     ▷ Eq. (193)
6:     $R_h \leftarrow$ SUMRIGHT($A_R, \tilde{h}, \delta/10$)                  ▷ Eq. (193)
7:     $(\omega, X) \leftarrow$ ARNOLDI($X \to H_{\text{eff}}(p)X$,'sr',$N, \delta/10$)     ▷ call the function EFFECTIVEH below
8:     **return** $\omega, X$
9: **end procedure**
10: **procedure** EFFECTIVEH($Y, p, \{A_L, A_R, C, A_C\}, L_h, R_h$)
11:     $B \leftarrow$ EFFPARAMATRIZATION($Y, V_L$)                  ▷ Eq. (179)
12:     compute $R_B$ from Eq. (194)
13:     compute $L_1$ and $R_1$ from Eqs. (186) and (187)
14:     compute $\tilde{H}_{\text{eff}}(p)B$ from Eq. (197)
15:     $Y' \leftarrow$ INVEFFPARAMTRIZATION($\tilde{H}_{\text{eff}}(p)B, V_L$)        ▷ Eq. (198)
16:     **return** $Y'$
17: **end procedure**

---

## 6.4 Dynamical correlations

As we have mentioned before, the excitation spectrum determines the dynamical correlation functions or spectral functions. We will use the following definition of the spectral function:

$$S^{\alpha\alpha}(q, \omega) = \int_{-\infty}^{+\infty} dt\, e^{i\omega t} \sum_{n \in \mathbb{Z}} e^{-iqn} \langle \Psi(A)| O_n^\alpha(t) O_0^\alpha(0) |\Psi(A)\rangle, \tag{199}$$

where the time-evolved operator $O_n^\alpha(t) = e^{iHt} O_n^\alpha(0) e^{-iHt}$ is introduced. By projecting the time evolution on all excited states of $H$, we obtain the following representation

$$S^{\alpha\alpha}(q, \omega) = \sum_\gamma \int_{-\infty}^{+\infty} dt\, e^{i\omega t} e^{-i(E_\gamma - E_0)t} \sum_{n \in \mathbb{Z}} e^{-iqn} \langle \Psi(A)| O_n^\alpha(0) |\gamma\rangle \langle \gamma| O_0^\alpha(0) |\Psi(A)\rangle, \tag{200}$$

where $\gamma$ labels all excited states of the system with excitation energies $E_\gamma - E_0$. Let us now take only the one-particle excitations into account (the excitations corresponding to isolated branches in the excitation spectrum), for which we know that they can be described by the excitation ansatz. For these states, which have a well-defined momentum, the sum is rewritten as

$$\sum_{\gamma, 1p} |\gamma\rangle \langle \gamma| = \sum_{\gamma \in \Gamma_1} \int_{\mathcal{R}_\gamma} \frac{dp}{2\pi} |\Phi_p^\gamma(B)\rangle \langle \Phi_p^\gamma(B)|, \tag{201}$$

where we have introduced $\Gamma_1$ as the set of all isolated branches in the spectrum, $\mathcal{R}_\gamma$ as the momentum region where every branch $\gamma$ exists. Because of translation invariance, we have

$$\sum_n e^{-iqn} \langle \Psi(A)| O_n^\alpha(0) |\Phi_p^\gamma(B)\rangle = 2\pi\delta(p - q) \langle \Psi(A)| O_0^\alpha(0) |\Phi_p^\gamma(B)\rangle, \tag{202}$$

so that we obtain for the one-particle part of the spectral function

$$S_{1p}^{\alpha\alpha}(q, \omega) = \sum_{\gamma \in \Gamma_1(q)} 2\pi\delta\left(\omega - \omega_\gamma(q)\right) \left| \langle \Psi(A)| O_0^\alpha(0) |\Phi_p^\gamma(B)\rangle \right|^2, \tag{203}$$

where $\Gamma_1(q)$ denotes the set of one-particle states in the momentum sector $q$ and $\omega_\gamma(\cdot)$ is the excitation energy of that mode.

The spectral weights are easily computed. First, we again define the following partial contractions

$$
L_B = \left( \cdots (1-e^{-ip}E_R^R)^P \cdots \right) \quad \text{and} \quad R_B = \left( (1-e^{+ip}E_L^L)^P \cdots \right), \quad (204)
$$

so that we have the following contractions

$$
\langle \Psi(A)| O_0^\alpha(0) |\Phi_p(B)\rangle = \left( \begin{matrix} B \\ O^\alpha \\ \bar{A}_C \end{matrix} \right) + e^{+ip} \left( \begin{matrix} A_L \\ O^\alpha \; R_B \\ \bar{A}_L \end{matrix} \right) + e^{-ip} \left( \begin{matrix} A_R \\ L_B \; O^\alpha \\ \bar{A}_R \end{matrix} \right). \quad (205)
$$

### 6.5 Topological excitations

The elementary excitations in one-dimensional spin systems are not always of the simple form that we have introduced earlier. In the case of symmetry breaking, where the ground state is degenerate, the elementary excitations are typically kinks or domain walls, i.e. particles that interpolate between the different ground states. These excitations are called topological, as they cannot be created by a local operator acting on one of the ground states. It is not clear that they are local (i.e. that the interpolation between the two ground states is happening over a small region), and, in fact, this is not at all obvious from other approaches such as the Bethe ansatz [47]. One expects, however, that the proof for excitations in the trivial sector in Ref. [35] can be extended to topological excitations, and we can apply the quasiparticle ansatz here as well.

Because it is formulated in the thermodynamic limit directly, our framework can be easily extended to target these topological sectors.[13] Suppose we have a twofold-degenerate ground state, approximated by two uMPS $|\Psi(A)\rangle$ and $|\Psi(\tilde{A})\rangle$. The obvious ansatz for a domain wall excitation is[14]

$$
\begin{aligned}
|\Phi_p(B)\rangle &= \sum_n e^{ipn} \sum_{\{s\}} \mathbf{v}_L^\dagger \left[ \prod_{m<n} A_L^{s_m} \right] B^{s_n} \left[ \prod_{m>n} \tilde{A}_R^{s_m} \right] \mathbf{v}_R |\{s\}\rangle \\
&= \sum_n e^{ipn} \cdots \underbrace{A_L}_{\cdots} \; \underbrace{A_L}_{s_{n-1}} \; \underbrace{B}_{s_n} \; \underbrace{\bar{A}_R}_{s_{n+1}} \; \underbrace{\bar{A}_R}_{\cdots}, \quad (206)
\end{aligned}
$$

i.e. the domain wall interpolates between the two ground states [41]. All the calculations of the previous sections can be repeated in order to determine gauge-fixing conditions, compute expectation values and solve the eigenvalue problem. The only difference concerns the appearance of mixed transfer matrices such as

$$
\tilde{E} = \begin{matrix} \bar{A}_R \\ | \\ \bar{A}_L \end{matrix}, \quad (207)
$$

---

[13]In finite systems with periodic boundary conditions, topological excitations always have to be described in pairs. In order to capture them in finite systems, non-trivial boundary conditions have to be applied.

[14]In quantum field theory, this ansatz has been proposed earlier [48] to study the kink excitations in the sine-Gordon model.

which determines the correlation functions corresponding to string-like operators that interpolate between the two ground states. This matrix has spectral radius smaller than one – otherwise the two ground states would not be orthogonal – such that the geometric sums involving these transfer matrices should be computed with the full inverse.

Yet there is one problem with considering topological excitations. Strictly speaking the momentum of the ansatz [Eq. (206)] is not well defined: multiplying the tensor $\tilde{A}_R$ with an arbitrary phase factor $\tilde{A}_R \leftarrow \tilde{A}_R e^{i\phi}$ shifts the momentum with $p \leftarrow p + \phi$. The origin of this ambiguity is the fact that one domain wall cannot be properly defined when using periodic boundary conditions. Physically, however, domain walls should come in pairs. For these states the total momentum is well defined, although the individual momenta can be arbitrarily transferred between the two domain walls. A heuristic way to fix the kink momentum unambiguously is related to the above mixed transfer matrix; it can be imposed that its spectrum be symmetric with respect to the real axis. This will give rise to a kink spectrum that is symmetric in the momentum. This problem disappears, as we will see, when considering excitations with two topological particles.

## 6.6 Larger blocks

There is no guarantee that the variational energies converge to the exact excitation energy of the full hamiltonian, even for a clearly isolated excitation branch. The reason is that the effect of physical operators of growing size cannot always be reproduced by the excitation ansatz, even by growing the bond dimension. This can pose a problem for one-particle excitations that are very wide, because e.g. they are very close to a scattering continuum in the spectrum.

The excitation ansatz can be systematically extended, however, in order to capture larger and larger regions. Instead of inserting a one-site tensor, one can introduce larger blocks, which leads to the ansatz [15, 35]

$$|\Phi_p(B)\rangle = \sum_n e^{ipn} \dots \boxed{A_L} - \boxed{A_L} - \boxed{\phantom{xxxxxx} B \phantom{xxxxxx}} - \boxed{\tilde{A}_R} - \boxed{\tilde{A}_R} - . \tag{208}$$

In principle this approach is guaranteed to converge to the exact excitation energy – assuming the ground state energy is converged – but the number of the variational parameters in the big $B$ tensor grows exponentially in the number of sites, so that, practically, this becomes infeasible quickly.

The same gauge freedom is present for these larger blocks, and the same gauge conditions can be imposed. The left gauge condition reads

$$\left( \begin{array}{c} \boxed{B} \\ \boxed{\bar{A}_L} \end{array} \right) = 0, \tag{209}$$

and can be enforced by going to the effective parametrization of the $B$ tensor

$$-\boxed{\phantom{xx} B \phantom{xx}}- = -\boxed{V_L}-\boxed{\phantom{x} X \phantom{x}}-, \tag{210}$$

where $X^{s_2,\dots,s_M}$ is a $(D(d-1) \times d \times \dots \times d \times D)$ tensor containing all variational parameters. With this effective parametrization, the overlap of states again reduces to the Euclidean norm on the tensor $X$, and the variational optimization reduces to an eigenvalue problem. For further details of this implementation we refer to Ref. [49].

## 6.7 Two-particle states

The excitations that were introduced in the previous section can be naturally interpreted as particles living on a strongly-correlated background state, and we can ask the question as to how to describe the interactions between these effective particles [42, 44, 50]. As an answer to that question, in this section we show how to construct two-particle states and how to compute the two-particle S matrix. We will start from a one-particle spectrum consisting of a number of different types of particles, labelled by $\alpha$, with dispersion relations $\Delta_\alpha(p)$. In the thermodynamic limit, constructing the two-particle spectrum is trivial: the momentum and energy are the sum of the individual momenta and energies of the two particles. The two-particle wavefunction, however, depends on the particle interactions. These depend on both the hamiltonian and the ground state correlations, and are reflected in the wavefunction in two ways: (i) the asymptotic wavefunction has different terms, with the S matrix elements as the relative coefficients, and (ii) the local part of the wavefunction.

**Variational ansatz**

In order to capture both effects of the interactions on the wavefunction, we introduce the following ansatz for describing states with two localized, particle-like excitations with total momentum $P$

$$|\Upsilon(P)\rangle = \sum_{n=0}^{+\infty} \sum_{j=1}^{L_n} c^j(n) |\chi_{P,j}(n)\rangle \,, \tag{211}$$

where the basis states are

$$|\chi_{P,j}(n=0)\rangle = \sum_{n_1=-\infty}^{+\infty} e^{iPn_1} \sum_{\{s\}=1}^{d} \boldsymbol{v}_L^\dagger \left[\prod_{m<n_1} A^{s_m}\right] B_{(j)}^{s_{n_1}} \left[\prod_{m>n_1} A^{s_m}\right] \boldsymbol{v}_R |\{s\}\rangle \tag{212}$$

$$|\chi_{P,(j_1,j_2)}(n>0)\rangle = \sum_{n_1=-\infty}^{+\infty} e^{iPn_1} \sum_{\{s\}=1}^{d} \boldsymbol{v}_L^\dagger \left[\prod_{m<n_1} A^{s_m}\right] B_{(j_1)}^{s_{n_1}}$$

$$\times \left[\prod_{n_1<m<n_1+n} A^{s_m}\right] B_{(j_2)}^{s_{n_1+n}} \left[\prod_{m>n_1+n} A^{s_m}\right] \boldsymbol{v}_R |\{s\}\rangle \,. \tag{213}$$

We collect the variational coefficients either in one half-infinite vector $\boldsymbol{C}$ with $C^{j,n} = c^j(n)$ or using the finite vectors $\boldsymbol{c}(n)$ with entries $\{c^j(n), j=1,\ldots,L_n\}$ for every $n=0,1,\ldots$. Here, we have $L_0 = (d-1)D^2$ and $L_{n>0} = [(d-1)D^2]^2$. Note that the sum in Eq. (211) only runs over values $n \geq 0$, because a sum over all integers would result in an overcomplete basis.

At this point, we will reduce the number of variational parameters to keep the problem tractable. The terms with $n=0$ (corresponding to the basis vectors in Eq. (212)) are designed to capture the situation where the two particles are close together. No information on how this part should look like is available a priori, so we keep all variational parameters $c^j(0)$, $j=1,\ldots,L_0 = D^2(d-1)$. The terms with $n>0$ corresponding to the basis vectors in Eq. (213) represent the situation where the particles are separated. We know that, as $n \to \infty$, the particles decouple and we should obtain a combination of one-particle solutions. With this in mind, we restrict the range of $j_1$ and $j_2$ to the first $\ell$ basis tensors $\{B_{(i)}, i=1,\ldots,\ell\}$, which can be chosen so as to capture the momentum dependent solutions of the one-particle problem. Consequently, the number of basis states of Eq. (213) for $n>0$ satisfies $L_n = \ell^2$, which we will henceforth denote as just $L$.

This might seem like a big approximation for small $n$: when the two particles approach, their wavefunctions might begin to deform, so that the $B$ tensors that were obtained as solutions for the one-particle problem, no longer apply. Note, however, that the local ($n = 0$) and non-local ($n > 0$) part are not orthogonal, so that the local part is able to correct for the part of the non-local wavefunction where the one-particle description is no longer valid.

As the state (211) is again linear in its variational parameters $C$, optimizing the energy amounts to solving a generalized eigenvalue problem

$$H_{\text{eff,2p}}(P)C = \omega N_{\text{eff,2p}}(P)C\,, \tag{214}$$

with $\omega$ the total energy of the state and

$$(H_{\text{eff,2p}}(P))_{n'j',nj} = \langle \chi_{P,j'}(n')| H |\chi_{P,j}(n)\rangle \tag{215}$$

$$(N_{\text{eff,2p}}(P))_{n'j',nj} = \langle \chi_{P,j'}(n')|\chi_{P,j}(n)\rangle \tag{216}$$

two half-infinite matrices. They have a block matrix structure, where the submatrices are labelled by $(n', n)$ and are of size $L_{n'} \times L_n$. The computation of the matrix elements is quite involved and technical – each element contains three infinite sums with each term containing two $B$ tensors in both ket and bra layer – so we refer to Ref. [49] for the explicit formulas.

Since the eigenvalue problem is still infinite, it cannot be diagonalized directly. Since we actually know the possible energies $\omega$ for a scattering state with total momentum $P$ (it follows from the one-particle energies), we can also interpret Eq. (214) as an overdetermined system of linear equations for the coefficients $C^{j,n} = c^j(n)$. In the next two sections we will show how to reduce this problem to a finite linear equation.

**Asymptotic regime**

First we solve the problem in the asymptotic regime, where the two particles are completely decoupled. This regime corresponds to the limit $n', n \to \infty$, where the effective norm and hamiltonian matrices, consisting of blocks of size $L \times L$, take on a simple form. Indeed, if we properly normalize the basis states, the asymptotic form of the effective norm matrix reduces to the identity, while the effective hamiltonian matrix is a repeating row of block matrices centred around the diagonal

$$(H_{\text{eff,2p}}(P))_{n',n} \to A_{n-n'}, \qquad n, n' \to \infty. \tag{217}$$

The blocks decrease exponentially as we go further from the diagonal, so we can, in order to solve the problem, consider them to be zero if $|n - n'| > M$ for a sufficiently large $M$. In this approximation, the coefficients $c(n)$ obey

$$\sum_{m=-M}^{M} A_m c(n+m) = \omega c(n), \qquad n \to \infty. \tag{218}$$

We can reformulate this as a recurrence relation for the $c(n)$ vectors and therefore look for elementary solutions of the form $c(n) = \mu^n v$. For fixed $\omega$, the solutions $\mu$ and $v$ are now determined by the polynomial eigenvalue equation

$$\sum_{m=-M}^{M} A_m \mu^m v = \omega v. \tag{219}$$

From the special structure of the blocks $A_m$ [49] and their relation to the effective one-particle hamiltonian $H_{\text{eff}}(p)$, we already know a number of solutions to Eq. (219). Indeed, if we can find $\Gamma$ combinations of two types of particles $(\alpha, \beta)$ with individual momenta $(p_1, p_2)$ such that

$P = p_1 + p_2$ and $\omega = \Delta_\alpha(p_1) + \Delta_\beta(p_2)$, then the polynomial eigenvalue problem will have $2\Gamma$ solutions $\mu$ on the unit circle. These solutions take the form $\mu = e^{ip_2}$ and the corresponding eigenvectors are given by

$$\boldsymbol{v} = \boldsymbol{u}_\alpha(p_1) \otimes \boldsymbol{u}_\beta(p_2), \tag{220}$$

where $\boldsymbol{u}_\alpha(p)$ is a vector corresponding to the one-particle solution of type $\alpha$ with momentum $p$ with respect to the reduced basis $\{B_{(i)}, i = 1, \ldots, \ell\}$ (in the case of degenerate eigenvalues we can take linear combinations of these eigenvectors that no longer have this product structure). Every combination is counted twice, because we can have particle with momentum $p_1$ on the left and momentum $p_2$ on the right, and vice versa.

Moreover, since $A_m^\dagger = A_{-m}$, the number of eigenvalues within and outside the unit circle should be equal. This allows for a classification of the eigenvalues $\mu$ as

$$\left|\mu_i\right| < 1 \quad \text{for} \quad i = 1, \ldots, LM - \Gamma \tag{221}$$

$$\left|\mu_i\right| = 1 \quad \text{for} \quad i = LM - \Gamma + 1, \ldots, LM + \Gamma \tag{222}$$

$$\left|\mu_i\right| > 1 \quad \text{for} \quad i = LM + \Gamma + 1, \ldots, 2LM. \tag{223}$$

The last eigenvalues with modulus bigger than one are not physical (because the corresponding $\boldsymbol{c}(n) \sim \mu_i^n \boldsymbol{v}_i$ yields a non-normalizable state) and should be discarded. The $2\Gamma$ eigenvalues with modulus 1 are the oscillating modes discussed above; we will henceforth label them with $\gamma = 1, \ldots, 2\Gamma$ such that $\mu = e^{ip_\gamma}$ ($p_\gamma$ being the momentum of the particle of the right) and the corresponding eigenvector is given by

$$\boldsymbol{v}_\gamma = \boldsymbol{u}_{\alpha_\gamma}(P - p_\gamma) \otimes \boldsymbol{u}_{\beta_\gamma}(p_\gamma). \tag{224}$$

Finally, the first eigenvalues are exponentially decreasing and represent corrections when the excitations are close to each other. We henceforth denote them as $e^{-\lambda_i}$ with $\text{Re}(\lambda_i) > 0$ for $i = 1, \ldots, LM - \Gamma$ and denote the corresponding eigenvectors as $\boldsymbol{w}_i$.

With these solutions, we can represent the general asymptotic solution as

$$\boldsymbol{c}(n) \to \sum_{i=1}^{LM-\Gamma} q^i e^{-\lambda_i n} \boldsymbol{w}_i + \sum_{\gamma=1}^{2\Gamma} r^\gamma e^{ip_\gamma n} \boldsymbol{v}_\gamma. \tag{225}$$

Of course, we still have to determine the coefficients $\{q^i, r^\gamma\}$ by solving the local problem.

**Solving the full eigenvalue equation**

Since the energy $\omega$ was fixed by the solution of the asymptotic problem, the generalized eigenvalue equation is reduced to the linear equation

$$(H_{\text{eff,2p}} - \omega N_{\text{eff,2p}})\boldsymbol{C} = 0. \tag{226}$$

We know that in the asymptotic regime this equation is fulfilled if and only if $\boldsymbol{c}(n)$ is of the form of Eq. (225). We will introduce the approximation that the elements for the effective hamiltonian matrix [Eq. (215)] and norm matrix [Eq. (216)] have reached their asymptotic values when either $n > M + N$ or $n' > M + N$, where $N$ is a finite value and should be chosen sufficiently large. This implies that we can safely insert the asymptotic form for $n > N$ in the wavefunction, which we can implement by rewriting the wavefunction as

$$\boldsymbol{C} = \boldsymbol{Z} \cdot \boldsymbol{x}, \tag{227}$$

where

$$\boldsymbol{Z} = \begin{pmatrix} \mathbb{1}_{\text{local}} & & \\ & \{e^{-\lambda_i n} \boldsymbol{w}_i\} & \{e^{-ip_\gamma n} \boldsymbol{v}_\gamma\} \end{pmatrix}.$$

The $\{e^{-\lambda_i n} \boldsymbol{w}_i\}$ and $\{e^{-ip_\gamma n} \boldsymbol{v}_\gamma\}$ are the vectors corresponding to the damped, resp. oscillating modes, while the identity matrix is inserted to leave open all parameters in $\boldsymbol{c}(n)$ for $n \leq N$. The number of parameters in $x$ is reduced to the finite value of $D^2(d-1) + NL + LM + \Gamma$.

Since the equation is automatically fulfilled after $M + N$ rows, we can reduce $H_{\text{eff,2p}}$ and $N_{\text{eff,2p}}$ to the first rows, so we end up with the following linear equation

$$[H - \omega N]_{\text{red}} \cdot \boldsymbol{Z} \cdot \boldsymbol{x} = 0, \tag{228}$$

with

$$[H - \omega N]_{\text{red}} = \left( \begin{array}{c|cccc} & 0 & 0 & \dots & 0 \\ & \vdots & \vdots & \ddots & \vdots \\ & 0 & 0 & \dots & 0 \\ (H_{\text{eff,2p}} - \omega N_{\text{eff,2p}})_{\text{ex}} & A_M & 0 & \dots & 0 \\ & A_{M-1} & A_M & \dots & 0 \\ & \vdots & \vdots & \ddots & \vdots \\ & A_1 & A_2 & \dots & A_M \end{array} \right). \tag{229}$$

This 'effective scattering matrix' consists of the first $(M + N) \times (M + N)$ blocks of the exact effective hamiltonian and norm matrix and the $A$ matrices of the asymptotic part [Eq. (217)] to make sure that these matrices remain the truncated versions of a hermitian problem. This matrix has $D^2(d-1) + (N+M)L$ rows, which implies that the linear equation (228) has $\Gamma$ exact solutions, which is precisely the number of scattering states we expect to find. Every solution consists of a local part ($D^2(d-1) + NL$ elements), the $LM - \Gamma$ coefficients $\boldsymbol{q}$ of the decaying modes and the $2\Gamma$ coefficients $\boldsymbol{r}$ of the asymptotic modes.

**S matrix and normalization**

After having shown how to find the solutions of the scattering problem, we can now elaborate on the structure of the asymptotic wavefunction and define the S matrix.

We start from $\Gamma$ linearly independent scattering eigenstates $|\Upsilon_i(P, \omega)\rangle$ $(i = 1, \dots, \Gamma)$ at total momentum $P$ and energy $\omega$ with asymptotic coefficients $\boldsymbol{r}_i(P, \omega)$. The asymptotic form of these eigenstates is thus a linear combination of all possible non-decaying solutions of the asymptotic problem:

$$|\Upsilon_i(P, \omega)\rangle_{\text{as}} = \sum_{\gamma=1}^{2\Gamma} r_i^\gamma(P, \omega) \times \sum_{n > N} \sum_j e^{ip_\gamma n} v_\gamma^j(p_\gamma) |\chi_{j,P}(n)\rangle, \tag{230}$$

where the coefficients are obtained from solving the local problem. The number of eigenstates equals half the number of oscillating modes that appear in the linear combination. With every oscillating mode $\gamma$ we can associate a function $\omega_\gamma(p)$ giving the energy of this mode as a function of the momentum $p_\gamma$ of the second particle at a fixed total momentum $P$. If $\gamma$ corresponds to the two-particle mode with particles $\alpha_\gamma$ and $\beta_\gamma$, this function is given by $\omega_\gamma(p) = \Delta_{\alpha_\gamma}(P - p) + \Delta_{\beta_\gamma}(p)$. The derivative of this function, which will prove of crucial importance, is $\omega_\gamma'(p) = \Delta_{\beta_\gamma}'(p) - \Delta_{\alpha_\gamma}'(P - p)$. It can be interpreted as the difference in group velocity between the two particles, i.e. the relative group velocity in the center of mass frame.

Much like the proof of conservation of particle current in one-particle quantum mechanics, it can be shown [49] that, if (230) is to be the asymptotic form of an eigenstate, the coefficients $r_i^\gamma(P, \omega)$ should obey

$$\sum_\gamma \left| r_i^\gamma(P, \omega) \right|^2 \left( \frac{d\omega_\gamma}{dp}(p_\gamma) \right) = 0. \tag{231}$$

This equation can indeed be read as a form of conservation of particle current, with $\omega_\gamma'(p_\gamma)$ playing the role of the (relative) group velocity of the asymptotic mode $\gamma$. As any linear

combination of eigenstates with the same energy $\omega$ is again an eigenstate, this relation can be extended to

$$\sum_\gamma \overline{r_j^\gamma(P,\omega)} r_i^\gamma(P,\omega) \left( \frac{\mathrm{d}\omega_\gamma}{\mathrm{d}p}(p_\gamma) \right) = 0. \tag{232}$$

With this equation satisfied, we can define the two-particle S matrix $S(P,\omega)$. Firstly, the different modes are classified according to the sign of the derivative: the incoming modes have $\frac{\mathrm{d}\omega}{\mathrm{d}p} > 0$ (two particles moving towards each other), the outgoing modes have $\frac{\mathrm{d}\omega}{\mathrm{d}p} < 0$ (two particles moving away from each other), so that we have

$$\sum_{\gamma \in \Gamma_{\mathrm{in}}} \overline{r_j^\gamma(P,\omega)} r_i^\gamma(P,\omega) \left| \frac{\mathrm{d}\omega_\gamma}{\mathrm{d}p}(p_\gamma) \right| = \sum_{\gamma \in \Gamma_{\mathrm{out}}} \overline{r_j^\gamma(P,\omega)} r_i^\gamma(P,\omega) \left| \frac{\mathrm{d}\omega_\gamma}{\mathrm{d}p}(p_\gamma) \right|. \tag{233}$$

If we group the coefficients of all solutions in (square) matrices $R_{\mathrm{in}}(P,\omega)$ and $R_{\mathrm{out}}(P,\omega)$, so that the $i$'th column is a vector with the coefficients $r_i^\gamma$ for the in- and outgoing modes of the $i$'th solution, we can rewrite this equation as

$$R_{\mathrm{in}}(P,\omega)^\dagger V_{\mathrm{in}}^2(P,\omega) R_{\mathrm{in}}(P,\omega) = R_{\mathrm{out}}(P,\omega)^\dagger V_{\mathrm{out}}^2(P,\omega) R_{\mathrm{out}}(P,\omega), \tag{234}$$

with $V_{\mathrm{in,out}}(P,\omega)_{ij} = \delta_{ij} \left| \frac{\mathrm{d}\omega_\gamma}{\mathrm{d}p}(p_\gamma) \right|^{1/2}$ a diagonal matrix. As $R_{\mathrm{in}}(P,\omega)$ and $R_{\mathrm{out}}(P,\omega)$ should be connected linearly, we can define a unitary matrix $S(P,\omega)$ as

$$V_{\mathrm{out}}(P,\omega) R_{\mathrm{out}}(P,\omega) = S(P,\omega) V_{\mathrm{in}}(P,\omega) R_{\mathrm{in}}(P,\omega). \tag{235}$$

This definition corresponds to the S matrix that is known in standard scattering theory. Note, however, that $S(P,\omega)$ is only defined up to a set of phases. Indeed, since the vectors $\boldsymbol{v}_\gamma$ [Eq. (224)] can only be determined up to a phase, the coefficient matrices $R_{\mathrm{in}}$ and $R_{\mathrm{out}}$ are only defined up to a diagonal matrix of phase factors. These arbitrary phase factors show up in the S matrix as well. In the case of elastic scattering of two identical particles the phase can be fixed heuristically; in the case where we have different outgoing channels only the square of the magnitude of the S-matrix elements is physically well-defined.

This formalism allows to calculate the norm of the scattering states in an easy way. Indeed, the general overlap between two scattering states is given by

$$\langle \Upsilon_{i'}(P',\omega') | \Upsilon_i(P,\omega) \rangle$$
$$= 2\pi\delta(P - P') \left( \sum_{\gamma,\gamma'} \overline{r_{i'}^{\gamma'}(P',\omega')} r_i^\gamma(P,\omega) \boldsymbol{v}_{\gamma'}^\dagger \boldsymbol{v}_\gamma \sum_{n,n'>N} e^{i(p_\gamma - p_{\gamma'}')n} + \text{finite} \right) \tag{236}$$
$$= 2\pi\delta(P - P') \left( \sum_{\gamma,\gamma'} \overline{r_{i'}^{\gamma'}(P',\omega')} r_i^\gamma(P,\omega) \boldsymbol{v}_{\gamma'}^\dagger \boldsymbol{v}_\gamma \pi\delta(p_\gamma(\omega) - p_{\gamma'}'(\omega')) + \text{finite} \right). \tag{237}$$

The $\delta$ factor for the momenta $p_\gamma$ is obviously only satisfied if $\omega = \omega'$, so we can transform this to a $\delta(\omega - \omega')$. Moreover, if $p_\gamma(\omega) = p_{\gamma'}'(\omega')$ for $\gamma \neq \gamma'$, then necessarily $\boldsymbol{v}_{\gamma'}^\dagger \boldsymbol{v}_\gamma = 0$, so we can reduce the double sum in $\gamma,\gamma'$ to a single one. If we omit all finite parts, we have

$$\langle \Upsilon_{i'}(P',\omega') | \Upsilon_i(P,\omega) \rangle = 2\pi\delta(P - P')\pi\delta(\omega - \omega') \sum_\gamma \overline{r_{i'}^\gamma(P',\omega')} r_i^\gamma(P,\omega) \left| \frac{\mathrm{d}\omega_\gamma}{\mathrm{d}p}(p_\gamma) \right|. \tag{238}$$

With the $R_{\mathrm{in/out}}$ as defined above the overlap reduces to

$$\langle \Upsilon_{i'}(P',\omega') | \Upsilon_i(P,\omega) \rangle = 2\pi\delta(P - P')2\pi\delta(\omega - \omega') \left[ R_{\mathrm{in}}(P,\omega) \right]_{i'}^\dagger V_{\mathrm{in}}^2(P,\omega) \left[ R_{\mathrm{in}}(P,\omega) \right]_i \tag{239}$$
$$= 2\pi\delta(P - P')2\pi\delta(\omega - \omega') \left[ R_{\mathrm{out}}(P,\omega) \right]_{i'}^\dagger V_{\mathrm{out}}^2(P,\omega) \left[ R_{\mathrm{out}}(P,\omega) \right]_i. \tag{240}$$

**Two-particle contribution to spectral functions**

Similar to the one-particle contributions to the spectral functions, we can now compute the two-particle contribution as well. The projector on the two-particle subspace can be written as

$$\int \frac{\mathrm{d}P}{2\pi} \int \frac{\mathrm{d}\omega}{2\pi} \sum_{\gamma \in \Gamma_2(P,\omega)} |\Upsilon_\gamma(P,\omega)\rangle \langle \Upsilon_\gamma(P,\omega)| , \tag{241}$$

where $\Gamma_2$ is the set of all types of two-particle states at that momentum-energy. Here we have orthonormalized the two-particle states as

$$\langle \Upsilon_{\gamma'}(P',\omega')|\Upsilon_\gamma(P,\omega)\rangle = 4\pi^2 \delta(P'-P)\delta(\omega'-\omega)\delta_{\gamma\gamma'}. \tag{242}$$

The two-particle contribution to the spectral function is then given by

$$S_{2p}^{\alpha\alpha}(q,\omega) = \sum_{\gamma \in \Gamma_2(q,\omega)} \left| \langle \Psi(\bar{A})| O_0^\alpha(0) |\Upsilon_\gamma(q,\omega)\rangle \right|^2. \tag{243}$$

As compared to the one-particle contribution, this function is continuous (no $\delta$ peaks) in the momentum-energy region where two-particle states exist. The expressions for the spectral weights can be found in Ref. [44].

**Bound states**

Above we have seen how the one-particle ansatz can be extended to larger blocks in order to describe very broad excitations, a situation that arises when a bound state forms out of a two-particle continuum. We could, however, study these bound states with the two-particle ansatz as well. Specifically, the formation of a bound state out of a two-particle continuum should correspond to a continuous deformation of a two-particle wavefunction into a very broad, yet localized one-particle wavefunction. As the asymptotic part of the two-particle wavefunction is supposed to vanish in this process, we expect a non-analyticity in the S matrix – in particular, the scattering length diverges as the bound state forms [44].

In contrast to a scattering state the energy of a bound state is not known from the one-particle dispersions, so that we will have to scan a certain energy range in search of bound state solutions – of course, with the one-particle ansatz we can get a pretty good idea where to look. A bound state corresponds to solutions for the eigenvalue equation with only decaying modes in the asymptotic regime. In principle we should even be able to find bound-state solutions within a continuum of scattering states (i.e. a stationary bound-state, not a resonance within the continuum) by the presence of additional localized solutions for the scattering problem.

# 7 Transfer matrices and fixed points

Matrix product states have been used extensively as variational ansatz for ground states of local hamiltonians, but in the last years it has been observed that they can also provide accurate approximations for fixed points of *transfer matrices*. One-dimensional transfer matrices pop up whenever we want to contract two-dimensional tensor networks, which occur naturally in the context of two-dimensional classical many-body systems as representations of partition functions and can represent ground states and real-time evolution of one-dimensional quantum systems, e.g. for systems with local interactions in terms of Trotter-Suzuki decompositions. Additionally, they occur in the context of projected entangled-pair states (PEPS), the two-dimensional version of matrix product states. [51]

The contraction of a two-dimensional tensor network using MPS methods goes back to the corner transfer matrix of Baxter [52, 53] and the work of Nishino and Okunishi on classical partition functions in two dimensions [54, 55]. Ten years later these works led to contraction algorithms based on the time-evolving block decimation [17] or the corner transfer matrix renormalization group [56]. Complementary to these approaches, in this section we formulate tangent-space methods for one-dimensional transfer matrices [57].

A one-dimensional transfer matrix in the form of *matrix product operator* (MPO) [12, 58] is written as

$$T(O) = \sum_{\{i\}\{j\}} \left( \ldots O^{i_{n-1},j_{n_1}} O^{i_{n-1},j_{n_1}} O^{i_{n-1},j_{n_1}} \ldots \right)$$

$$\ldots |i_{n-1}\rangle \langle j_{n-1}| \otimes |i_n\rangle \langle j_n| \otimes |i_{n+1}\rangle \langle j_{n+1}| \ldots, \quad (244)$$

or represented diagrammatically as

$$T(O) = \ldots - \boxed{O} - \boxed{O} - \boxed{O} - \boxed{O} - \boxed{O} - \ldots. \quad (245)$$

Whenever we contract an infinite two-dimensional tensor network, we want to find the fixed point of this operator, i.e. we want to solve the fixed-point equation

$$T(O)|\Psi\rangle \propto |\Psi\rangle. \quad (246)$$

We now make the ansatz that the fixed point (leading eigenvector) of this operator is an MPS, such that it obeys the eigenvalue equation

$$\ldots \quad \propto \ldots - \boxed{A} - \boxed{A} - \boxed{A} - \boxed{A} - \boxed{A} - \ldots. \quad (247)$$

Let us first try to find a way to properly define this eigenvalue equation. Suppose we have indeed found an MPS representation $|\Psi(A)\rangle$ of the fixed point of $T(O)$, then the eigenvalue is given by

$$\Lambda = \langle \Psi(A)| T |\Psi(A)\rangle. \quad (248)$$

In order to determine $\Lambda$, we bring $|\Psi(A)\rangle$ in the mixed canonical form, such that

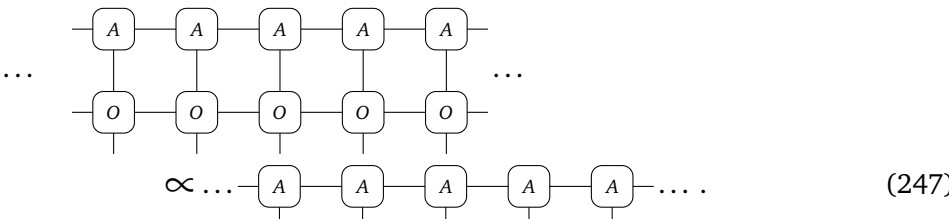

$$\Lambda = \ldots \quad \ldots. \quad (249)$$

Contracting this infinite network requires that we find $F_L$ and $F_R$, the fixed points of the left and right channel operators and $T_L$ and $T_R$, which are represented diagrammatically as

$$(250)$$

The fixed points $F_L$ and $F_R$ are normalized such that

$$\begin{matrix} \text{(diagram)} \end{matrix} = 1\,. \tag{251}$$

The eigenvalues $\lambda_L$ and $\lambda_R$ are necessarily the same value $\lambda$, so that $\Lambda$ is given by

$$\Lambda = \lim_{N\to\infty} \lambda^N, \tag{252}$$

where $N$ is the diverging number of sites. From a physical point of view, it is the 'free energy density' $f = -\frac{1}{N}\log\Lambda = -\log\lambda$ that is the most important quantity. In the case that we want to normalize the MPO, such that the leading eigenvalue is equal to one (or $f = 0$), we can just divide by $\lambda$: $O \to O/\lambda$.

## 7.1 The vumps algorithm for MPOs

The next step towards an algorithm [57,59] is stating an optimality condition for $|\Psi(A)\rangle$ such that it can serve as an approximate eigenvector of $T(O)$. Inspired by all the above tangent-space algorithms, we will require that the projection of the residual onto the tangent space is zero:

$$|\Phi(G;A_L,A_R)\rangle = \mathcal{P}_A\big(T(O)|\Psi(A)\rangle - \Lambda|\Psi(A)\rangle\big) = 0. \tag{253}$$

In the mixed canonical form, the tangent-space projector consists of two parts, yielding

$$|\Phi(G;A_L,A_R)\rangle = \sum_n \left( \cdots \begin{matrix} \text{(diagram)} \end{matrix} \cdots \right)$$
$$- \left( \cdots \begin{matrix} \text{(diagram)} \end{matrix} \cdots \right), \tag{254}$$

or using the left and right fixed points

$$|\Phi(G;A_L,A_R)\rangle \propto \sum_n \lambda^{-1} \left( \ldots \begin{array}{c} A_C \\ F_L \quad O \quad F_R \\ A_L \quad A_L \qquad A_R \quad A_R \end{array} \ldots \right)$$

$$- \left( \ldots \begin{array}{c} C \\ F_L \qquad F_R \\ A_L \quad A_L \qquad A_R \quad A_R \end{array} \ldots \right), \quad (255)$$

or in terms of the tensor $G = O_{A_C}(A_C) - A_L O_C(C)$ with the maps

$$O_{A_C} : \quad -X- \rightarrow \begin{array}{c} X \\ F_L \quad O \quad F_R \end{array} \times \lambda^{-1}, \quad (256)$$

and

$$O_C : \quad -X- \rightarrow \begin{array}{c} X \\ F_L \qquad F_R \end{array}. \quad (257)$$

Now the condition for having an optimal MPS representation is equivalent to having $\|G\| = 0$. Together with the consistency conditions, a fixed point is thus characterized by the set of equations

$$O_{A_C}(A_C) \propto A_C \tag{258}$$

$$O_C(C) \propto C \tag{259}$$

$$A_C = A_L C = C A_R. \tag{260}$$

In Sec. 4.4 we have seen how the vumps algorithm finds the fixed point iteratively. In every iteration of the algorithm, we (i) start from a given MPS $\{A_L^i, A_R^i, A_C^i, C^i\}$, (ii) determine $F_L$ and $F_R$, (iii) solve the two eigenvalue equations obtaining $A_C'$ and $C'$, and (iv) determine the $A_L^{i+1}$ and $A_R^{i+1}$ that minimize $\|A_C' - A_L^{i+1} C'\|$ and $\|A_C' - C' A_R^{i+1}\|$. The vumps algorithm for MPOs is summarized in Alg. 8.

---

**Algorithm 8** Find the optimal MPS approximation for the fixed point of the MPO $\hat{O}(T)$

---

1: **procedure** VUMPS($O, A, \eta$)           ▷ Initial guess $A$ and a tolerance $\eta$
2:  $\{A_L, A_R, C, A_C\} \leftarrow$ MIXEDCANONICAL($A, \eta$)       ▷ Algorithm 2
3:  **repeat**
4:    $\lambda, F_L \leftarrow$ FIXEDPOINTLEFT($A_L, O, \delta/10$)       ▷ Eq. (250)
5:    $\sim, F_R \leftarrow$ FIXEDPOINTRIGHT($A_R, O, \delta/10$)      ▷ Eq. (250)
6:    $F_L \leftarrow F_L/$OVERLAPFIXEDPOINTS($F_L, F_R, C$)     ▷ Eq. (251)
7:    $(\sim, A_C') \leftarrow$ ARNOLDI($X \rightarrow O_{A_C}(X), A_C,$ 'lm', $\delta/10$) ▷ the map $O_{A_c}$ in Eq. (256)
8:    $(\sim, C') \leftarrow$ ARNOLDI($X \rightarrow O_C(X), C,$ 'lm', $\delta/10$)  ▷ the map $O_C$ in Eq. (257)
9:    $(A_L, A_R, C, A_C) \leftarrow$ MINACC($A_C', C'$)
10:   $\delta \leftarrow \|O_{A_C}(A_C)/ - O_C(C)\|$
11:  **until** $\delta < \eta$
12:  **return** $\lambda, A_R$
13: **end procedure**

---

## 7.2 Excited states of an MPO

We can also apply the excitation ansatz to compute 'excitations' of a transfer matrix [51, 60, 61]. The algorithms for computing dispersion relations are quite similar to the case of hamiltonians, which we have studied extensively. In a first step, we renormalize the MPO such that the eigenvalue $\lambda$ of the fixed point equation equals one. Then we use the excitation ansatz,

$$|\Phi_p(B)\rangle = \sum_n e^{ipn} \dots \boxed{A_L} - \boxed{A_L} - \boxed{B} - \boxed{A_R} - \boxed{A_R} -$$
$$\dots \quad\quad s_{n-1} \quad s_n \quad s_{n+1} \quad \dots$$

to find the subleading eigenvectors. Again, we take recourse to the effective parametrization

$$-\boxed{B}- = -\boxed{V_L} - \boxed{X}- , \tag{261}$$

such that optimizing the variational parameters boils down to solving the eigenvalue equation,

$$T_{\text{eff}}(p)X = \omega X, \tag{262}$$

with the effective transfer and normalization matrix defined as

$$2\pi\delta(p - p')(X')^\dagger T_{\text{eff}}(p)X = \langle \Phi_{p'}(X')| T |\Phi_p(X)\rangle \tag{263}$$

$$2\pi\delta(p - p')(X')^\dagger N_{\text{eff}}(p)X = \langle \Phi_{p'}(X')|\Phi_p(X)\rangle . \tag{264}$$

In order to solve this eigenvalue eqation iteratively, we need the action of $T_{\text{eff}}(p)$ on a general input vector $X$. First we compute the tensor $B(X)$, and define the partial contractions

$$\boxed{L_B} = \boxed{F_L} \begin{array}{c} \boxed{B} \\ \boxed{O} \\ \boxed{\bar{A}_L} \end{array} \boxed{(1 - e^{-ip}E_L^L(O))^P} \tag{265}$$

and

$$R_B = \left( (1 - e^{+ip} E_R^R(O))^P \right) \begin{array}{c} B \\ O \\ \bar{A}_R \end{array} F_R \, , \tag{266}$$

where the channel operators are defined as

$$E_L^L(O) = \begin{array}{c} A_L \\ O \\ \bar{A}_L \end{array} \, , \quad \text{and} \quad E_R^R(O) = \begin{array}{c} A_R \\ O \\ \bar{A}_R \end{array} \, . \tag{267}$$

Again – if everything is properly normalized – these operators have a leading eigenvalue equal to one (with $F_L$ and $F_R$ as fixed points), so they should be regularized in order to define the inverses at momentum $p = 0$. The action of $\tilde{T}_{\text{eff}}(p)$ on the tensor $B$ is then given by

$$\tilde{T}_{\text{eff}}(p)B = e^{-ip} \left( L_B \begin{array}{c} A_C \\ O \end{array} F_R \right) + e^{+ip} \left( F_L \begin{array}{c} A_C \\ O \end{array} R_B \right) + \left( F_L \begin{array}{c} B \\ O \end{array} F_R \right) . \tag{268}$$

In the last step, we need the action of $T_{\text{eff}}(p)$ (without the tilde), so we need to perform the last contraction

$$T_{\text{eff}}(p)X = \left( \begin{array}{c} \tilde{T}_{\text{eff}}(p)B \\ \bar{V}_L \end{array} \right) . \tag{269}$$

Upon solving this eigenvalue equation for all momenta, we find the dispersion relation of the transfer matrix. The largest eigenvalue defines the gap of the transfer matrix, and is related to the correlation length of the two-dimensional tensor network [61]. The momentum at which this gap is reached defines the pitch vector of the correlations, and possibly indicates incommensurate correlations in the two-dimensional tensor network.

# 8 Continuous matrix product states

In this section we show that the tangent-space framework for uniform MPS can be extended to the case of continuous field theories. For the sake of simplicity, we explain this in detail for one-component Bose gases, and we work out the explicit equations for the Lieb-Liniger hamiltonian. This set-up can be easily extended – with a large notational overhead – to multi-component gases, fermions [62] and hamiltonians with superconducting terms [63] and exponentially-decaying interactions [64].

Continuous matrix product states were originally introduced [65] as the continuum limit of a particular subset of MPS, chosen so as to obtain a limiting state with valid physical properties.

We approximate the one-dimensional continuum by a chain with lattice spacing $a$ and send $a \to 0$. For simplicity, we restrict to a system containing a single flavor of bosonic particles, i.e. spinless bosons (we refer to e.g. [62] for the more general case). The local basis on each site $n$ on the chain consists of $|0\rangle_n$ (empty site) and $|k\rangle_n = \frac{1}{\sqrt{k!}}(b_n^\dagger)^k |0\rangle_n$ ($k \geq 1$ bosons). In order to obtain a state with a finite density of particles in the continuum limit, the probability of detecting $k$ particles on a site will have to scale as $a^k$. This quickly leads to the following parameterization

$$A^0 = \mathbb{1} + aQ \tag{270}$$

$$A^1 = \sqrt{a}R, \tag{271}$$

where the states corresponding to $k > 1$ are for most purposes irrelevant; the corresponding MPS matrices are completely determined in terms of $A^1$, i.e. in terms of the matrix $R$. If we now take the continuum limit $a \to 0$ and identify the bosonic creation operator on the site with a bosonic field operator as $\hat{\psi}^\dagger(na) = \hat{b}_n/\sqrt{a}$, we obtain (through a Taylor expansion of the path-ordered exponential)

$$|\Psi(Q,R)\rangle = \boldsymbol{v}_L^\dagger \mathrm{Pexp}\left( \int_{-\infty}^{+\infty} \mathrm{d}x \, Q \otimes \mathbb{1} + R \otimes \hat{\psi}^\dagger(x) \right) \boldsymbol{v}_R |\Omega\rangle. \tag{272}$$

An alternative approach to obtain cMPS as a continuous measurement process [66], whereby the physical degrees of freedom correspond to the field that leak out of a zero-dimensional cavity, which plays the role of the ancilla system and thus had $D$ internal levels. This interpretation also has a clear holographic interpretation, which provides on possible avenue towards higher-dimensional generalizations.

## 8.1 Gauge transformations and canonical forms

Just like for MPS, we focus on the case of translation invariant cMPS in the thermodynamic limit throughout these lecture notes, which are described by position-independent (i.e. uniform) matrices $Q$ and $R$. We start by computing the norm of a uniform cMPS, which is determined by the transfer matrix

$$T = Q \otimes \mathbb{1} + \mathbb{1} \otimes \bar{Q} + R \otimes \bar{R}. \tag{273}$$

This expression is related to the MPS transfer matrix as

$$T = \lim_{a \to 0} \frac{E - \mathbb{1}}{a} = \lim_{a \to 0} \frac{1}{a} \log E, \tag{274}$$

and the properties of $T$ can be obtained from this correspondence. In the generic (injective case), the eigenvalue $\lambda_1$ with largest real part is non-degenerate and purely real; its corresponding left and right eigenvector should correspond to positive definite matrices $l$ and $r$. The norm of the cMPS is given by

$$\langle \Psi(\bar{Q},\bar{R})|\Psi(Q,R)\rangle = \left( \boldsymbol{v}_L^\dagger \otimes \bar{\boldsymbol{v}}_L^\dagger \right) \mathrm{Pexp}\left( \int_{-\infty}^{+\infty} \mathrm{d}x \, T \right) \left( \boldsymbol{v}_R \otimes \bar{\boldsymbol{v}}_R \right), \tag{275}$$

which implies that, in order to have a properly normalized cMPS in the thermodynamic limit, the unique eigenvalue $\lambda_1$ of $T$ with largest real part should be zero. If this is not the case, the cMPS needs to be rescaled, which amounts to shifting $Q$ with the identity as $Q \to Q - \frac{\lambda_1}{2}\mathbb{1}$. The corresponding left- and right eigenvectors then obey the equations

$$lQ + Q^\dagger l + R^\dagger l R = 0 \tag{276}$$

$$Qr + rQ^\dagger + RrR^\dagger = 0, \tag{277}$$

and all other eigenvalues of $T$ now have a strictly negative real part. Under these conditions, the (path-ordered) exponential of the transfer matrix[15] reduces to a projector on the fixed points. If we also make sure that the overlap of the boundary vectors $v_L^\dagger$ and $v_R$ with these fixed points are unity, then the uniform cMPS is properly normalized $\langle \Psi(\bar{Q}, \bar{R})|\Psi(Q, R)\rangle = 1$.

The parametrization of the cMPS in terms of matrices $Q$ and $R$ is not unique, because gauge transformations of the form

$$Q \to g^{-1}Qg, \quad R \to g^{-1}Rg \tag{278}$$

leave the cMPS invariant. This gauge freedom in $Q$ and $R$ can be used to find canonical forms. Choosing $g^{-1} = C_L$ where $l = C_L C_L^\dagger$ brings the cMPS matrices into left-canonical form, where the new left fixed point is the unit matrix, i.e.

$$Q_L + Q_L^\dagger + R_L^\dagger R_L = 0. \tag{279}$$

Similarly, using $g = C_R$ where $r = C_R C_R^\dagger$ we obtain the right-canonical form, where the right fixed point is the identity matrix as expressed by

$$Q_R + Q_R^\dagger + R_R R_R^\dagger = 0. \tag{280}$$

Again, we can combine both canonical forms in order to arrive at a mixed canonical form, where an extra matrix $C$ is introduced linking the two

$$|\Psi(Q, R)\rangle = v_L^\dagger \mathrm{Pexp}\left( \int_{-\infty}^{a} dx \, Q_L \otimes \mathbb{1} + R_L \otimes \hat{\psi}^\dagger(x) \right)$$
$$\times C\,\mathrm{Pexp}\left( \int_{a}^{+\infty} dx \, Q_R \otimes \mathbb{1} + R_R \otimes \hat{\psi}^\dagger(x) \right) v_R |\Omega\rangle. \tag{281}$$

By diagonalizing the matrix $C = USV^\dagger$ we arrive at a Schmidt decomposition of the state

$$|\Psi(Q, R)\rangle = \sum_{i=1}^{D} S_i |\Psi_L^i(Q_L, R_L)\rangle \otimes |\Psi_R^i(Q_R, R_R)\rangle, \tag{282}$$

where we have redefined

$$Q_L \to U^\dagger Q_L U, \quad R_L \to U^\dagger R_L U \tag{283}$$
$$Q_R \to V^\dagger Q_R U, \quad R_R \to U^\dagger R_R U. \tag{284}$$

The fidelity between two different normalized cMPS $|\Psi(Q_1, R_1)\rangle$ and $|\Psi(Q_2, R_2)\rangle$ is computed similarly as before. Indeed, the overlap is given by

$$\langle \Psi(\bar{Q}_2, \bar{R}_2)|\Psi(Q_1, R_1)\rangle \propto \exp\left( \int_{-\infty}^{+\infty} dx \, T_{12} \right), \tag{285}$$

with the mixed transfer matrix

$$T_{12} = Q_1 \otimes \mathbb{1} + \mathbb{1} \otimes \bar{Q}_2 + R_1 \otimes \bar{R}_2. \tag{286}$$

The fidelity is determined by the eigenvalue $\lambda$ with largest real part of $T_{12}$, which should have a real part smaller than or equal to zero if both individual cMPS are properly normalized. The total fidelity then corresponds to zero or one respectively, so that it makes more sense to define $\mathrm{Re}\lambda$ itself as the logarithmic fidelity density, or to define

$$f = \exp(\mathrm{Re}\lambda), \tag{287}$$

such that the overlap on a segment of length $l$ scales as $f^l$.

---

[15]In the case of a uniform, i.e. constant, transfer matrix $T$, the path-ordering has no effect.

## 8.2 Evaluating expectation values

After having introduced the class of uniform cMPS, we show how to use them in actual calculations. All expectation values involve field operators, so the first step consists of finding an expression for the action of a field operator on a cMPS. All of the results below are obtained by using the following identity for computing the commutator between a general operator and a path-ordered exponential $\hat{U}(a,b) = \text{Pexp}\{\int_a^b \hat{A}(x)\,dx\}$

$$\left[\hat{O}, \hat{U}(a,b)\right] = \int_a^b \hat{U}(a,x)[\hat{O}, \hat{A}(x)]\hat{U}(x,b)\,dx. \tag{288}$$

Applying this approach to the bosonic field operator $\hat{O} = \hat{\psi}(x)$, and choosing $\hat{A}(x) = Q \otimes \mathbb{1} + R \otimes \hat{\psi}^\dagger(x)$, we obtain

$$\hat{\psi}(x)|\Psi(Q,R)\rangle = \boldsymbol{v}_L^\dagger \hat{U}(-\infty,x)R\hat{U}(x,+\infty)\boldsymbol{v}_R|\Omega\rangle. \tag{289}$$

The expectation value of the field operator is, therefore, given by

$$\langle\Psi(\bar{Q},\bar{R})|\hat{\psi}(x)|\Psi(Q,R)\rangle = (l|R \otimes \mathbb{1}|r) = \text{Tr}(Rrl). \tag{290}$$

Similarly, we find for the expectation value of the density operator

$$\langle\Psi(\bar{Q},\bar{R})|\hat{\psi}^\dagger(x)\hat{\psi}(x)|\Psi(Q,R)\rangle = (l|R \otimes \bar{R}|r) = \text{Tr}(RrR^\dagger l). \tag{291}$$

Acting with a second field operator on the same location just brings down a second matrix $R$, so that we obtain for a contact interaction

$$\langle\Psi(\bar{Q},\bar{R})|\hat{\psi}^\dagger(x)\hat{\psi}^\dagger(x)\hat{\psi}(x)\hat{\psi}(x)|\Psi(Q,R)\rangle = (l|R^2 \otimes \bar{R}^2|r) = \text{Tr}(R^2 r (R^2)^\dagger l). \tag{292}$$

By acting with field operators at different locations, we can compute correlation functions. The field-field correlation function is given by (we assume $x < y$)

$$\langle\Psi(\bar{Q},\bar{R})|\hat{\psi}^\dagger(y)\hat{\psi}(x)|\Psi(Q,R)\rangle = (l|(R \otimes \mathbb{1})\,\text{Pe}^{\int_x^y T(z)dz}\left(\mathbb{1} \otimes \bar{R}\right)|r) \tag{293}$$

$$= (l|(R \otimes \mathbb{1})\,e^{T(y-x)}\left(\mathbb{1} \otimes \bar{R}\right)|r), \tag{294}$$

and the density-density correlation function

$$\langle\Psi(\bar{Q},\bar{R})|\hat{\psi}^\dagger(y)\hat{\psi}(y)\hat{\psi}^\dagger(x)\hat{\psi}(x)|\Psi(Q,R)\rangle = (l|\left(R \otimes \bar{R}\right)e^{T(y-x)}\left(R \otimes \bar{R}\right)|r). \tag{295}$$

These expressions clearly show that a cMPS necessarily exhibits exponential decay of correlations. Indeed, if we split off the fixed-point projector from the transfer matrix (assuming a properly normalized cMPS with $\lambda_1 = 0$), we obtain

$$e^{Tx} = |r)(l| + \sum_{i=2}^{D^2} e^{\lambda_i x}|\lambda_i)(\lambda_i|, \tag{296}$$

so the second eigenvalue $\lambda_2$ (sorted by largest real part) of the transfer matrix determines the correlation length as $\xi = -1/\text{Re}(\lambda_2)$; the imaginary part of the subleading eigenvalues again determine the oscillations in the correlation function [13].

Using $\hat{\psi}^\dagger(p) = \int_{-\infty}^{+\infty} \frac{dx}{\sqrt{2\pi}}\hat{\psi}^\dagger(x)e^{ipx}$, we can compute the correlation function directly in momentum space,

$$\langle\Psi(\bar{Q},\bar{R})|\hat{\psi}^\dagger(p')\hat{\psi}(p)|\Psi(Q,R)\rangle = \delta(p-p')n(p), \tag{297}$$

so that we obtain the momentum distribution function $n(p)$ as

$$n(p) = \int_{-\infty}^{+\infty} dx \, e^{ipx} \langle \Psi(\bar{Q},\bar{R})| \hat{\psi}(x)^{\dagger} \hat{\psi}(0) |\Psi(Q,R)\rangle. \tag{298}$$

Using the above expression for the real-space correlation function, we obtain

$$n(p) = \int_{-\infty}^{0} dx (l| \left( \mathbb{1} \otimes \bar{R} \right) e^{(-T+ip)x} \left( R \otimes \mathbb{1} \right) |r) + \int_{0}^{+\infty} dx (l| (R \otimes \mathbb{1}) e^{(T+ip)x} \left( \mathbb{1} \otimes \bar{R} \right) |r). \tag{299}$$

In order to further work out this expression, we define a regularized transfer matrix by splitting off the fixed point projector of the exponentiated transfer matrix (see Eq. (296)); this allows to compute the integral

$$\int_{0}^{\infty} dx \, e^{(T+ip)x} = \left( \int_{0}^{\infty} dx e^{ipx} \right) |r)(l| - \left( \tilde{T} + ip \right)^{P}, \tag{300}$$

with[16]

$$(\tilde{T} + ip)^{P} = \sum_{i=2}^{D^2} (\lambda_i + ip)^{-1} |\lambda_i)(\lambda_i|, \tag{301}$$

and to compute the momentum distribution function

$$n(p) = 2\pi\delta(p)(l| \left( \mathbb{1} \otimes \bar{R} \right) |r)(l| (R \otimes \mathbb{1}) |r) \\ + (l| \left( \mathbb{1} \otimes \bar{R} \right) \left( -\tilde{T} + ip \right)^{P} (R \otimes \mathbb{1}) |r) + (l| (R \otimes \mathbb{1}) \left( -\tilde{T} - ip \right)^{P} \left( \mathbb{1} \otimes \bar{R} \right) |r). \tag{302}$$

Here, the $\delta$-function contribution signals long-range order, in particular, associated with the condensation of the bosonic particles in the ground state.

More advanced expectation values involve derivatives of field operators. Therefore, we differentiate the above expression for the action of $\hat{\psi}(x)$ on a cMPS [Eq. (289)] with respect to $x$,

$$\frac{d\hat{\psi}(x)}{dx} |\Psi(Q,R)\rangle = \frac{d}{dx} \mathbf{v}_L^{\dagger} \hat{U}(-\infty, x) R \hat{U}(x, +\infty) \mathbf{v}_R |\Omega\rangle. \tag{303}$$

Using the equations

$$\frac{d}{dx} \hat{U}(y,x) = +\hat{U}(y,x)(Q \otimes \mathbb{1} + R \otimes \hat{\psi}^{\dagger}(x)) \tag{304}$$

$$\frac{d}{dx} \hat{U}(x,y) = -(Q \otimes \mathbb{1} + R \otimes \hat{\psi}^{\dagger}(x))\hat{U}(x,y), \tag{305}$$

we obtain

$$\frac{d\hat{\psi}(x)}{dx} |\Psi(Q,R)\rangle = \mathbf{v}_L^{\dagger} \hat{U}(-\infty, x)\Big( (QR - RQ) \otimes \mathbb{1}$$
$$+ (R^2 - R^2) \otimes \hat{\psi}^{\dagger}(x) \Big) \hat{U}(x, +\infty) \mathbf{v}_R |\Omega\rangle \tag{306}$$

$$= \mathbf{v}_L^{\dagger} \hat{U}(-\infty, x) [Q,R] \hat{U}(x, +\infty) \mathbf{v}_R |\Omega\rangle. \tag{307}$$

---

[16]Computing the action of $(\tilde{T} \pm ip)^{P}$ on a vector (to the left or right) efficiently requires to use a iterative linear solver. When 'p=0', nothing needs to be done in principle to eliminate the contribution of the zero eigenvalue, as any contribution that would be generated is immediately killed by acting with the operator upon constructing the Krylov subspace. In the more general case, it is useful to explicitly project out any contribution in the subspace of eigenvalue zero, using $\mathbb{1} - |r)(l|$.

In this expression the cancellation of the term with a creation operator is automatically obeyed, but this is not the case for cMPS with multiple species of bosons and/or fermions; in the more general case, additional regularity conditions on the $R$ matrices have to imposed in order to obtain a finite kinetic energy (see Ref. [62]). The expectation value of a kinetic energy density term is given by

$$\langle\Psi(\bar{Q},\bar{R})|\frac{\mathrm{d}\hat{\psi}^{\dagger}(x)}{\mathrm{d}x}\frac{\mathrm{d}\hat{\psi}(x)}{\mathrm{d}x}|\Psi(Q,R)\rangle = (l|[Q,R]\otimes[\bar{Q},\bar{R}]|r). \tag{308}$$

### 8.3 Tangent vectors

We introduce a tangent vector in the uniform gauge as

$$
\begin{aligned}
|\Phi(V,W;Q,R)\rangle &= \int\mathrm{d}x\sum_{\alpha,\beta}\left(V_{\alpha,\beta}(x)\frac{\partial}{\partial Q_{\alpha,\beta}(x)}+W_{\alpha,\beta}(x)\frac{\partial}{\partial R_{\alpha,\beta}(x)}\right)|\Psi(Q,R)\rangle \\
&= \int\mathrm{d}x\,\boldsymbol{v}_{L}^{\dagger}\hat{U}(-\infty,x)\left(V\otimes\mathbb{1}+W\otimes\hat{\psi}^{\dagger}(x)\right)\hat{U}(x,+\infty)\boldsymbol{v}_{R}\,|\Omega\rangle,
\end{aligned}
\tag{309}
$$

where we have again used the notation

$$\hat{U}(a,b) = \mathrm{Pexp}\left(\int_{a}^{b}\mathrm{d}x\,Q\otimes\mathbb{1}+R\otimes\hat{\psi}^{\dagger}(x)\right). \tag{310}$$

For finding a proper parametrization of the tangent space, we first compute the overlap between two tangent vectors,

$$
\begin{aligned}
&\langle\Phi(\bar{V}',\bar{W}')|\Phi(V,W)\rangle \\
&= \int_{-\infty}^{+\infty}\mathrm{d}x\int_{x}^{+\infty}\mathrm{d}y\,(l|\left(V\otimes\mathbb{1}+W\otimes\bar{R}\right)e^{(y-x)T}\left(\mathbb{1}\otimes\bar{V}'+R\otimes\bar{W}'\right)|r) \\
&\quad + \int_{-\infty}^{+\infty}\mathrm{d}x\int_{-\infty}^{x}\mathrm{d}y\,(l|\left(\mathbb{1}\otimes\bar{V}'+R\otimes\bar{W}'\right)e^{(x-y)T}\left(V\otimes\mathbb{1}+W\otimes\bar{R}\right)|r) \\
&\quad + \int_{-\infty}^{+\infty}\mathrm{d}x\,(l|W\otimes\bar{W}|r).
\end{aligned}
\tag{311}
$$

This expression is further worked out using the above inversion of the transfer matrix.

$$
\begin{aligned}
\langle\Phi(\bar{V}',\bar{W}')|\Phi(V,W)\rangle = 2\pi\delta(0)\Big[&(l|W\otimes\bar{W}|r) \\
&+ (l|\left(V\otimes\mathbb{1}+W\otimes\bar{R}\right)(-\tilde{T})^{-1}\left(\mathbb{1}\otimes\bar{V}'+R\otimes\bar{W}\right)|r) \\
&+ (l|\left(\mathbb{1}\otimes\bar{V}'+R\otimes\bar{W}\right)(-\tilde{T})^{-1}\left(V\otimes\mathbb{1}+W\otimes\bar{R}\right)|r) \\
&+ 2\pi\delta(0)(l|\left(V\otimes\mathbb{1}+W\otimes\bar{R}\right)|r)(l|\left(\mathbb{1}\otimes\bar{V}'+R\otimes\bar{W}\right)|r)\Big].
\end{aligned}
\tag{312}
$$

The diverging prefactor corresponds to the infinite system size and originates from the fact that the tangent vectors represent momentum zero plane waves, that cannot be normalized to one. The additional divergence in the square brackets can be traced back to the possible overlap with the ground state, and vanishes if orthogonality to the ground state is enforced as

$$\langle\Psi(\bar{Q},\bar{R})|\Phi(V,W)\rangle = 2\pi\delta(0)(l|\left(V\otimes\mathbb{1}+W\otimes\bar{R}\right)|r) = 0. \tag{313}$$

The gauge freedom in the cMPS parametrization induces a redundancy in the parametrization of the states $|\Phi_p(V,W)\rangle$, i.e. these states are invariant under the additive gauge transformation $V \leftarrow V + Q_L X - X Q_R + ipX$ and $W \leftarrow W + R_L X - X R_R$ for an arbitrary matrix $X$. This gauge freedom can be used to choose a parametrization that allows us to omit the non-local terms in the expressions above, e.g. by restricting to representations $(V,W)$ that satisfy

$$(l|\left(V \otimes \mathbb{1} + W \otimes \bar{R}_1\right) = 0 \Leftrightarrow V = -l^{-1}R^\dagger l W. \tag{314}$$

This condition is henceforth referred to as the *left gauge condition*; it is typically used in combination with a left canonical choice for $Q$ and $R$ such that $l = \mathbb{1}$ and we simply have $V = -R^\dagger W$. Similarly one can choose instead a right gauge condition $V = -W r R^\dagger r^{-1}$, which simplifies in the case of a right canonical cMPS with $r = \mathbb{1}$.

Before proceeding, we generalize the definition of tangent vectors to

$$|\Phi_p(V,W;Q_1,R_1,Q_2,R_2)\rangle$$
$$= \int dx\, e^{ipx} \boldsymbol{v}_L^\dagger \hat{U}_1(-\infty,x)\left(V \otimes \mathbb{1} + W \otimes \hat{\psi}^\dagger(x)\right)\hat{U}_2(x,+\infty)\boldsymbol{v}_R |\Omega\rangle, \tag{315}$$

which contain a boost so as to represent a momentum eigenstate with momentum $p$, and where $\hat{U}_1$ and $\hat{U}_2$ are defined in terms of two different pairs of matrices $Q_1, R_1$ and $Q_2, R_2$, respectively. We can work in a mixed gauge by using $Q_1 = Q_L, R_1 = R_L$ and $Q_2 = Q_R, R_2 = R_R$, or even $Q_2 = \tilde{Q}_R$ and $R_2 = \tilde{R}_R$ when there is a second ground state available and we want to target a non-trivial topological sector. Still using the parameterization $V = -l_1^{-1}R^\dagger l_1 W$ with $l_1$ the left fixed point of the transfer matrix of $Q_1$ and $R_1$, we obtain the local expression

$$\langle\Phi_{p'}(\bar{V}',\bar{W}')|\Phi_p(V,W)\rangle = 2\pi\delta(p-p')(l_1|W \otimes \bar{W}'|r_2), \tag{316}$$

where $r_2$ is the right fixed point of the transfer matrix of $Q_2, R_2$.

For both the time-dependent variational principle and for the quasiparticle ansatz, it is useful to know how an annihilation operator acts on a tangent vector,

$$\hat{\psi}(y)|\Phi_p(V,W)\rangle$$
$$= \int_y^{+\infty} dx\, e^{ipx} \boldsymbol{v}_L^\dagger \hat{U}_1(-\infty,y)R_1\hat{U}_1(y,x)\left(V \otimes \mathbb{1} + W \otimes \hat{\psi}^\dagger(x)\right)\hat{U}_2(x,+\infty)\boldsymbol{v}_R |\Omega\rangle$$
$$+ \int_{-\infty}^y dx\, e^{ipx} \boldsymbol{v}_L^\dagger \hat{U}_1(-\infty,x)\left(V \otimes \mathbb{1} + W \otimes \hat{\psi}^\dagger(x)\right)\hat{U}_2(x,y)R_2\hat{U}_2(y,+\infty)\boldsymbol{v}_R |\Omega\rangle$$
$$+ e^{ipy} \boldsymbol{v}_L^\dagger \hat{U}_1(-\infty,y)W\hat{U}_2(y,+\infty)\boldsymbol{v}_R |\Omega\rangle. \tag{317}$$

The same can be done for two annihilation operators $\hat{\psi}(z), \hat{\psi}(y)$ where we assume $z \leq y$,

$$\hat{\psi}(z)\hat{\psi}(y)|\Phi_p(V,W)\rangle$$
$$= \int_y^{+\infty} dx\, e^{ipx} \boldsymbol{v}_L^\dagger \hat{U}_1(-\infty,z)R_1\hat{U}_1(z,y)R_1\hat{U}_1(y,x)\left(V \otimes \mathbb{1} + W \otimes \hat{\psi}^\dagger(x)\right)\hat{U}_2(x,+\infty)\boldsymbol{v}_R |\Omega\rangle$$
$$+ \int_z^y dx\, e^{ipx} \boldsymbol{v}_L^\dagger \hat{U}_1(-\infty,z)R_1\hat{U}_1(z,x)\left(V \otimes \mathbb{1} + W \otimes \hat{\psi}^\dagger(x)\right)\hat{U}_2(x,y)R_2\hat{U}_2(y,+\infty)\boldsymbol{v}_R |\Omega\rangle$$
$$+ \int_{-\infty}^z dx\, e^{ipx} \boldsymbol{v}_L^\dagger \hat{U}_1(-\infty,x)\left(V \otimes \mathbb{1} + W \otimes \hat{\psi}^\dagger(x)\right)\hat{U}_2(x,z)R_2\hat{U}_2(z,y)R_2\hat{U}_2(y,+\infty)\boldsymbol{v}_R |\Omega\rangle$$
$$+ e^{ipz} \boldsymbol{v}_L^\dagger \hat{U}_1(-\infty,z)W\hat{U}_2(z,y)R_2\hat{U}_2(y,\infty)\boldsymbol{v}_R |\Omega\rangle$$
$$+ e^{ipy} \boldsymbol{v}_L^\dagger \hat{U}_1(-\infty,z)R_1\hat{U}_1(z,y)W_2\hat{U}_2(y,+\infty)\boldsymbol{v}_R |\Omega\rangle. \tag{318}$$

For the kinetic energy we need to know how $\frac{d\hat{\psi}(y)}{dy}$ acts on a tangent vector, i.e. we have to take the derivative of the above equation

$$
\begin{aligned}
\frac{d\hat{\psi}(y)}{dy} &|\Phi_p(V,W)\rangle \\
&= \int_y^{+\infty} dx \, e^{ipx} \boldsymbol{v}_L^\dagger \hat{U}_1(-\infty,y)[Q_1,R_1]\hat{U}_1(y,x)\Big(V\otimes\mathbb{1}+W\otimes\hat{\psi}^\dagger(x)\Big)\hat{U}_2(x,+\infty)\boldsymbol{v}_R|\Omega\rangle \\
&\quad + \int_{-\infty}^y dx \, e^{ipx}\boldsymbol{v}_L^\dagger\hat{U}_1(-\infty,x)\Big(V\otimes\mathbb{1}+W\otimes\hat{\psi}^\dagger(x)\Big)\hat{U}_2(x,y)[Q_2,R_2]\hat{U}_2(y,+\infty)\boldsymbol{v}_R|\Omega\rangle \\
&\quad + e^{ipy}\boldsymbol{v}_L^\dagger\hat{U}_1(-\infty,y)\Big([V,R]+[Q,W]+ipW\Big)\hat{U}_2(y,+\infty)\boldsymbol{v}_R|\Omega\rangle.
\end{aligned} \tag{319}
$$

One can check that a number of potentially problematic (infinite norm) terms which have a creation operator $\hat{\psi}^\dagger(y)$ at the fixed position $y$ all nicely cancel.

## 8.4 Ground-state optimization and time-dependent variational principle

The time-dependent varational principle for cMPS can be obtained in a similar way as for MPS. Again, we restrict to uniform cMPS and translation invariant hamiltonians; we refer to Ref. [67] for the more general case.

Starting from a uniform cMPS with time-dependent matrices $Q(t)$ and $R(t)$, we obtain for the left-hand side of the Schrödinger equation

$$
i\frac{d\,|\Psi(Q,R)\rangle}{dt} = |\Phi(i\dot{Q},i\dot{R};Q,R)\rangle, \tag{320}
$$

i.e. a tangent vector (momentum zero) with $V=i\dot{Q}$ and $W=i\dot{R}$. The TDVP prescribes to choose $\dot{Q}$ and $\dot{R}$ such that $\||\Phi(i\dot{Q},i\dot{R};Q,R)\rangle-H|\Psi(Q,R)\rangle\|^2$ is minimized. Let us first compute the general overlap $\langle\Phi(V,W;Q,R)|\hat{H}|\Psi(Q,R)\rangle$, where we take as an example the Lieb-Liniger hamiltonian

$$
\hat{H} = \int_{-\infty}^{+\infty} dx \left\{ \frac{d\hat{\psi}^\dagger(x)}{dx}\frac{d\hat{\psi}(x)}{dx} - \mu\hat{\psi}^\dagger(x)\hat{\psi}(x) + g\hat{\psi}^\dagger(x)^2\hat{\psi}(x)^2 \right\}. \tag{321}
$$

We define the quantities

$$
(L_h| = (l|\left\{[Q,R]\otimes[\bar{Q},\bar{R}]-\mu R\otimes\bar{R}+gR^2\otimes\bar{R}^2\right\}(-\tilde{T})^P, \tag{322}
$$
$$
|R_h) = (-\tilde{T})^P\left\{[Q,R]\otimes[\bar{Q},\bar{R}]-\mu R\otimes\bar{R}+gR^2\otimes\bar{R}^2\right\}|r), \tag{323}
$$

which play a similar role as the equally named quantities in the MPS case [Eq. (115)]. Assuming that $\langle\Phi(V,W)|\Psi(Q,R)\rangle \propto (l|\mathbb{1}\otimes\bar{V}+R\otimes\bar{W}|r) = 0$, we now obtain

$$
\begin{aligned}
\langle\Phi(V,W)|&\hat{H}|\Psi(Q,R)\rangle = \\
&2\pi\delta(0)\Big[(l|\left\{[Q,R]\otimes([\bar{V},\bar{R}]+[\bar{Q},\bar{W}])-\mu R\otimes\bar{W}+gR^2\otimes(\bar{W}\bar{R}+\bar{R}\bar{W})\right\}|r) \\
&\quad + (l|\left\{\mathbb{1}\otimes\bar{V}+R\otimes\bar{W}\right\}|R_h) + (L_h|\left\{\mathbb{1}\otimes\bar{V}+R\otimes\bar{W}\right\}|r)\Big].
\end{aligned} \tag{324}
$$

To solve the optimization problem

$$
\begin{aligned}
\min_{V,W}\||\Phi(V,W)\rangle &- \hat{H}|\Psi(Q,R)\rangle\|^2 = \\
&\min_{V,W}\Big(\langle\Phi(V,W)|\Phi(V,W)\rangle - \langle\Phi(V,W)|\hat{H}|\Psi(Q,R)\rangle - \langle\Psi(Q,R)|\hat{H}|\Psi(V,W)\rangle + \text{constant}\Big)
\end{aligned}
$$

we also need $\langle\Phi(V,W)|\Phi(V,W)\rangle$. We now exploit the gauge invariance in the cMPS manifold, which enables us to choose $V = -l^{-1}R^\dagger lW$, which simplifies the latter (and also makes the first term on the second line of Eq. (324) vanish. The minimum is then obtained by setting the derivative with respect to $\bar{W}$ equal to zero, resulting in

$$lWr = Q^\dagger l[Q,R]r - l[Q,R]rQ^\dagger - lR[Q,R]rR^\dagger + lRl^{-1}R^\dagger l[Q,R]r - \mu lRr$$
$$+ glR^2 rR^\dagger + gR^\dagger lR^2 r + L_h Rr - lRl^{-1}L_h r \quad (325)$$

or thus, by also using the defining equations of $l$ and $r$,

$$i\dot{R} = W = -[Q,[Q,R]] - \mu R + l^{-1}R^\dagger l(gR^2 + [Q,R])$$
$$+ (gR^2 + [Q,R])rR^\dagger r^{-1} + [R, l^{-1}R^\dagger l[Q,R] - l^{-1}L_h] \quad (326)$$
$$i\dot{Q} = V = -l^{-1}R^\dagger lW. \quad (327)$$

Note that, when the cMPS is itself in the left canonical gauge $Q_L + Q_L^\dagger + R_L^\dagger R_L = 0$ and $l = \mathbb{1}$, we can parameterise $Q_L = iK_L - 1/2R_L^\dagger R_L$, with $K_L$ a Hermitian matrix. The time derivative $\dot{Q}_L = -R_L^\dagger \dot{R}_L$ is compatible with preserving this canonical form at all times, and can be cast into a direct equation for the time derivative of $\dot{K}_L$ as

$$-\dot{K}_L = \frac{1}{2}(\dot{R}_L^\dagger R_L - R_L^\dagger \dot{R}_L). \quad (328)$$

These first-order coupled differential equations can then be solved using standard ODE solvers. By replacing $t \to -i\tau$, we can evolve in imaginary time and obtain an algorithm to converge a random cMPS to the ground state, as was first used in Ref. [68]. Indeed, as in the MPS case, the right hand side of the TDVP equation is essentially the tangent-space gradient, and as such imaginary-time evolution effectively amounts to a continuous gradient descent.

Let us now, in the spirit of established MPS algorithms, try to formulate a mixed gauge approach. The starting point is to approximate $H|\Psi\rangle$ using the more general formulation of tangent vectors

$$|\Phi(V,W;Q_L,R_L,Q_R,R_R)\rangle = |\Phi_0(V,W;Q_L,R_L,Q_R,R_R)\rangle,$$

where the left canonical matrices $Q_L, R_L$ and the right canonical matrices $Q_R, R_R$ are related by a gauge transform $C$. Let us now also define $Q_C = Q_L C = CQ_R$ and $R_C = R_L C = CR_R$.

Furthermore, we redefine

$$(L_h| = (l|\left\{[Q_L,R_L]\otimes[\bar{Q}_L,\bar{R}_L] - \mu R_L\otimes\bar{R}_L + gR_L^2\otimes\bar{R}_L^2\right\}(-\tilde{T}_L^L)^P, \quad (329)$$
$$|R_h) = (-\tilde{T}_R^R)^P\left\{[Q_R,R_R]\otimes[\bar{Q}_R,\bar{R}_R] - \mu R_R\otimes\bar{R}_R + gR_R^2\otimes\bar{R}_R^2\right\}|r), \quad (330)$$

with $T_L^L = Q_L\otimes\mathbb{1} + \mathbb{1}\otimes\bar{Q}_L + R_L\otimes\bar{R}_L$ and similarly for $T_R^R$. We now define $F(V,W)$ as

$$\langle\Phi(V,W;Q_L,R_L,Q_R,R_R)|\hat{H}|\Psi(Q,R)\rangle = 2\pi\delta(0)F(V,W) \quad (331)$$

and find

$$F(V,W) = (\mathbb{1}|\left\{(Q_L R_C - R_L Q_C)\otimes(\bar{Q}_L\bar{W} - \bar{R}_L\bar{V}) + (Q_C R_R - R_C Q_R)\otimes(\bar{V}\bar{R}_R - \bar{W}\bar{Q}_R)\right.$$
$$\left. - \mu R_C\otimes\bar{W} + g(R_L R_C)\otimes(\bar{R}_L\bar{W}) + g(R_C R_R)\otimes(\bar{W}\bar{R}_R)\right\}|\mathbb{1})$$
$$+ (\mathbb{1}|\left\{C\otimes\bar{V} + R_C\otimes\bar{W}\right\}|R_h) + (L_h|\left\{C\otimes\bar{V} + R_C\otimes\bar{W}\right\}|\mathbb{1}). \quad (332)$$

Because of the gauge transformation, there are many equivalent ways of writing this expression. We have chosen to position $C$ (or $R_C$ or $Q_C$) in the cMPS ket state in such a way that it coincides

with the position of $V$ and $W$ in the bra state. Gauge invariance still enables us to choose a gauge condition for $V$ and $W$, which could now be $(\mathbb{1}|(C \otimes \bar{V} + R_C \otimes \bar{W}) = 0$ or $(C \otimes \bar{V} + R_C \otimes \bar{W})|\mathbb{1}) = 0$, i.e. $V = -R_L^{\dagger} W$ or $V = -W R_R^{\dagger}$, both of which make $\||\Phi(V, W; Q_L, R_L, Q_R, R_R)\rangle\|^2 = 2\pi\delta(0)\mathrm{Tr}(W W^{\dagger})$. We then obtain

$$W = \frac{\partial F(V, W)}{\partial \bar{W}} - R_L \frac{\partial F(V, W)}{\partial \bar{V}} \tag{333}$$

$$V = \left( -R_L^{\dagger} \frac{\partial F(V, W)}{\partial \bar{W}} - Q_L^{\dagger} \frac{\partial F(V, W)}{\partial \bar{V}} \right) - Q_L \frac{\partial F(V, W)}{\partial \bar{V}} \tag{334}$$

or

$$W = \frac{\partial F(V, W)}{\partial \bar{W}} - \frac{\partial F(V, W)}{\partial \bar{V}} R_R \tag{335}$$

$$V = \left( -\frac{\partial F(V, W)}{\partial \bar{W}} R_R^{\dagger} - \frac{\partial F(V, W)}{\partial \bar{V}} Q_R^{\dagger} \right) - \frac{\partial F(V, W)}{\partial \bar{V}} Q_R, \tag{336}$$

where

$$\frac{\partial F(V, W)}{\partial \bar{W}} = Q_L^{\dagger}(Q_L R_C - R_L Q_C) - (Q_C R_R - R_C Q_R) Q_R^{\dagger} - \mu R_C$$
$$+ g R_L^{\dagger} R_L R_C + g R_C R_R R_R^{\dagger} + R_C R_h + L_h R_c, \tag{337}$$

$$\frac{\partial F(V, W)}{\partial \bar{V}} = -R_L^{\dagger}(Q_L R_C - R_L Q_C) + (Q_C R_R - R_C Q_R) R_R^{\dagger} + C R_h + L_h C. \tag{338}$$

Now we need to relate $V$ and $W$ to an update of the cMPS matrices. In the mixed gauge representation of tangent vectors, we can identify

$$-iW = i\dot{R}_L C = \dot{R}_C - R_L \dot{C} \tag{339}$$

$$-iV = \dot{Q}_L C = \dot{Q}_C - Q_L \dot{C} \tag{340}$$

or

$$-iW = C\dot{R}_R = \dot{R}_C - \dot{C} R_R \tag{341}$$

$$-iV = C\dot{Q}_R = \dot{Q}_C - \dot{C} Q_R. \tag{342}$$

It thus makes sense to identify

$$i\dot{C} = \frac{\partial F(V, W)}{\partial \bar{V}} \tag{343}$$

$$i\dot{R}_C = \frac{\partial F(V, W)}{\partial \bar{W}} \tag{344}$$

$$i\dot{Q}_C = -R_L^{\dagger} \frac{\partial F(V, W)}{\partial \bar{W}} - Q_L^{\dagger} \frac{\partial F(V, W)}{\partial \bar{V}} = -\frac{\partial F(V, W)}{\partial \bar{W}} R_R^{\dagger} - \frac{\partial F(V, W)}{\partial \bar{V}} Q_R^{\dagger} \tag{345}$$

because of the final identity, which can easily be verified using the definitions of the various quantities involved. The final equations then become

$$i\dot{C} = -R_L^{\dagger}(Q_L R_C - R_L Q_C) + (Q_C R_R - R_C Q_R) R_R^{\dagger} + C R_h + L_h C \tag{346}$$

$$i\dot{R}_C = +Q_L^{\dagger}(Q_L R_C - R_L Q_C) - (Q_C R_R - R_C Q_R) Q_R^{\dagger} - \mu R_C$$
$$+ g R_L^{\dagger} R_L R_C + g R_C R_R R_R^{\dagger} + R_C R_h + L_h R_c, \tag{347}$$

$$i\dot{Q}_C = R_L^{\dagger}(Q_L C R_R - R_L C Q_R) Q_R^{\dagger} - Q_L^{\dagger}(Q_L C R_R - R_L C Q_R) R_R^{\dagger}$$
$$- g R_L^{\dagger} R_L C R_R R_R^{\dagger} + Q_C R_h + L_h Q_C, \tag{348}$$

where, in the last equation, we used some more algebra (substituting definitions). These equations can then be integrated for a small time step, after which a new $Q_L$ and $R_L$ (and corresponding $Q_R$ and $R_R$) need to be extracted from the updated $C$, $R_C$ and $Q_C$.

For finding the best cMPS ground state approximation of a given Hamiltonian, we can evolve according to these equations in imaginary time, i.e. setting $t \to -i\tau$. Indeed, a variational optimum is characterized by the right hand side of the above equations becoming zero. Nonetheless, imaginary time evolution is not necessarily the fastest way to approach the variational optimum, in particular for systems near or at criticality. In the case of MPS, the VUMPS algorithm can be understood as being obtained from imaginary time TDVP by promoting the evolution equations to eigenvalue equations for the center site, and then taking bigger steps corresponding to the replacing the center site by the lowest eigenvector of that effective hamiltonian. In the case of cMPS, this is less clear. Indeed, because the cMPS ansatz is not simply a multilinear functional of the different $Q(x)$ and $R(x)$, it cannot be expected that such an interpretation as eigenvalue problem exists. An alternative approach that starts from the center site point of view was proposed and investigated in Ref. [69], and was found to work quite well.

## 8.5  Quasiparticle ansatz

Finally, we can also apply the MPS quasiparticle ansatz to the continuous field-theory setting, as was first explored in Ref. [63]. Indeed, we have already provided a generalized definition for a "boosted" tangent vector $|\Phi_p(V, W; Q_1, R_1, Q_2, R_2)\rangle$ with good momentum quantum number $p$ in Eq. (315). For a topologically trivial excitation, we will use the mixed gauge by setting $Q_1 = Q_L$, $R_1 = R_L$ and $Q_2 = Q_R$, $R_2 = R_R$. In case of symmetry breaking, we can construct domain wall excitations by using the (right canonical) cMPS matrices of a different ground state for $Q_2$ and $R_2$. For simplicity, we restrict to the topologically trivial case below, though the topologically non-trivial case is completely analogous.

We still have gauge freedom $V \to V + Q_L X - X Q_R + \mathrm{i} p X$ and $W \to W + R_L X - X R_R$[17], which we use to parameterize $V = -R_L^\dagger W$ (or alternatively $V = -W R_R^\dagger$). The overlap between the two ansatz wavefunctions is then given by

$$\langle \Phi_{p'}(V', W')|\Phi_p(V, W)\rangle = 2\pi\delta(p - p')(\mathbb{1}|W \otimes \bar{W}'|\mathbb{1}) \tag{349}$$

$$= 2\pi\delta(p - p')\mathrm{Tr}(W(W')^\dagger). \tag{350}$$

The physical norm reduces to the Euclidean norm on the parameters in $W$. Furthermore, the gauge condition ensures that the excited state is always orthogonal to the ground state. This implies that the variational optimization of the ansatz wavefunction,

$$\min_{W} \frac{\langle \Phi_p(V, W)|H|\Phi_p(V, W)\rangle}{\langle \Phi_p(V, W)|\Phi_p(V, W)\rangle} \tag{351}$$

reduces to an ordinary eigenvalue problem

$$H_{\mathrm{eff}}(p)\boldsymbol{W} = \omega(p)\boldsymbol{W}, \tag{352}$$

where we have defined the effective hamiltonian matrix as

$$\langle \Phi_{p'}(V', W')|H - e|\Phi_p(V, W)\rangle = 2\pi\delta(p - p')(\boldsymbol{W'})^\dagger H_{\mathrm{eff}}(p)\boldsymbol{W}. \tag{353}$$

---

[17]This can be verified by noting that the choice $V = Q_L X - X Q_R + \mathrm{i} p X$ and $W = R_L X - X R_R$ corresponds to

$$|\Phi_p(V, W; Q_L, R_L, Q_R, R_R)\rangle = \int dx \frac{d}{dx}\left(\mathrm{e}^{\mathrm{i} p x}\boldsymbol{v}_L^\dagger \hat{U}_L(-\infty, x) X \hat{U}_R(x, +\infty)\boldsymbol{v}_R|\Omega\rangle\right),$$

where upon integration the contribution of $X$ at $+\infty$ or $-\infty$ is irrelevant and both terms therefore cancel.

In order to implement this eigenvalue problem, we need to find an expression for the expectation value of the hamiltonian, for which we can use the expressions derived in Sec. 8.3. Again, we restrict ourselves to the terms in the Lieb-Liniger hamiltonian. The expectation value of the density operator is given as

$$
\int_{-\infty}^{+\infty} dx \, \langle \Phi_{p'}(\bar{V}', \bar{W}')| \hat{\psi}^\dagger(x) \hat{\psi}(x) |\Phi_p(V, W)\rangle = 2\pi\delta(p - p')\times
$$

$$
(\mathbb{1}|\Big[ R_L \otimes \bar{R}_L \Big(-T_L^L\Big)^P W \otimes \bar{W}' + W \otimes \bar{W}' \Big(-T_R^R\Big)^P R_R \otimes \bar{R}_R
$$

$$
+ R_L \otimes \bar{R}_L \Big(-T_L^L\Big)^P \Big(V \otimes \mathbb{1} + W \otimes \bar{R}_L\Big)\Big(-T_L^R + ip\Big)^P \Big(\mathbb{1} \otimes \bar{V}' + R_R \otimes \bar{W}'\Big)
$$

$$
+ R_L \otimes \bar{R}_1 \Big(-T_L^L\Big)^P \Big(\mathbb{1} \otimes \bar{V}' + R_L \otimes \bar{W}'\Big)\Big(-T_R^L - ip\Big)^P \Big(V \otimes \mathbb{1} + W \otimes \bar{R}_R\Big)
$$

$$
+ R_L \otimes \bar{W}' \Big(-T_R^L - ip\Big)^P \Big(V \otimes \mathbb{1} + W \otimes \bar{R}_R\Big)
$$

$$
+ W \otimes \bar{R}_L \Big(-T_L^R + ip\Big)^P \Big(\mathbb{1} \otimes \bar{V}' + R_R \otimes \bar{W}'\Big) + W \otimes \bar{W}'\Big]|\mathbb{1}), \tag{354}
$$

whereas the interaction energy is

$$
\int_{-\infty}^{+\infty} dx \, \langle \Phi_{p'}(\bar{V}', \bar{W}')| \hat{\psi}^\dagger(x) \hat{\psi}^\dagger(x) \hat{\psi}(x) \hat{\psi}(x) |\Phi_p(V, W)\rangle = 2\pi\delta(p - p')\times
$$

$$
(\mathbb{1}|\Big[ R_L^2 \otimes \bar{R}_L^2 \Big(-T_L^L\Big)^P W \otimes \bar{W}' + W \otimes \bar{W}' \Big(-T_R^R\Big)^P R_R^2 \otimes \bar{R}_R^2
$$

$$
+ R_L^2 \otimes \bar{R}_L^2 \Big(-T_L^L\Big)^P \Big(V \otimes \mathbb{1} + W \otimes \bar{R}_1\Big)\Big(-T_L^R + ip\Big)^P \Big(\mathbb{1} \otimes \bar{V}' + R_R \otimes \bar{W}'\Big)
$$

$$
+ R_L^2 \otimes \bar{R}_L^2 \Big(-T_L^L\Big)^P \Big(\mathbb{1} \otimes \bar{V}' + R_L \otimes \bar{W}'\Big)\Big(-T_R^L - ip\Big)^P \Big(V \otimes \mathbb{1} + W \otimes \bar{R}_2\Big)
$$

$$
+ R_L^2 \otimes (\bar{R}_L \bar{W}' + \bar{W}' \bar{R}_R)\Big(-T_R^L - ip\Big)^P \Big(V \otimes \mathbb{1} + W \otimes \bar{R}_R\Big)
$$

$$
+ (R_L W + W R_R) \otimes \bar{R}_L^2 \Big(-T_L^R + ip\Big)^P \Big(\mathbb{1} \otimes \bar{V}' + R_R \otimes \bar{W}'\Big)
$$

$$
+ (R_L W + W R_R) \otimes (\bar{R}_L \bar{W}' + \bar{W}' \bar{R}_R)\Big]|\mathbb{1}), \tag{355}
$$

and the kinetic energy term

$$\int_{-\infty}^{+\infty} dx \, \langle \Phi_{p'}(\bar{V}', \bar{W}')| \frac{d\hat{\psi}^\dagger(x)}{dx} \frac{d\hat{\psi}(x)}{dx} |\Phi_p(V, W)\rangle = 2\pi\delta(p - p') \times$$

$$(\mathbb{1}| \Bigg[ \Big( [Q_L, R_L] \otimes [\bar{Q}_L, \bar{R}_L] \Big) \Big( -T_L^L \Big)^P \Big( W \otimes \bar{W}' \Big)$$

$$+ \Big( W \otimes \bar{W}' \Big) \Big( -T_R^R \Big)^P \Big( [Q_R, R_R] \otimes [\bar{Q}_R, \bar{R}_R] \Big)$$

$$+ \Big( [Q_L, R_L] \otimes [\bar{Q}_L, \bar{R}_L] \Big) \Big( -T_L^L \Big)^P$$
$$\times \Big( V \otimes \mathbb{1} + W \otimes \bar{R}_L \Big) \Big( -T_L^R + ip \Big)^P \Big( \mathbb{1} \otimes \bar{V}' + R_R \otimes \bar{W}' \Big)$$

$$+ \Big( [Q_L, R_L] \otimes [\bar{Q}_L, \bar{R}_L] \Big) \Big( -T_L^L \Big)^P$$
$$\times \Big( \mathbb{1} \otimes \bar{V}' + R_L \otimes \bar{W}' \Big) \Big( -T_R^L - ip \Big)^P \Big( V \otimes \mathbb{1} + W \otimes \bar{R}_R \Big)$$

$$+ \Big( ((Q_L W - W Q_R) + (V R_R - R_L V) + ip W) \otimes [\bar{Q}_L, \bar{R}_L] \Big)$$
$$\Big( -T_L^R + ip \Big)^P \Big( \mathbb{1} \otimes \bar{V}' + R_R \otimes \bar{W}' \Big)$$

$$+ \Big( [Q_L, R_L] \otimes ((\bar{Q}_L \bar{W}' - \bar{W}' \bar{Q}_R) + (\bar{V}' \bar{R}_R - \bar{R}_L \bar{V}') - ip\bar{W}') \Big)$$
$$\times \Big( -T_R^L - ip \Big)^P \Big( V \otimes \mathbb{1} + W \otimes \bar{R}_R \Big)$$

$$+ \Big( (Q_L W - W Q_R) + (V R_R - R_L V) + ip W \Big)$$
$$\otimes \Big( (\bar{Q}_L \bar{W}' - \bar{W}' \bar{Q}_R) + (\bar{V}' \bar{R}_R - \bar{R}_L \bar{V}') - ip\bar{W}' \Big) \Bigg] |\mathbb{1}). \qquad (356)$$

Here, one always needs to insert $V = -R_L^\dagger W$. Furthermore, we have defined the mixed transfer matrices $T_R^L = Q_L \otimes \mathbb{1} + \mathbb{1} \otimes \bar{Q}_R + R_L \otimes \bar{R}_R$ and vice versa for $T_L^R$, on top of the transfer matrices $T_L^L$ and $T_R^R$ that we defined in the previous section. Note that for all of these, the left and right eigenvectors of zero eigenvalue are easy combinations of $\mathbb{1}$, $C$ and $C^\dagger$: $T_L^L$ has left eigenvector $\mathbb{1}$ and right eigenvector $CC^\dagger$, for $T_R^R$ we have $C^\dagger C$ and $\mathbb{1}$ as left and right eigenvector. $T_L^R$ has $C$ and $C^\dagger$ as left and right eigenvector, whereas $T_R^L$ has $C^\dagger$ and $C$ as left and right eigenvector. These are needed to compute the "pseudo-inverses" using an iterative linear solver, as explained above.

# 9 Outlook

In these lecture notes we have explained the most important tangent-space methods for uniform matrix product states in full detail. Yet, this picture is far from complete, and many new applications are still to be expected in the near future – these lecture notes should in the first place be read as an invitation to further develop the framework. In this last section, we give a short overview of some of the topics that we have omitted in the main text, as well as the most exciting open directions.

**Symmetries, fermions, larger unit cells and finite size**

In the above we have exclusively dealt with translation-invariant matrix product states in the thermodynamic limit without any constraints.

First of all, in many spin chains translation invariance is spontaneously broken, where the ground state is invariant only under translations over a larger number of sites. The correct variational MPS is constructed by periodically repeating the same multi-site unit cell of tensors $\{A_1, A_2, \ldots, A_N\}$. For example, an MPS with three-site unit cell is written down as

$$|\Psi(\{A_1, A_2, A_3\})\rangle = \ldots \boxed{A_1} \boxed{A_2} \boxed{A_3} \boxed{A_1} \boxed{A_2} \boxed{A_3} \ldots . \tag{357}$$

Just as for uniform MPS, we can find canonical forms, develop ground-state optimization algorithms, and implement a quasiparticle excitation ansatz. For all details, we point the reader to Refs. [16, 43].

Another extension of the above framework consists of implementing global symmetries of the ground state on the level of the MPS tensor. This strategy is made possible by the fundamental theorem of MPS, according to which we know that if an MPS is symmetric under a global symmetry operation, the MPS tensor itself transforms under this operation:

$$U_g^{\otimes N} |\Psi(A)\rangle \propto |\Psi(A)\rangle \qquad \rightarrow \qquad \boxed{A}_{U_g} = e^{i\phi_g} \; V_g \boxed{A} V_g^\dagger . \tag{358}$$

This implies that the virtual degrees of freedom in the MPS transform under a projective representation of the symmetry group, a property that has led to the classification of symmetry-protected topological phases in one dimension using MPS [70, 71]. On the numerical level this property can be exploited to great advantage. Indeed, this symmetry property imposes a sparseness for the MPS tensor (if it is written in the correct basis), and therefore a more efficient representation of the symmetric state itself. Moreover, symmetries in the MPS tensor can also be used to label the quasiparticle ansatz with specific quantum numbers. For all details, we point the reader to Ref. [43].

The traditional method to simulate fermionic chains using MPS consists of first mapping the system to a bosonic spin chain using a Jordan-Wigner transformation. However, the uniform MPS framework allows to parametrize fermionic states on the chain directly using the formalism of super vector spaces [72]. This formalism allows to translate all of the above methods for describing interacting fermions on a chain as well.

Finally, many of the above methods have a counterpart for finite systems with open boundary conditions and without translation invariance. In that case, each tensor in the MPS is different. In particular, the time-dependent variational principle can be nicely formulated on a finite chain, allowing for simulating time evolution with arbitrary hamiltonians [27]. On a finite system momentum is no longer a good quantum number, such that the quasiparticle ansatz does not have an analog on a finite chain. Tangent-space methods can also be formulated on systems with periodic boundary conditions [73], but, just like all MPS methods, suffer from a higher computational cost.

**Real-time evolution with conserved quantities**

In Sec. 5 we have seen that the TDVP respects the conservation of energy during the time evolution, as well as other conserved quantities that commute with the tangent-space projector. This property, which is not shared by other time-dependent MPS algorithms, has been exploited recently [74] to capture the long-time dynamics of thermalizing spin chains, despite

huge truncation errors. Also, we have seen that the TDVP gives rise to an effective classical hamiltonian system with a Poisson bracket, which has recently allowed to relate the dynamics of spin chains to classical chaotic systems [75].

It remains a matter of further research to what extent conserved quantities that are not contained within the tangent space – e.g., the higher conserved quantities in integrable systems – are respected in the time evolution according to the TDVP, and whether a more generalized version can be formulated that takes more and more conserved quantities in account. Also, the relation to hydrodynamic approaches for quantum dynamics remains an important open question.

**Many-particle physics on top of an MPS**

In Sec. 6 we have shown that the tangent-space framework yields a natural language for describing elementary excitations as interacting quasiparticles on a strongly-correlated MPS background. This result suggests that these quasiparticle excitations are the relevant degrees of freedom for describing the low-energy dynamics in strongly-correlated spin chains. Therefore, we expect that the extension of the framework towards real-time and finite-temperature properties of these spin chains will prove very interesting.

In Refs. [50] and [44] it was shown that the information on the one-particle disperson relation and the two-particle S matrix makes it possibly to apply the formalism of the Bethe ansatz in an approximate way to describe the condensation of magnons in a magnetic field. This approach can be extended to out-of-equilibrium situations as well. Ideally, it would be extremely interesting to develop an interacting many-particle theory (possibly in second quantization) that describe these quasiparticles.

**cMPS**

As with MPS, cMPS are not restricted to translation-invariant systems and can easily be formulated for inhomogeneous systems, e.g. finite systems with open boundary conditions, systems with periodic boundary conditions [76] or infinite systems with a finite-length unit cell, by making the cMPS matrices $Q$ and $R$ spatially dependent. The differential equation that follows from the TDVP equation in this non-uniform setting can be interpreted as a non-commuting version of the Gross-Pitaevskii equation [67]. However, because $Q$ and $R$ will then depend on a continuous coordinate $x$, any representation in terms of a finite number of parameters will have to resort to some sort of discretization or exploit a family of basis functions. For example, the use of splines was investigated in Ref. [77].

More stringently, however, is the fact that a good optimization algorithm for such generic cMPS is lacking. This is in sharp contrast to the case of MPS, where DMRG (in one of its modern flavors) is still de facto the most robust and efficient way for optimizing a generic MPS. Ref. [78] has investigated to leverage the robustness of DMRG while trying to construct the continuum limit numerically.

More generally, many of the well-known methods from the MPS toolbox, such as simulations of local quenches, finite temperature, or non-equilibrium situations with dissipation, have no counterpart yet in terms of cMPS. Finally, in the context of tangent-space methods, we have explained how to describe single-particle excitations. Extracting scattering information by constructing variational approximations to two-particle states has so far not been addressed with cMPS, but should be a straightforward generalization of the MPS case. Any further extension that is discussed here in the context of MPS, equally applies to cMPS.

**Projected entangled-pair states**

The class of uniform matrix product states can be straightforwardly generalized to two dimensions. These states are known as projected entangled-pair states (PEPS) [79] and can be represented as

$$|\Psi(A)\rangle = \cdots \qquad \cdots \,, \tag{359}$$

where now the state is fully described by a single five-leg tensor $A$.

The variational optimization of the PEPS ground-state approximation for a given model hamiltonian is generally taken to be a hard problem. Traditionally, this is done by performing imaginary-time evolution: a trial PEPS state is evolved with the operator $e^{-\tau H}$, which should result in a ground-state projection for very long times $\tau$. This imaginary-time evolution is integrated by applying small time steps $\delta\tau$ with a Trotter-Suzuki decomposition and, after each time step, truncating the PEPS bond dimension in an approximate way. This truncation can be done by a purely local singular-value decomposition – the so-called simple-update [80] algorithm – or by taking the full PEPS wavefunction into account – the full-update [81] or fast full-update algorithm [82]. Although computationally very cheap, ignoring the environment in the simple-update scheme is often a bad approximation for systems with large correlations. The full-update scheme takes the full wavefunction into account for the truncation, but requires the inversion of the effective environment which is potentially ill-conditioned.

Recently, important steps were taken towards the formulation of tangent-space methods in two dimensions. Variational optimization schemes were introduced in Refs. [83, 84] that aim to optimize the energy density in the thermodynamic limit directly. In both approaches, an efficient summation of an infinite number of terms was needed in order to compute the gradient, similarly as we have seen in Sec. 4. A generic method for contracting overlaps of tangent vectors – a crucial ingredient in any tangent-space method – was introduced in Ref. [61], and a benchmark of the quasiparticle excitation ansatz was performed [85].

# Acknowledgements

We would like to thank Boye Buyens, Ignacio Cirac, Damian Draxler, Matthew Fishman, Christian Lubich, Michaël Mariën, Tobias Osborne, Gertian Roose, Maarten Van Damme, Bram Vanhecke and Valentin Zauner-Stauber for fruitful collaborations. This work was supported by the Flemish Research Foundation, the Austrian Science Fund (ViCoM, FoQuS), and the European Commission (QUTE 647905, ERQUAF 715861).

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
