# Peer review of "Tangent-space methods for uniform matrix product states"

_SciPost Physics Lecture Notes, doi:SciPost Phys. Lect. Notes 7 (2019)_

## Round 1 · Referee Report · Anonymous · 2019-1-6

Strengths
1- These lecture notes provide a very good introduction to tangent-space techniques for MPS.
2- Despite the technical nature of the notes, the authors provide concise descriptions of the algorithms for practical implementations.
3- Most of the paper is very pedagogical.
Weaknesses
1- It would have been interesting to add a few numerical benchmarks.
Report
The lecture notes give a good introduction to the concept of the tangent-space for uniform MPS. Some sections are technical but overall the document is very pedagogical and accessible for anyone with some knowledge of tensor network methods. This review will be very useful as an introduction to various standard algorithms that are explicitly described. My only remark is that it would have been interesting for the readers to have access to some numerical comparisons/benchmarks between methods using the tangent-space formalism and other algorithms. Therefore I recommend the publication in SciPost Lecture notes. I have a few minor remarks and have found some typos (see below).
Requested changes
1- equation (46) : there is an extra n in the second term of the right-hand site of the first line.
2- In the paragraph below eq (76) there is a typo ("recudes")
3- Just below eq (185) I think q should be replaced by p.
4- It seems that the link to arXiv of ref [57] does not send me to the right webpage.
5- Is it possible to add a comment about the feasibility (or not) of applying these methods to study periodic systems in finite-size ?

---

## Round 3 · Author Response

The authors would like to thank the referee for the careful reading of the manuscript. We have changed the manuscript according to the referee's comments, and hope that the paper can be published.

---

## Round 3 · List of Changes

In response to the referee's comments we have changed the manuscript as follows:

1- equation (46) : there is an extra n in the second term of the right-hand site of the first line -> we have omitted the extra n
2- In the paragraph below eq (76) there is a typo ("recudes") -> corrected typo
3- Just below eq (185) I think q should be replaced by p -> corrected typo
4- It seems that the link to arXiv of ref [57] does not send me to the right webpage -> this paper is published by now, we have changed the reference
5- Is it possible to add a comment about the feasibility (or not) of applying these methods to study periodic systems in finite-size ? -> we have added a sentence in the outlook concerning periodic boundary conditions, and added a reference that is relevant for this matter

---

## Editorial Decision

published